# On Empirical Risk Minimization with Dependent and Heavy-Tailed Data

Abhishek Roy[*]        Krishnakumar Balasubramanian[†]        Murat A. Erdogdu[‡]

## Abstract

In this work, we establish risk bounds for the Empirical Risk Minimization (ERM) with both dependent and heavy-tailed data-generating processes. We do so by extending the seminal works [Men15, Men18] on the analysis of ERM with heavy-tailed but independent and identically distributed observations, to the strictly stationary exponentially $\beta$-mixing case. Our analysis is based on explicitly controlling the multiplier process arising from the interaction between the noise and the function evaluations on inputs. It allows for the interaction to be even polynomially heavy-tailed, which covers a significantly large class of heavy-tailed models beyond what is analyzed in the learning theory literature. We illustrate our results by deriving rates of convergence for the high-dimensional linear regression problem with dependent and heavy-tailed data.

## 1 Introduction

Given a random vector $(X, Y) \in \mathbb{R}^d \times \mathbb{R}$, with joint distribution $(X, Y) \sim \pi$, and a class of closed, convex set of functions $\mathcal{F} \subset L_2(\pi)$, the objective in statistical learning theory is to find the *best* function in the set $\mathcal{F}$ that maps the input $X$ to the target $Y$. The quality of this mapping is measured by a user-defined loss function $\ell : \mathbb{R} \to \mathbb{R}^+ \cup \{0\}$. The most well-studied approach for the above task is that of *risk minimization*, where the best function is defined as the one that minimizes the expected loss over the set $\mathcal{F}$:

$$f^* = \operatorname*{argmin}_{f \in \mathcal{F}} P\ell_f \coloneqq \operatorname*{argmin}_{f \in \mathcal{F}} \mathbb{E}_\pi \left[ \ell \left( f(X) - Y \right) \right].$$

The above problem requires the knowledge of the distribution $\pi$ which is typically unknown in practice. However, we are usually given observations $Z_i = (X_i, Y_i)$ for $i = 1, \ldots, N$, from the distribution $\pi$ which leads to the *Empirical Risk Minimization* (ERM) procedure defined as

$$\hat{f} = \operatorname*{argmin}_{f \in \mathcal{F}} P_N \ell_f \coloneqq \operatorname*{argmin}_{f \in \mathcal{F}} \frac{1}{N} \sum_{i=1}^{N} \ell \left( f(X_i) - Y_i \right).$$

The convergence of the empirical risk minimizer $\hat{f}$ to the true risk minimizer $f^*$ is typically analyzed by considering the underlying empirical process, a topic which dates back to the seminal work of [VC71]; see also [VDVW96, vdG00, BBM05, Kol06, Kol11]. In a representative analysis in this

---

[*]Department of Statistics, University of California, Davis.abroy@ucdavis.edu. Research of this author was supported in part by NSF TRIPODS grant CCF-1934568

[†]Department of Statistics, University of California, Davis. kbala@ucdavis.edu. Research of this author was supported in part by UC Davis CeDAR (Center for Data Science and Artificial Intelligence Research) Innovative Data Science Seed Funding Program.

[‡]Department of Computer Science and Department of Statistical Sciences at the University of Toronto, and Vector Institute. erdogdu@cs.toronto.edu. Research of this author was supported in part by NSERC Grant [2019-06167], Connaught New Researcher Award, CIFAR AI Chairs program, and CIFAR AI Catalyst grant

setting, a majority of the works assume the observations $Z_i$ are generated independent and identically distributed (iid) from $\pi$, and the analysis is based on uniform concentration. However, there are important limitations associated with this approach, particularly due to the (Talagrand's) contraction principle which naturally requires a Lipschitz loss function (see, for example, [LT13, Corollary 3.17] or [Kol11, Theorem 2.3]). As a result, in order to work with standard (unbounded) loss functions such as squared-error loss or Huber loss, it is generally assumed that the range of $f \in \mathcal{F}$ is uniformly bounded and/or the noise $\xi := Y - f(X)$ is also uniformly bounded $\pi$-almost surely.

Several attempts have been made in the literature to overcome the limitations of the standard ERM analysis. A significant progress was made by Mendelson [Men15, Men18], who proposed the so-called *learning without concentration* framework for analyzing ERM procedures with unbounded noise or loss functions. The approach is based on a combination of small-ball type assumption on the input samples $X_i$, along with developing multiplier empirical process inequalities under weaker moment assumptions. We refer the interested reader, for example, to [Men17b, Men17a, LM18, LRS15, GM20] for details. The aforementioned works, while relaxing the prior analysis of ERM to handle heavy-tailed data-generating process (DGP), still require the more stringent iid assumption for their analysis. This restricts the practical applicability of the developed theoretical results significantly. Indeed, heavy-tailed and dependent data appear naturally in various practical learning scenarios [BF89, JM01, DKBR07]; however, theoretical guarantees are still missing.

**Our Contributions:** Aiming to fill the above gap, we analyze ERM with convex loss functions (that are locally strongly-convex around the origin) when the DGP is both heavy-tailed and non-iid. We do so by extending the small-ball technique of [Men15, Men18] to the strictly stationary exponentially $\beta$-mixing data. In the iid case, the interaction between the noise and the inputs is handled by an analysis based on multiplier empirical process. However, developing similar techniques in the non-iid case is fundamentally restrictive due to the limitations of the analysis based on empirical process. We side-step this issue for the non-iid case by directly making assumptions on the interaction, which allows for it to be either exponentially or polynomially heavy-tailed. For the exponentially heavy-tailed interactions, we leverage the concentration inequalities developed by [MPR11]. For the polynomially heavy-tailed case, we develop new concentration inequalities extending the recent work [BMdlP20] to $\beta$-mixing random variables. We illustrate our results in the context of ERM with sparse linear function class and stationary $\beta$-mixing DGP under both squared and Huber loss.

**Motivation:** A natural question arises in this context: *Why study ERM with convex loss functions when the DGP is heavy-tailed?* Firstly, convex loss functions cover a large class of robust loss function that are tailored to deal with the heavy-tailed behavior present in the noise and/or input data. Some examples include the Huber loss [Hub92], conditional value-at-risk [RU02, RS06, MGW20, SY20] and the so-called spectral risk measures [Ace02, HH21]. While there exist studies for nonconvex loss functions suited for heavy-tailed input data (for example, [Loh17]), such analyses are mostly in a model-based setting and focus on estimation error. Secondly, while alternatives to ERM have also been proposed and analyzed in the literature for the iid case (with the most prominent one being the median-of-means framework and its variants [MM19, LM19, LL20, BM21]), it is not immediately clear how to extend such methods to the dependent DGP that we consider in this paper. We view our work as taking the first step in developing risk bounds for statistical learning when the DGP is both heavy-tailed and dependent.

**Related Works:** The seminal work [Yu94] extended the analysis based on empirical process to the stationary mixing process using a blocking technique. [Irl97] and [BR97] studied consistency of non-parametric regression methods under mixing and exchangeability conditions on the DGP, respectively. [Nob99] established lower bounds to achieving consistency when learning from dependent data. [SHS09] studied consistency of ERM with $Z_i$ being an $\alpha$-mixing (not necessarily stationary) process, when $\mathcal{F}$ is a reproducing kernel Hilbert space. More recently, [Han21] and [DT20] studied learnability under a general stochastic process setup. The works [AV90, BL97, Pes10, Gam03] extend Valiant's Probably Approximately Correct (PAC) learning model to Markovian and related drifting DGP, assuming bounded loss function and/or noise to obtain rates of convergence. Furthermore, [ZCY12] and [HS14] analyzed ERM for least-squares regression (with bounded noise) with clipped loss functions and an $\alpha$-mixing DGP. Rademacher complexity results for predominantly stationary dependent processes were developed by [MS11] and [MR08]. [RSS10] and [ALW13] developed PAC-Bayes bounds in the non-iid setting. [RST15, RS14] developed notions of sequential Rademacher complexity to characterize the complexity in online nonparametric learning in the worst-case. More recently [DDDJ19, KDD$^+$21] considered learning under weakly-dependent data for specific models.

However, such works mainly rely on bounded loss functions in their analysis. Furthermore, [Mei00] and [AW12] studied model selection for time series forecasting in a possibly unbounded setup but do not consider conditional prediction and is limited to light-tailed cases.

Apart from the aforementioned works, the recent works [KM17, HW19, WLT20] are closely related to our setup as they consider rates of convergence of ERM under heavy-tailed and dependent DGP. In [KM17, Section 8], generalization bounds are developed when $Z_i$ is an asymptotically stationary $\beta$-mixing sequence. However, their conditions on the function class $\mathcal{F}$ are rather opaque and it is not clear if their method actually handles the heavy-tailed DGP that we focus on. [HW19] considered a setup based on a statistical model: for $i = 1, \ldots, N$, $Y_i = f^*(X_i) + \epsilon_i$, with the following conditions: (i) $\epsilon_i$ being independent of $X_i$, (ii) $X_i$ is iid, and (iii) $\epsilon_i$ is iid. For this setting, they assumed that the noise has a bounded $p$-th moment (with $p \geq 2$) and $\mathcal{F}$ satisfies the standard entropy condition (see, for example [Kol06, Example 4]) with exponent $\alpha \in (0, 2)$ and obtained convergence rates of the order $O(N^{-\frac{1}{2+\alpha}} + N^{-\frac{1}{2} + \frac{1}{2p}})$. Finally, [WLT20] provides an analysis of $L_1$-regularized ERM with quadratic loss and linear function class where the heavy-tailed behavior is induced by a sub-Weibull assumption, and data dependency is characterized by a stationary $\beta$-mixing condition. However, their analysis specializes to sparse linear function classes and their focus is on parameter estimation error and in-sample prediction accuracy.

## 2 Assumptions and Preliminaries

We assume that there exists a function $f^* \in \mathcal{F}$ that minimizes the population risk $\mathbb{E}[\ell(f(X) - Y)]$. In what follows, we provide the conditions that we require on the DGP, specifically, the exponentially $\beta$-mixing condition, for characterizing the dependency among data points.

**Definition 1** ([Yu94]). *Suppose that $\{Z_i\}_{i=-\infty}^{\infty}$ is a strictly stationary sequence of random variables. For any $i, j \in \mathbb{Z} \cup \{-\infty, \infty\}$, let $\sigma_i^j$ denote the $\sigma$-algebra generated by $\{Z_b\}_{b=i}^{j}$. Then for any positive integer $b$, the $\beta$-mixing coefficient of the stochastic process $\{Z_i\}_{i=-\infty}^{\infty}$ is defined as*

$$\beta(b) = \sup_n \mathbb{E}_{B \in \sigma_{-\infty}^n} \left[ \sup_{A \in \sigma_{n+b}^{\infty}} |\mathbb{P}(A \mid B) - \mathbb{P}(A)| \right].$$

*The sequence $\{Z_i\}_{i=-\infty}^{\infty}$ is said to be $\beta$-mixing if $\beta(b) \to 0$ as $b \to \infty$. Furthermore, it is said to be exponentially $\beta$-mixing if there exist $\beta_0, \beta_1, r > 0$ such that $\beta(b) \leq \beta_0 \exp(-\beta_1 b^r)$ for all $b$.*

The $\beta$-mixing condition is frequently used when studying non-iid DGP, and imposes a dependence structure between data samples that weakens over time. The coefficient $\beta(b)$ is a measure of the dependence between events that occur within $b$ units in time. Indeed, $\beta$-mixing is often used in the analysis of non-iid data in statistics and machine learning, [Vid13]. Before stating our assumptions formally, we present the following decomposition of empirical risk for a convex loss using Taylor's expansion:

$$P_N \ell_f \geq \frac{1}{16N} \sum_{i=1}^{N} \ell''(\widetilde{\xi}_i)(f - f^*)^2(X_i) + \frac{1}{N} \sum_{i=1}^{N} \ell'(\xi_i)(f - f^*)(X_i), \tag{1}$$

where $\widetilde{\xi}_i$ is a suitably chosen midpoint between $f(X_i) - Y_i$ and $f^*(X_i) - Y_i := \xi_i$. For quadratic loss functions $\ell(t) = t^2$, $\ell'(\xi_i) = 2\xi_i$ and $\ell''(\widetilde{\xi}_i) = 2$, $\forall i$. At a high level, establishing risk bounds boils down to proving a positive lower bound on the second term on the Right Hand Side (RHS) of (1), and a concentration result for the first term on the RHS with high probability. We now introduce the precise assumptions we make on the DGP and the function class to formalize the above strategy. For the sake of clearer exposition, we first introduce our assumptions in the context of quadratic loss and then indicate the changes required to handle more general locally strongly-convex loss functions.

**Assumption 2.1** (Squared loss). *The DGP $\{Z_i\}_{i=-\infty}^{\infty}$ and the function class $\mathcal{F}$ satisfy the following:*

(a) *$\beta$-**mixing data**. The process $\{Z_i\}_{i=-\infty}^{\infty}$ is a strictly stationary exponentially $\beta$-mixing sequence, i.e., $\beta(k) \leq \exp(-ck^{\eta_1})$, for some $c, \eta_1 > 0$, with strict stationary distribution $\pi$.*

(b) ***Small ball condition**. Let $\mathcal{F} \subset L_2(\pi)$ be closed, convex class of functions and define $\mathcal{F} - \mathcal{F} := \{f - h : f, h \in \mathcal{F}\}$. Then, the function class $\mathcal{F}$ is such that there is a $\tau > 0$ for which $Q_{\mathcal{F} - \mathcal{F}}(2\tau) > 0$, where $Q_{\mathcal{H}}(u) = \inf_{h \in \mathcal{H}} \mathbb{P}(|h| \geq u \|h\|_{L_2})$.*

*(c)* ***Deviations of interaction.*** *The stationary noise $\xi_1$ and the error $(f - f^*)(X_1)$ satisfy either:*

    *(i) For all $f \in \mathcal{F}$, and $Z_1 \sim \pi$, for some $\eta_2 > 0$ we have*

$$\mathbb{P}\left(|\xi_1(f - f^*)(X_1) - \mathbb{E}\left[\xi_1(f - f^*)(X_1)\right]| \geq t\right) \leq \exp(1 - t^{\eta_2}). \tag{2}$$

    *or,*

    *(ii) For all $f \in \mathcal{F}$, and $Z_1 \sim \pi$, for some $\eta_2 > 2$, and $c > 0$ we have*

$$\mathbb{P}\left(|\xi_1(f - f^*)(X_1) - \mathbb{E}\left[\xi_1(f - f^*)(X_1)\right]| \geq t\right) \leq ct^{-\eta_2}. \tag{3}$$

*(d)* ***Heavy-tail/data-mixing trade-off.*** *Under condition (c)-(i), $1/\eta := 1/\eta_1 + 1/\eta_2 > 1$.*

Condition (a) above, exponentially $\beta$-mixing data, has been assumed in various works, see for example [Vid13, WLT20, KM17], to obtain rates of convergence for ERM procedures in general. Indeed, (exponential) mixing assumption holds in several time-series applications. For example, [Mok88] showed that certain ARMA processes can be modeled as an exponentially $\beta$-mixing stochastic process. Furthermore [VK06] showed that globally exponentially stable unforced dynamical systems subjected to finite-variance continuous density input noise give rise to exponentially mixing stochastic process; see also [FS12]. Condition (b), referred to as the well-known small-ball condition, has been previously employed in the `iid` case [Men15]. Intuitively, it models *heavy-tailedness* by restricting the mass allowed near any small neighborhoods of zero; thus, forcing the tails to be necessarily heavy. To our knowledge, the small-ball condition has not been used under dependent DGP assumptions. Condition (c) is proposed in this work as a way to model the interaction between the stationary noise $\xi_1$ and the stationary error $(f - f^*)(X_1)$. For the `iid` setting, [Men15] modeled the interaction between $\xi_1$ and $(f - f^*)(X_1)$ uniformly over the class of $\mathcal{F}$ via the multiplier empirical process and captured the complexity through a parameter $\alpha_N$ (see (108) for the definition). This requires using symmetrization argument in the proof which is not applicable in the non-`iid` setting that we consider in this work. In addition, how different tail conditions on the data and noise affect the high-probability statement on the learning rate is not apparent from the parameter $\alpha_N$. We revisit the relationship between our condition (in the context of `iid` observations) and the multiplier empirical process approach used in [Men15] in Section F. Finally, condition (d) models the relationship between the allowed degree of dependency and the allowed degree of interaction between $\xi_1$ and $(f - f^*)(X_1)$.

Next, we modify condition (c) in Assumption 2.1 for the case of general locally strongly-convex loss functions because now the interaction part involves $\ell'(\xi)$ instead of $\xi$ (recall the decomposition (1)).

**Assumption 2.2** (Convex loss). *When the loss function is locally strongly-convex around the origin and globally convex, condition (c) in Assumption 2.1 is modified as:*

*(c)* ***Deviations of interaction.*** *The stationary noise $\xi_1$ and the error $(f - f^*)(X_1)$ satisfy either,*

    *(i) For all $f \in \mathcal{F}$, and $Z_1 \sim \pi$, for some $\eta_2 > 0$ we have*

$$\mathbb{P}\left(|\ell'(\xi_1)(f - f^*)(X_1) - \mathbb{E}\left[\ell'(\xi_1)(f - f^*)(X_1)\right]| \geq t\right) \leq \exp(1 - t^{\eta_2}). \tag{4}$$

    *or,*

    *(ii) For all $f \in \mathcal{F}$, and $Z_1 \sim \pi$, for some $\eta_2 > 2$, and $c > 0$ we have*

$$\mathbb{P}\left(|\ell'(\xi_1)(f - f^*)(X_1) - \mathbb{E}\left[\ell'(\xi_1)(f - f^*)(X_1)\right]| \geq t\right) \leq ct^{-\eta_2}. \tag{5}$$

**Complexity Measures:** We now introduce the complexity measures that play a crucial role in characterizing the rates of convergence. The use of $\beta$-mixing assumption enables us to define complexity measures based on the blocking technique proposed by [Yu94], also utilized by the works of [MR08, KM17, WLT20]. We partition the training sample of size $N$, $S := \{Z_i\}_{i=1}^N$, into two sequences of blocks $S_a$ and $S_b$. Each block in $S_a$, and $S_b$ is of length $a$, and $b$ respectively. $S_a$ and $S_b$, both are of length $\mu$, i.e., $\mu(a + b) = N$. Formally, $S_a$ and $S_b$ are given by

$$S_a = \left(Z_1^{(a)}, Z_2^{(a)}, \cdots, Z_\mu^{(a)}\right) \quad \text{with } Z_i^{(a)} = \{z_{(i-1)(a+b)+1}, \cdots, z_{(i-1)(a+b)+a}\},$$

$$S_b = \left(Z_1^{(b)}, Z_2^{(b)}, \cdots, Z_\mu^{(b)}\right) \quad \text{with } Z_i^{(b)} = \{z_{(i-1)(a+b)+a+1}, \cdots, z_{(i-1)(a+b)+a+b}\}. \tag{6}$$

Based on this blocking technique, we require the following definition of Rademacher complexity.

**Definition 2.** *Let $\{\widetilde{X}_i\}_{i=1}^{\mu}$ be an **iid** sample from the strict stationary distribution $\pi$. Let $\mathcal{D}$ be the unit-$L_2(\pi)$ ball centered at $f^*$. For every $\gamma > 0$, define*

$$\omega_\mu(\mathcal{F} - \mathcal{F}, \gamma) := \inf \left\{ r > 0 : \mathbb{E} \left[ \sup_{h \in (\mathcal{F}-\mathcal{F}) \cap r\mathcal{D}} \left| \frac{1}{\mu} \sum_{i=1}^{\mu} \epsilon_i h(\widetilde{X}_i) \right| \right] \leq \gamma r \right\}, \qquad (7)$$

*where $\{\epsilon_i\}_{i=1}^{\mu}$ are **iid** Rademacher variables taking values $\pm 1$ with probability $1/2$.*

The quantity $\omega_\mu(\mathcal{H}, \gamma)$ provides a localized complexity measure for the function class $\mathcal{F}$, and serves as a generalization of the standard Rademacher complexity in the non-**iid** setting. For the case of locally strongly-convex losses, we need the following related measures of complexity.

**Definition 3.** *For a function class $\mathcal{H} \subset L_2(\pi)$, a sample of size $N$ from a strictly stationary $\beta$-mixing sequence with stationary distribution $\pi$ satisfying Assumption 2.1-(a), and $\zeta_1, \zeta_2 > 0$, we define:*

$$\omega_1(\mathcal{H}, N, \zeta_1) = \inf \left\{ r > 0 : \mathbb{E}\left[ \|G\|_{\mathcal{H} \cap rD} \right] \leq \zeta_1 r N^{\frac{\eta_1}{2(1+\eta_1)}} \right\} \quad \text{and} \quad \omega_2(\mathcal{H}, \mu, \zeta_2) = \omega_\mu(\mathcal{H}, \zeta_2),$$

*where $\mu = N^{\frac{\eta_1}{(1+\eta_1)}}$, $\|G\|_{\mathcal{H}} = \sup_{h \in \mathcal{H}} G_h$, and $\{G_h : h \in \mathcal{H}\}$ is the canonical Gaussian process indexed by $\mathcal{H}$ with a covariance induced by $L_2(\pi)$. Moreover, we let*

$$\omega_Q(\mathcal{F} - \mathcal{F}, N, \zeta_1, \zeta_2) := \max(\omega_1(\mathcal{H}, N, \zeta_1), \omega_2(\mathcal{H}, \mu, \zeta_2)).$$

Note that in Definitions 2 and 3, the scaling is in terms of number of blocks $\mu$ instead of $N$. The number of blocks $\mu$ can be thought of as the effective sample size under dependency, and as $\eta_1 \to \infty$, one has $\mu \to N$. The term $\mathbb{E}\left[ \|G\|_{\mathcal{H} \cap rD} \right]$ appearing in Definition 3 is termed as the localized Gaussian width and is also a widely used complexity measure in the literature. Note that while in the case of quadratic loss function, the bound is in terms of local Rademacher-based complexity measure, whereas in the convex case, we require both Gaussian and Rademacher-based complexity measures to establish the bound, mostly due to technical reasons in the proof. An intuitive explanation for this has eluded us thus far; see also [Men18, Lemma 4.2].

**Concentration Inequalities for Heavy-tails:** We now restate [MPR11, Theorem 1], in a form adapted to our setting below. This result is required to handle interactions of noise and input that satisfy condition (c)-(i) of Assumption 2.1. It is straightforward to check that the conditions required by [MPR11] are immediately satisfied under our Assumption 2.1. Indeed, while the results in [MPR11] are stated for $\tau$-mixing sequences, condition (a) in Assumption 2.1 implies that the process $\{Z_i\}_{i=-\infty}^{\infty}$ is exponentially $\tau$-mixing [CG14], i.e., for a constant $c' > 0$, $\tau(k) \leq e^{-c'k^{\eta_1}}$.

**Lemma 2.1** ([MPR11]). *Let $\{W_j\}_{j \geq 1}$ be a sequence of zero-mean real-valued random variables satisfying conditions (a), (c)-(i), and (d) of Assumption 2.1. Define $\varkappa_M(x) = (x \wedge M) \vee (-M)$, for some $M$, where $(x \wedge y) = \min(x, y)$, and $(x \vee y) = \max(x, y)$, and set,*

$$V := \sup_{M \geq 1} \sup_{i > 0} \left( \mathrm{VAR}(\varkappa_M(W_i)) + 2 \sum_{j > i} |\mathrm{COV}(\varkappa_M(W_i), \varkappa_M(W_j))| \right). \qquad (8)$$

*Note that $V$ is finite. Then, for any $N \geq 4$, there exist positive constants $C_1, C_2, C_3$, and $C_4$ depending only on $c, \eta_1, \eta_2$ such that, for any $t > 0$, we have*

$$\mathbb{P}\left( \sup_{j \leq N} \left| \sum_{i=1}^{j} W_i \right| \geq t \right) \leq N e^{-\frac{t^\eta}{C_1}} + e^{-\frac{t^2}{C_2 N V}} + e^{-\frac{t^2}{C_3 N} \exp\left( \frac{t^{\eta(1-\eta)}}{C_4 (\log t)^\eta} \right)}. \qquad (9)$$

To deal with polynomially tailed interactions, i.e., under condition (c)-(ii) of Assumption 2.1, we prove a concentration inequality for the sum of exponentially $\beta$-mixing random variables with polynomially heavy-tails, which may be of independent interest.

**Lemma 2.2** (Concentration for heavy-tailed $\beta$-mixing sum). *Let $\{W_j\}_{j \geq 1}$ be a sequence of zero-mean real valued random variables satisfying conditions (a) and (c)-(ii) of Assumption 2.1, for some $\eta_2 > 2$. Then for any positive integer $N$, $0 \leq d_1 \leq 1$, and $d_2 \geq 0$, and for any $t > 1$, we have,*

$$\mathbb{P}\left( \sup_{j \leq N} \left| \sum_{i=1}^{j} W_i \right| \geq t \right) \leq \frac{2^{\eta_2+3}}{(d_2 \log t)^{\frac{1-\eta_2}{\eta_1}}} \frac{N}{t^{(1+d_1(\eta_2-1))}} + 8 \frac{N}{t^{(1+c'd_2)}} + 2e^{-\frac{t^{2-2d_1}(d_2 \log t)^{1/\eta_1}}{9N}}, \qquad (10)$$

*where $c' > 0$ is a constant.*

Note that we do not need condition (d) of Assumption 2.1 for Lemma 2.2. Since the tail probabilities decay polynomially and the mixing coefficients decay exponentially fast, the effect of heavy-tail dominates and hence, there is no trade-off between $\eta_1$ and $\eta_2$. Lemma 2.2 extends the results of [BMdlP20] (on iid heavy-tailed random variables) to the exponentially $\beta$-mixing setting. The results of [BMdlP20] show that even for the iid case, the tail probability of the sum decays polynomially with $t$. We show that similar polynomial tail bounds can be obtained (up to log factors) even in the dependent setting. Furthermore, when $\beta(k) = 0, k > 0$, the sequence is iid, in which case we recover the result of [BMdlP20]. We also remark that the above two results are crucial to derive our convergence rates for ERM. As a preview, when the DGP has only $(2 + \delta)$-moments, $\delta > 0$, Lemma 2.2 leads to risk bounds that hold with polynomial probability, whereas for sub-Weibull DGP (see Definition 4), Lemma 2.1 would lead to risk bounds that hold with exponential probability.

# 3 Main Results

In this section, we state our main results on the rates of convergence of ERM for both squared and convex loss functions. First, we consider the squared loss.

**Theorem 3.1** (Rates of ERM with squared loss). *Consider the ERM procedure with the squared error loss. For $\tau_0 < \tau^2 Q_{\mathcal{H}}(2\tau)/8$, setting $\mu = N^r Q_{\mathcal{H}}(2\tau) c^{\frac{1}{\eta_1}}/4$, for some constants $c, c' > 0$, and $0 < r < 1$, we have, for sufficiently large $N$, and some positive constants $\widetilde{C}_1, \widetilde{C}_2$, the following:*

1. *Under conditions (a), (b), (c)-(i), and (d) of Assumption 2.1, for $0 < \iota < 1/4$,*

$$\|\hat{f} - f^*\|_{L_2} := \left( \int (\hat{f} - f^*)^2 d\pi \right)^{\frac{1}{2}} \leq \max\left\{ N^{-\frac{1}{4}+\iota}, \omega_\mu\left( \mathcal{F} - \mathcal{F}, \frac{\tau Q_{\mathcal{F}-\mathcal{F}}(2\tau)}{16} \right) \right\}, \quad (11)$$

   *with probability at least (for $V$ as defined in (8))*

$$1 - \widetilde{C}_1 N^r Q_{\mathcal{H}}(2\tau) c^{\frac{1}{\eta_1}} \exp(-N^{(1-r)\eta_1}) - \widetilde{C}_2 N \exp\left( -(N^{\frac{1}{2}+2\iota}\tau_0)^\eta \right). \quad (12)$$

2. *Under conditions (a), (b), and (c)-(ii) of Assumption 2.1, for $0 < \iota < (1 - 1/\eta_2)/4$,*

$$\|\hat{f} - f^*\|_{L_2} \leq \max\left\{ N^{-\frac{1}{4}\left(1-\frac{1}{\eta_2}\right)+\iota}, \omega_\mu\left( \mathcal{F} - \mathcal{F}, \frac{\tau Q_{\mathcal{F}-\mathcal{F}}(2\tau)}{16} \right) \right\}. \quad (13)$$

   *with probability at least*

$$1 - \widetilde{C}_1 N^r Q_{\mathcal{H}}(2\tau) c^{\frac{1}{\eta_1}} \exp(-N^{(1-r)\eta_1}) - \widetilde{C}_2 \tau_0^{-\frac{2\eta_2}{1+\eta_2}} N^{-\frac{4\iota\eta_2}{1+\eta_2}}. \quad (14)$$

*The detailed expression of the probabilities are provided in the Appendix (Theorem B.1).*

**Remark 1.** *To the best of our knowledge, the above result is the first result on understanding rates of convergence of ERM with squared error loss functions with unbounded noise (as well as the loss) for heavy-tailed dependent data. For a wide range of function classes $\mathcal{F}$ used in practice (see Section 4), the dominant term in the rate of convergence is $N^{-\frac{1}{4}+\iota}$, for $0 < \iota < 1/4$. Furthermore, under the stronger condition (c)-(i) of Assumption 2.1, the risk bound holds with exponential probability, whereas under the weaker condition (c)-(ii), it holds only with polynomial probability.*

We now show that when the small-ball condition in Assumption 2.1 is replaced with the stronger norm-equivalence assumption, also considered in [MZ20], one could obtain improved rates. Examples of random vectors that satisfy the norm-equivalence conditions include multivariate student $t$-distribution and sub-exponential random variables. We refer to Section 4 for illustrative examples.

**Assumption 3.1** ($L_p - L_2$ norm-equivalence). *Let $\mathcal{F} \subset L_q(\pi)$ be a class of functions for some $q \geq 3$. The function class $\mathcal{F} - \mathcal{F} = \{f - h : f, h \in \mathcal{F}\}$ is $L_p - L_2$ norm-equivalent for some $p > 2$, if there exists an $M_1 > 0$ such that, $\|h\|_{L_p} := (\int |h|^p d\pi)^{1/p} \leq M_1 \|h\|_{L_2}, \forall h \in \mathcal{F} - \mathcal{F}$.*

**Corollary 3.1.** *For the ERM procedure with squared error loss, under Assumptions 2.1, with condition (b) replaced by Assumption 3.1 with $p = 8$, for some $0 < \iota < \frac{1}{2}$ and $r, \mu$ and $\tau_0$ same as in Theorem 3.1, for sufficiently large $N$, we have*

$$\|\hat{f} - f^*\|_{L_2} \leq \max\left\{ N^{-\frac{1}{2}+\iota}, \omega_\mu(\mathcal{F} - \mathcal{F}, \tau Q_{\mathcal{F}-\mathcal{F}}(2\tau)/16) \right\}, \quad (15)$$

*with probability at least (for some constants $\widetilde{C}_1$ and $\widetilde{C}_2$)*

$$1 - \widetilde{C}_1 N^r Q_{\mathcal{H}}(2\tau) c^{\frac{1}{\eta_1}} \exp(-N^{(1-r)\eta_1}) - \widetilde{C}_2 N \exp\left( -(N^{2\iota}\tau_0)^\eta / M_1 \right).$$

**Remark 2.** *In the model-based nonparametric regression setting (as discussed in **Related Works**) with $X_i$ allowed to be dependent on $\xi_i$ for all $i = 1, \ldots, n$, but $(X_i, \xi_i)$ being $iid$, in [HW19, Proposition 3] authors show a lower bound of $N^{-\frac{1}{2+\epsilon}}$ for some $\epsilon > 0$, for sufficiently heavy-tailed input $X_i$. The above result provides an upper bound of similar order, for a more general setting in comparison to [HW19]. We also remark that for Corollary 3.1, in the model-based setting, if we assume that $\xi_i$ is independent of $X_i$, we have the same conclusion with just $p = 4$ instead of $p = 8$. Furthermore, note that in part 2 of Theorem 3.1, $L_8$ norm does not exist for $\eta_2 \leq 8$. We have elaborated more on how this result compares Theorem 3.1 in [Men15] later in Section F.*

We now present our results for the class of convex loss functions that are locally strongly-convex.

**Assumption 3.2** (Convex loss). *The loss function $\ell : \mathbb{R} \to \mathbb{R}^+ \cup \{0\}$ is a convex loss function which is strongly convex in the neighborhood of $0$, i.e., there exists a $t_2 > 0$ such that for any $x, y \in [-t_2, t_2]$, $\ell(y) \geq \ell(x) + \ell(x)'(y-x) + \mu_c(y-x)^2/2$ for some constant $\mu_c > 0$.*

**Theorem 3.2** (Rates of ERM with convex loss). *Consider ERM with loss functions that satisfy Assumption 3.2. For $\tau_0 < c_2 Q_{\mathcal{F}-\mathcal{F}}(2\tau)\rho(0, t_2)\tau^2$, $t_2 = \mathcal{O}((\kappa_0 + 1/\sqrt{Q_{\mathcal{H}}(2\tau)})\|\xi\|_{L_2})$, setting $\mu = N^{\eta_1/(1+\eta_1)}$, for some constants $c, c' > 0$, we have, for any $N \geq 4$, the following:*

1. *Under conditions (a), (b), (c)-(i), and (d) of Assumption 2.1, for $0 < \iota < \frac{1}{4}$,*

$$\|\hat{f} - f^*\|_{L_2} \leq \max\left\{ N^{-\frac{1}{4}+\iota}, 2\omega_Q(\mathcal{F} - \mathcal{F}, N, \zeta_1, \zeta_2) \right\}, \tag{16}$$

   *with probability at least (for $V$ is defined in (8) and some positive $c_9, c_{10}, \widetilde{C}_3$)*

$$1 - c_9 Q_{\mathcal{H}}(2\tau)^{1-\frac{1}{\eta_1}} N^{\eta_1/(1+\eta_1)} e^{-c_{10} Q_{\mathcal{H}}(2\tau)^{1+\frac{1}{\eta_1}} N^{\frac{\eta_1}{1+\eta_1}}} - \widetilde{C}_3 N \exp\left( -(N^{\frac{1}{2}+2\iota}\tau_0)^{\eta}/C_1 \right).$$

2. *Under conditions (a), (b), and (c)-(ii) of Assumption 2.1, for $0 < \iota < (1 - 1/\eta_2)/4$,*

$$\|\hat{f} - f^*\|_{L_2} \leq \max\left\{ N^{-\frac{(1-1/\eta_2)}{4}+\iota}, 2\omega_Q(\mathcal{F} - \mathcal{F}, N, \zeta_1, \zeta_2) \right\}, \tag{17}$$

   *with probability at least (for constants $c_9, c_{10}, \widetilde{C}_4 > 0$)*

$$1 - c_9 Q_{\mathcal{H}}(2\tau)^{1-\frac{1}{\eta_1}} N^{\eta_1/(1+\eta_1)} e^{-c_{10} Q_{\mathcal{H}}(2\tau)^{1+\frac{1}{\eta_1}} N^{\eta_1/(1+\eta_1)}} - \widetilde{C}_4 \tau_0^{-\frac{2\eta_2}{1+\eta_2}} N^{-\frac{4\iota\eta_2}{1+\eta_2}}. \tag{18}$$

To the best of our knowledge, the above result is the first result on understanding rates of convergence of ERM with convex loss functions with unbounded noise (as well as the loss) for heavy-tailed dependent data. The above result highlights the advantage of using a robust loss function, e.g. Huber loss, over a quadratic loss function. For example, if $(f - f^*)(X)$ has a sub-Weibull tail and the noise $\xi$ has polynomial tail, one can still obtain risk bounds with exponential probability. This is because in this case $\ell'(\xi)(f - f^*)(X)$ can still be sub-Weibull for a suitable chosen $\ell'(\xi)$. Such a situation arises, for example, when there are outliers even if the data is light-tailed. With a squared error loss, one won't be able to obtain a risk bound with exponential probability in this scenario. We will illustrate this in Section 4.3 through Huber loss, a popular choice of robust loss function in robust statistics. Similar to the quadratic case, we also have an improved result when the small-ball condition is replaced with the norm-equivalence condition. Due to space constraints, we state and prove it in the Appendix B.2.

## 4 Illustrative Examples

We illustrate the results of Section 3 with three examples, based on sub-Weibull random variables and Pareto random variables, that are canonical models of heavy-tailed data in the literature.

### 4.1 Example 1: $\beta$-mixing Sub-Weibull DGP with Squared Error Loss

Here, we consider sub-Weibull random variables to model the heavy-tailed behavior in the DGP.

**Definition 4** (Sub-Weibull random vectors). *A real-valued random variable $X$ is said to be sub-Weibull with parameter $\eta > 0$, if there are constants $K_1, K_2 > 0$, such that we have $\mathbb{P}(|X| > t) \leq 2\exp\left(-(t/K_1)^{\eta}\right)$, or equivalently $\|X\|_p = \mathbb{E}\left[|X|^p\right]^{1/p} \leq K_2 p^{\frac{1}{\eta}}$. Based on this, a random vector $X \in \mathbb{R}^d$ is said to be marginally sub-Weibull with parameter $\eta > 0$ if each coordinate of $X$ is sub-Weibull with $\eta$. We use $X \sim \text{sw}(\eta)$ to represent this fact.*

The above family of distributions define a rich class of random variables, allowing for heavier tails than sub-Gaussian tails ($\eta = 2$) or sub-exponential tails ($\eta = 1$). Let $\{\delta_i\}_{i \in \mathbb{Z}^+}$ be an $\texttt{iid}$ sequence of $d$-dimensional random vectors with independent coordinates with $\delta_i \sim \texttt{sw}(\eta_\delta)$. Assume that the dependent input vectors are generated according to the model

$$X_i = AX_{i-1} + \delta_i, \tag{19}$$

where $A \in \mathbb{R}^{d \times d}$ with spectral radius less than 1. For simplicity, let $A = \sigma_0^2 I_d$ where $\sigma_0^2 < 1$, and $\{Y_i\}_{i \in \mathbb{Z}^+}$ be a univariate response sequence given by $Y_i = {\theta^*}^\top X_i + \xi_i$, where $\theta^* \in B_1^d(R)$ belongs to the $\ell_1$-norm ball in $\mathbb{R}^d$ with the radius $R$, and $\{\xi_i\}_{i=1}^n$ is an i.i.d sequence independent of $X_i$ $\forall i$, and $\xi_i \sim \texttt{sw}(\eta_\xi)$ for $0 < \eta_\xi < 1$ has independent coordinates. To proceed with learning framework, we consider ERM with squared loss and the function class $\mathcal{F} := \mathcal{F}_R = \left\{ \langle \theta, \cdot \rangle : \theta \in B_1^d(R) \right\}$. We denote the difference function class $\mathcal{F}_R - \mathcal{F}_R$ by $\mathcal{H}_R$. We verify conditions (a), (b), (c), and (d) of Assumption 2.1, and Assumption 3.1, provide the rate of ERM in this setting, and compare our result to that of [WLT20] in a similar setting in Appendix C.

## 4.2 Example 2: $\beta$-mixing Pareto DGP with Squared Error Loss

Let $\widetilde{X}_{t,i}$ denote the $i$-th coordinate of the vector $\widetilde{X}_t \in \mathbb{R}^d$. We consider the process given in [Pil91]: For $i = 1, 2, \cdots, d$, $\eta_3 > 2 + 2\iota$, where $\iota > 0$ is a small number, and $t = 0, 1, \cdots$, define

$$\widetilde{X}_{t,i} = \begin{cases} 2^{\frac{1}{\eta_3}} \widetilde{X}_{t-1,i} & \text{with probability } 1/2 \\ \min\left(2^{\frac{1}{\eta_3}} \widetilde{X}_{t-1,i}, \delta_{t,i}\right) & \text{with probability } 1/2 \end{cases} \tag{20}$$

where $\{\delta_{t,i}\}_{i=1,2,\cdots,d,t=1,2,\cdots}$ is a sequence of $\texttt{iid}$ Pareto random variables with the distribution $L_+(\delta; \eta_3, d_i) = \eta_3(d_i\delta)^{\eta_3-1}/\left(1 + (d_i\delta)^{\eta_3}\right)^2$ for $\delta > 0, \eta_3 > 2 + 2\iota$, and we write $X \sim L_+(\eta, \sigma)$ to denote that $X$ is a Pareto random variable with parameters $\eta$ and $\sigma$. The survival function of $\delta \sim L_+(\eta_3, d)$ is given by, $\mathbb{P}(\delta > t) = (1 + (dt)^{\eta_3})^{-1}$, for $t > 0$. Let $\widetilde{X}_{1,t}$ and $\widetilde{X}_{2,t}$ be two independent trails of the process in (20). Let $\{U_t\}_{t=0,1,\cdots}$ be a sequence of $\texttt{iid}$ $U[0,1]$ random variables. Now consider the process $X_{t,i} = \widetilde{X}_{1,t,i} \mathbb{1}(U_t \leq 1/2) - \widetilde{X}_{2,t,i} \mathbb{1}(U_t > 1/2)$. The marginal distribution of $X_{t,i}$ is given by symmetric Pareto distribution, i.e.,

$$L(x; \eta_3, d_i) = \frac{\eta_3(d_i|\delta|)^{\eta_3-1}}{2\left(1 + (d_i|\delta|)^{\eta_3}\right)^2} \quad -\infty < \delta < \infty, \eta_3 > 2 + 2\iota. \tag{21}$$

Now, let $\{Y_i\}_{i \in \mathbb{Z}^+}$ is given by $Y_i = {\theta^*}^\top X_i + \upsilon_i$, where $\theta^* \in B_1^d(R)$, as before. Let $\{\upsilon_i\}_{i=1}^n$ be an $\texttt{iid}$ sequence of $0$ mean random variables independent of $X_i$ for all $i$, with heavy tails such that for all $t > 0$, $\mathbb{P}\left(|\upsilon_i| \geq t\right) \leq 1/(1 + t^{\eta_4}), \eta_4 > 2 + 2\iota$. Like in Example 4.1, we consider ERM with squared loss and the function class $\mathcal{F} := \mathcal{F}_R = \left\{ \langle \theta, \cdot \rangle : \theta \in B_1^d(R) \right\}$, where $B_1^d(R)$ denotes the $d$-dimensional $\ell_1$-ball with radius $R$. We denote the difference function class $\mathcal{F}_R - \mathcal{F}_R$ by $\mathcal{H}_R$. We show that conditions (a), (b), and (3)-(ii) of Assumption 2.1 hold here in the following section.

### 4.2.1 Verification of Assumption 2.1 for Example 4.2

[Pil91] shows that the AR(1) process given by (20) is strictly stationary if $\tilde{X}_{0,i} \sim L_+(\eta_3, d_i)$, and $\tilde{X}_{0,i}$ are independent of each other for $i = 1, 2, \cdots, d$. The stationary distribution is given by $L_+(\eta_3, d_i)$. Let $\pi(X_{t_1}, X_{t_2}, \cdots, X_{t_n})$ be the joint distribution of $X_{t_1}, X_{t_2}, \cdots, X_{t_n}$ for a set of time points $t_1, t_2, \cdots, t_n$. Now for any positive integer $k$,

$$\pi(\{X_{t_i+k}\}_{i=1}^n) = \pi(\{\tilde{X}_{1,t_i+k} \mathbb{1}(U_{t_i+k} \leq 1/2) - \tilde{X}_{2,t_i+k} \mathbb{1}(U_{t_i+k} > 1/2)\}_{i=1}^n)$$
$$= \pi(\{\tilde{X}_{1,t_i} \mathbb{1}(U_{t_i} \leq 1/2) - \tilde{X}_{2,t_i} \mathbb{1}(U_{t_i} > 1/2)\}_{i=1}^n) = \pi(\{X_{t_i}\}_{i=1}^n). \tag{22}$$

The second equality above follows from the fact $\tilde{X}_{1,t}$ and $\tilde{X}_{2,t}$ are strictly stationary process and $\{U_t\}$ is $\texttt{iid}$. So $X_t$ is a strictly stationary process with marginal distribution of $X_{t,i}$ given by symmetric Pareto distribution

$$L(x; \eta_3, d_i) = \frac{\eta_3(d_i|\delta|)^{\eta_3-1}}{2\left(1 + (d_i|\delta|)^{\eta_3}\right)^2} \quad -\infty < \delta < \infty, \eta_3 > 2 + 2\iota. \tag{23}$$

Without loss of generality we will assume that $d_1 \leq d_2 \leq \cdots \leq d_d$, $\sum_{i=1}^d 1/d_i = K_0$, and $d_1 \geq C_6'$ for some constants $K_0, C_6' > 0$. It is shown in Lemma 1 of [Ris08], that the AR(1) process in (20) is $\phi$-mixing with $\phi(k) = k \log 2/(2^k - 1)$, $k = 1, 2, \cdots$. where $\phi$-mixing coefficients are defined as in [Bra05]. We also have $\beta(k) \leq \phi(k) \leq e^{-k/3}$. Since $X_t$ depends only on $X_{1,t}$, $X_{2,t}$, and $U_t$, $X_{1,t}$, and $X_{2,t}$ are independent and exponentially $\beta$-mixing, and $U_t$ is iid, $X_t$ is also exponentially $\beta$-mixing.

- Since $Y_i$ depends only on $X_i$, and $\upsilon_i$ are iid, $\{(X_i, Y_i)\}$ is a strictly stationary $\beta$-mixing sequence with $\beta(k) \leq e^{-k/3}$. So condition (a) of Assumption 2.1 is true here.

- Now we will verify condition (b) of Assumption 2.1. Let $\sigma_{X,p}$ denote $\|X\|_p$ for $p > 0$. Then

$$\mathbb{E}\left[\left(\theta^\top X\right)^2\right] = \sum_{i=1}^d \frac{\theta_i^2 \sigma_{X_i,2}^2}{d_i^2} \geq \sigma_0^2 \sum_{i=1}^d \frac{\theta_i^2}{d_i^2}, \tag{24}$$

where $\sigma_0 = \min_i \sigma_{X_i,2}$, $i = 1, 2, \cdots, d$. Since $X_i \sim L(\eta_3)$, for any $\theta_i \in \mathbb{R}$ we have $\|\theta_i X_i\|_{\eta_3-0.5\iota} \leq K_1 |\theta_i| \sigma_{X,\eta_3-0.5\iota}/d_i$. So,

$$\|\theta^\top X\|_{\eta_3-0.5\iota} \leq K_1 \sigma_{X,\eta_3-0.5\iota} \sum_{i=1}^d \frac{|\theta_i|}{d_i} \leq \frac{K_1 \sigma_{X,\eta_3-0.5\iota} \sum_{i=1}^d \frac{|\theta_i|}{d_i}}{\sigma_0 \sqrt{\sum_{i=1}^d \frac{\theta_i^2}{d_i^2}}} \|\theta^\top X\|_2 \tag{25}$$

$$\leq \frac{K_1 \sigma_{X,\eta_3-0.5\iota} \sqrt{d}}{\sigma_0} \|\theta^\top X\|_2. \tag{26}$$

Then from Lemma 4.1 of [Men15] we have that the condition (b) of Assumption 2.1 is true here.

- Since $\upsilon_i$ and $X_i$ are independent, and $\mathbb{E}[\upsilon_i] = 0$, we have $\mathbb{E}\left[\upsilon_i t^\top X_i\right] = 0$. Let $\eta_2 = \min(\eta_3, \eta_4)$. Then using Markov's inequality, for any $t \in \mathcal{H}_R$ and $\forall i$, we have,

$$\mathbb{P}\left(\left|\upsilon_i t^\top X_i - \mathbb{E}\left[\upsilon_i t^\top X_i\right]\right| \geq \tau\right) \leq \frac{\|\upsilon_i\|_{\eta_2}^{\eta_2} \left(\sum_{j=1}^d \|t_j X_{i,j}\|_{\eta_2}\right)^{\eta_2}}{\tau^{\eta_2}} \leq \frac{(R\sigma_\upsilon \sigma_{X,\eta_2} d_H)^{\eta_2}}{\tau^{\eta_2}},$$

where $d_H = \sum_{i=1}^d d_j^{-1}$, and for all $i, j$, $\|\upsilon_i\|_{\eta_2} = \sigma_\upsilon$, and $\|X_{i,j}\|_{\eta_2} = \sigma_{X,\eta_2}/d_j$. This implies that condition (c)-(ii) of Assumption 2.1 is true in this setting.

**Proposition 4.1.** *Consider the learning problem described in Section 4.2. Then with probability at least* (14)*, we have,*

$$\|\hat{f} - f^*\|_{L_2} \leq \max\left\{N^{-\frac{1}{4}\left(1-\frac{1}{\eta_2}\right)+\iota}, \frac{C_9 R}{\tau Q_{\mathcal{H}_R}(2\tau)^{3/2}} d^{1/(\eta_2-0.5\iota)+\iota/8} N^{-1/2+\iota}\right\}. \tag{27}$$

The proof of Proposition 4.1 could be found in the Appendix D.

**Remark 3.** *Note that the heaviness of the tail dominates the exponential $\beta$-mixing rate $\eta_1$ in determining the rates of convergence. Observe that as $\eta_2 \to \infty$ all the moments exist, and $N^{-(1-1/\eta_2)/4+\iota} \to N^{-1/4+\iota}$. This rate is the same as the one that we obtain under condition (c)-(i), although with weaker polynomial probability. Furthermore, the dimension dependency is polynomial here. On a related note, [ZZ18] analyzes $\ell_1$-regression with a truncated loss in the iid setting with the assumption $\eta_2 > 2$ and gets $\sqrt{d/N}$ rate with exponential probability. We point out that in the iid setting Medians-of-mean method achieves the optimal rate with exponential probability under the stronger assumption that $\log d$ moments exist [LM19].*

### 4.3 Example 3: $\beta$-mixing sub-Gaussian data and Pareto noise with Huber Loss

In this example, we consider Huber loss which satisfies Assumption 3.2:

$$\ell_{T_h}(t) = \begin{cases} t^2/2 & \text{if } |t| \leq T_h \\ T_h |t| - T_h^2/2 & \text{if } |t| \geq T_h. \end{cases}$$

We now formally establish the benefits of using Huber loss when the noise $\xi$ has a polynomial tail but $(f - f^*)(X)$ has a sub-Weibull tail. Let $\{\delta_i\}_{i \in \mathbb{Z}^+}$ be an iid sequence of $d$-dimensional standard

Gaussian random vectors $\delta_i \sim N(0, I_d)$. To compare the performance of ERM under Huber loss to that of squared loss, for simplicity, we allow $X$ to be Gaussian; but a similar result will hold for sub-Weibull $\delta_i$. Assume that the input vectors are generated according to (19) where $A$ is a $d \times d$ matrix with spectral radius less than 1. For simplicity, let $A = \sigma_0^2 I_d$ where $\sigma_0^2 < 1$. [WZLL20] showed that this time series is stable, strict sense stationary, with $X_i \sim N(0, 1/(1 - \sigma_0^4)I_d)$. Let $\{Y_i\}_{i \in \mathbb{Z}^+}$, be the response sequence given by $Y_i = \theta^{*\top} X_i + \xi_i$, where $\theta^* \in B_1^d(R)$, $\ell_1$-norm ball in $\mathbb{R}^d$. Let $\{\xi_i\}_{i=1}^n$ be independent of $X_i$ for all $i$, and be an $\mathtt{iid}$ sequence of 0 mean random variables with heavy tails such that for all $t > 0$, $\mathbb{P}(|\xi_i| \geq t) \leq 1/(1 + t^{\eta_4})$, $\eta_4 > 2 + 2\iota$ and $\mathrm{VAR}(\xi) \leq \sigma_\xi$. Set $T_h = 3\sigma_\xi$. We verify the required assumptions for this example in the following section.

### 4.3.1 Verification of Assumptions for Example 4.3

Condition (a) of Assumption 2.1 and condition (b) of Assumption 2.2 are true here by the same argument as in Example 4.1. This also implies that Assumption 3.1 is true in this case for $p = 8$. Since we consider Huber loss, a lipschitz continuous loss, we have $l'(\xi) \leq \min(|\xi/2|, T_h)$. Set $T_h = c_{11}(\kappa_0 + 1/\sqrt{\epsilon})(\sigma_\xi + 2R)$. As an immediate consequence of [VGNA20, Proposition 2.3], we have that condition (c)-(i) in Assumption 2.2 is valid here with $\eta_2 = 2$. Note that if a sequence is exponentially $\beta$-mixing, i.e., satisfies condition (a) of Assumption 2.1 with coefficient $\eta_1 > 0$, then the same condition is true for all $\eta < \eta_1$. So choosing $\eta_1' < \eta_2/(\eta_2 - 1)$ we get Condition (d) of Assumption 2.1 is true.

**Proposition 4.2.** *Consider the learning problem described in Section 4.3. Then, for some constant $c_{12} > 0$, with probability at least*

$$1 - c_9 \epsilon^{1 - \frac{1}{\eta_1}} N^{\eta_1/(1+\eta_1)} e^{-c_{10} \epsilon^{1 + \frac{1}{\eta_1}} N^{\eta_1/(1+\eta_1)}} - \widetilde{C}_2 N \exp\left(-(N^{2\iota}\tau_0)^{\eta}/M_1\right),$$

*we have*

$$\|\hat{f} - f^*\|_{L_2} \leq \max\left\{N^{-\frac{1}{2}+\iota}, c_{12} R \sqrt{\log(ed/N)} N^{-\frac{1}{2}}\right\}.$$

The proof of Proposition 4.2 could be found in the Appendix E.

**Remark 4.** *In this setting, using squared loss would mean that condition (c)-(ii) of Assumption 2.1 is true. So, by part 2 of Theorem 3.1, the obtained rate would be of order $N^{-1/8}$ with polynomial probability given by (14), which is significantly worse than that of Proposition 4.2 with Huber loss.*

## 5 Proof Sketch of Theorem 3.1 and 3.2

Recall the decomposition (1). The first and the last terms in the RHS of (1) are handled respectively by condition (b) and (c) of Assumption 2.1. The idea is to show that if for some $f \in \mathcal{F}$, $\|f - f^*\|_{L_2}$ is large, then with high probability $T_1 := N^{-1}\sum_{i=1}^N (f - f^*)^2(X_i) \geq \underline{B}$ (Lemma B.1) and $T_2 := 2N^{-1}\sum_{i=1}^N \xi_i(f - f^*)(X_i) \geq \bar{B}$ (see (67)) where $\underline{B} + \bar{B} > 0$. But since $\hat{f}$ minimizes $P_N L_f$ and $f^* \in \mathcal{F}$, $P_N L_f \leq 0$. So with high probability $\|\hat{f} - f^*\|_{L_2}$ is small. In contrast to [Men15], we face two major challenges: 1- For the lower bound on $T_1$, the symmetrization argument used in the $\mathtt{iid}$ case (e.g. [Men15, Men18]) is not applicable under our dependency structure. 2- For the term $T_2$, [Men15, Men18] use the complexity measure $\alpha_N^*(\gamma, \delta)$ (see (108)) in the $\mathtt{iid}$ case to control the noise-input interactions, which is not possible to do in our setting; our analysis to control $T_2$ is different, through which we can show how different tail conditions on the data and noise affect the high-probability statements on the learning rate. The proof of Theorem 3.2 for the locally strongly-convex loss functions, follows a similar strategy although the technical details are more involved.

## 6 Conclusion

In this work, we analyzed the performance of ERM with squared and convex loss functions, when the DGP is both dependent (specifically, exponentially $\beta$-mixing) and heavy-tailed. We derived explicit rates using a combination of small-ball method and concentration inequalities. We demonstrated the applicability of our results on a high-dimensional linear regression problem, and showed that our assumptions are easily verified for a certain classes of sub-Weibull and Pareto DGP. Our results clearly show the benefits of using Huber loss over the squared error loss for ERM with heavy-tailed data in our setting. For future work, we plan to study median-of-means based techniques and examine establishing similar rates of convergence for dependent heavy-tailed data.

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
