# A Proof of Lemma 2.2

**Lemma A.1** (Concentration for heavy-tailed $\beta$-mixing sum, Lemma 2.2). *Let $\{W_j\}_{j\geq 1}$ be a sequence of zero-mean real valued random variables satisfying conditions (a) and (c)-(ii) of Assumption 2.1, for some $\eta_2 > 2$. Then for any positive integer $N$, $0 \leq d_1 \leq 1$, and $d_2 \geq 0$, and for any $t > 1$, we have,*

$$\mathbb{P}\left(\sup_{j\leq N}\left|\sum_{i=1}^{j} W_i\right| \geq t\right) \leq \frac{2^{\eta_2+3}}{(d_2 \log t)^{\frac{1-\eta_2}{\eta_1}}}\frac{N}{t^{(1+d_1(\eta_2-1))}} + 8\frac{N}{t^{(1+c'd_2)}} + 2e^{-\frac{t^{2-2d_1}(d_2 \log t)^{1/\eta_1}}{9N}},$$

*where $c' > 0$ is a constant.*

*Proof.* [Proof of Lemma 2.2] Let $W_{i,M}$ denote the truncated random variable $W_i$ such that $W_{i,M} = \max(\min(W_i, M), -M)$. Then define $\Sigma_N := \sum_{i=1}^{N} W_i$. Consider the partition of the samples into blocks of length $A$, $I_i = \{1 + (i-1)A, \cdots, iA\}$ for $i = 1, 2, \cdots, 2\mu_1$ where $\mu_1 = [N/(2A)]$. Also let $I_{2\mu_1+1} = \{2\mu_1 A + 1, \cdots, N\}$. Define for a finite set $I$ of positive integers, define $\Sigma_{N,M}(I) = \sum_{i\in I} W_{i,M}$. Then we can write, for $j \leq N$

$$\Sigma_j = \sum_{i=1}^{j} W_i \tag{28}$$

$$= \sum_{i=1}^{j}(W_i - W_{i,M}) + \sum_{i=1}^{j} W_{i,M} \tag{29}$$

$$= \sum_{i=1}^{j}(W_i - W_{i,M}) + \sum_{i\leq[j/A]}\Sigma_{N,M}(I_{2i}) + \sum_{i\leq[j/A]}\Sigma_{N,M}(I_{2i-1}) + \sum_{i=A[j/A]+1}^{j} W_{i,M}. \tag{30}$$

Then we have,

$$|\Sigma_j| \leq \sum_{i=1}^{j}|W_i - W_{i,M}| + \left|\sum_{i\leq[j/A]}\Sigma_{N,M}(I_{2i})\right| + \left|\sum_{i\leq[j/A]}\Sigma_{N,M}(I_{2i-1})\right| + 2AM \tag{31}$$

$$\sup_{j\leq N}|\Sigma_j| \leq \sum_{i=1}^{N}|W_i - W_{i,M}| + \sup_{j\leq N}\left|\sum_{i\leq[j/A]}\Sigma_{N,M}(I_{2i})\right| + \sup_{j\leq N}\left|\sum_{i\leq[j/A]}\Sigma_{N,M}(I_{2i-1})\right| + 2AM. \tag{32}$$

Now we will establish concentration for each of the terms in the above expression. Using Markov's inequality,

$$\mathbb{P}\left(\sum_{i=1}^{N}|W_i - W_{i,M}| \geq t\right) \leq \frac{1}{t}\sum_{i=1}^{N}\mathbb{E}\left[|W_i - W_{i,M}|\right] \leq \frac{2}{t}\sum_{i=1}^{N}\int_{M}^{\infty}\mathbb{P}(|W_i| \geq x)dx$$

$$\leq \frac{2N}{t}\int_{M}^{\infty} x^{-\eta_2}dx = \frac{2N}{t(\eta_2 - 1)}M^{1-\eta_2}. \tag{33}$$

Using Lemma 5 of [DP04], we get independent random variables $\{\Sigma_{N,M}^*(I_{2i})\}_{1\leq i\leq \mu_1}$, where $\Sigma_{N,M}^*(I_{2i})$ has the same distribution as $\Sigma_{N,M}(I_{2i})$, such that,

$$\mathbb{E}\left[\left|\Sigma_{N,M}(I_{2i}) - \Sigma_{N,M}^*(I_{2i})\right|\right] \leq A\tau(A). \tag{34}$$

Then, using Markov's inequality we have,

$$\mathbb{P}\left(\sup_{j\leq N}\left|\sum_{i\leq[j/A]}\Sigma_{N,M}(I_{2i})\right| \geq t\right)$$

$$\leq \mathbb{P}\left(\sup_{j\leq N}\left|\sum_{i\leq[j/A]}(\Sigma_{N,M}(I_{2i}) - \Sigma_{N,M}^*(I_{2i}))\right| + \sup_{j\leq N}\left|\sum_{i\leq[j/A]}\Sigma_{N,M}^*(I_{2i})\right| \geq t\right)$$

$$\leq \mathbb{P}\left(\sup_{j\leq N}\left|\sum_{i\leq [j/A]}(\Sigma_{N,M}(I_{2i}) - \Sigma_{N,M}^*(I_{2i}))\right| \geq \frac{t}{2}\right) + \mathbb{P}\left(\sup_{j\leq N}\left|\sum_{i\leq [j/A]}\Sigma_{N,M}^*(I_{2i})\right| \geq \frac{t}{2}\right)$$

$$\leq \frac{2\mathbb{E}\left[\sup_{j\leq N}\left|\sum_{i\leq [j/A]}(\Sigma_{N,M}(I_{2i}) - \Sigma_{N,M}^*(I_{2i}))\right|\right]}{t} + \mathbb{P}\left(\sup_{j\leq N}\left|\sum_{i\leq [j/A]}\Sigma_{N,M}^*(I_{2i})\right| \geq \frac{t}{2}\right)$$

$$\leq \frac{2\mathbb{E}\left[\sup_{j\leq N}\sum_{i\leq \mu_1}|\Sigma_{N,M}(I_{2i}) - \Sigma_{N,M}^*(I_{2i})|\right]}{t} + \mathbb{P}\left(\sup_{j\leq N}\left|\sum_{i\leq [j/A]}\Sigma_{N,M}^*(I_{2i})\right| \geq \frac{t}{2}\right)$$

$$\leq \frac{2A\mu_1\tau(A)}{t} + \mathbb{P}\left(\sup_{j\leq N}\left|\sum_{i\leq [j/A]}\Sigma_{N,M}^*(I_{2i})\right| \geq \frac{t}{2}\right).$$

The same results holds for $\{\Sigma_{N,M}^*(I_{2i-1})\}_{i=1,2,\cdots,k}$. So for any $t \geq 2AM$, we have,

$$\mathbb{P}\left(\sup_{j\leq N}|\Sigma_j| \geq 6t\right) \leq \frac{2N}{t(\eta_2 - 1)}M^{1-\eta_2} + \frac{4A\mu_1\tau(A)}{t} + \mathbb{P}\left(\sup_{j\leq N}\left|\sum_{i\leq [j/A]}\Sigma_{N,M}^*(I_{2i})\right| \geq t\right)$$

$$+ \mathbb{P}\left(\sup_{j\leq N}\left|\sum_{i\leq [j/A]}\Sigma_{N,M}^*(I_{2i-1})\right| \geq t\right). \tag{35}$$

Now, for $\lambda > 0$

$$\mathbb{P}\left(\sup_{j\leq N}\left|\sum_{i\leq [j/A]}\Sigma_{N,M}^*(I_{2i})\right| \geq t\right) \leq e^{-\lambda t}\mathbb{E}\left[\exp\left(\lambda \sup_{j\leq N}\left|\sum_{i\leq [j/A]}\Sigma_{N,M}^*(I_{2i})\right|\right)\right]$$

$$\leq e^{-\lambda t}\mathbb{E}\left[\exp\left(\lambda \sum_{i\leq \mu_1}|\Sigma_{N,M}^*(I_{2i})|\right)\right]$$

$$\leq e^{-\lambda t}\Pi_{i=1}^{\mu_1}\mathbb{E}\left[\exp\left(\lambda |\Sigma_{N,M}(I_{2i})|\right)\right].$$

We have $|\Sigma_{N,M}(I_{2i})| \leq AM$. So $|\Sigma_{N,M}(I_{2i})|$ is a sub-gaussian random variable and consequently,

$$\mathbb{P}\left(\sup_{j\leq N}\left|\sum_{i\leq [j/A]}\Sigma_{N,M}^*(I_{2i})\right| \geq t\right) \leq e^{-\lambda t}e^{\frac{\lambda^2\mu_1 A^2 M^2}{2}}.$$

Optimizing over $\lambda > 0$ we have,

$$\mathbb{P}\left(\sup_{j\leq N}\left|\sum_{i\leq [j/A]}\Sigma_{N,M}^*(I_{2i})\right| \geq t\right) \leq e^{-\frac{t^2}{2\mu_1 A^2 M^2}}. \tag{36}$$

Similarly, we also obtain

$$\mathbb{P}\left(\sup_{j\leq N}\left|\sum_{i\leq [j/A]}\Sigma_{N,M}^*(I_{2i-1})\right| \geq t\right) \leq e^{-\frac{t^2}{2\mu_1 A^2 M^2}}. \tag{37}$$

From (35), (36), and (37) we have,

$$\mathbb{P}\left(\sup_{j\leq N}|\Sigma_j| \geq 6t\right) \leq \frac{2N}{t(\eta_2 - 1)}M^{1-\eta_2} + \frac{4A\mu_1\tau(A)}{t} + 2e^{-\frac{t^2}{2\mu_1 A^2 M^2}}.$$

Condition (a) in Assumption 2.1 implies that the process $\{Z_i\}_{i=-\infty}^{\infty}$ is exponentially $\tau$-mixing [CG14], i.e., for a constant $c' > 0$, $\tau(k) \leq e^{-c'k^{\eta_1}}$. Then we have

$$\mathbb{P}\left(\sup_{j\leq N}|\Sigma_j| \geq t\right) \leq \frac{12N}{t(\eta_2 - 1)}M^{1-\eta_2} + \frac{24A\mu_1\exp(-c'A^{\eta_1})}{t} + 2e^{-\frac{t^2}{72\mu_1 A^2 M^2}}.$$

As $2A\mu_1 \leq N \leq 3A\mu_1$ and $\eta_2 > 2$,

$$\mathbb{P}\left(\sup_{j \leq N} |\Sigma_j| \geq t\right) \leq \frac{12NM^{1-\eta_2}}{t} + \frac{8N\exp(-c'A^{\eta_1})}{t} + 2e^{-\frac{t^2}{36NAM^2}}.$$

Now choosing

$$M = \frac{t^{d_1}}{2(d_2 \log t)^{\frac{1}{\eta_1}}}, \quad A = (d_2 \log t)^{\frac{1}{\eta_1}}, \qquad 0 \leq d_1 \leq 1, \quad d_2 \geq 0, \tag{38}$$

we have, $2AM \leq t$, and

$$\mathbb{P}\left(\sup_{j \leq N} |\Sigma_j| \geq t\right) \leq \frac{2^{\eta_2+3}}{(d_2 \log t)^{\frac{1-\eta_2}{\eta_1}}} Nt^{-(1+d_1(\eta_2-1))} + 8Nt^{-(1+d_2c')} + 2e^{-\frac{t^{2-2d_1}(d_2 \log t)^{1/\eta_1}}{9N}}.$$

$\blacksquare$

# B    Proofs of Section 3

## B.1    Proofs for squared error loss

Similar to the decomposition (1), for squared loss we have

$$P_N L_f = \frac{1}{N} \sum_{i=1}^{N} (f - f^*)^2(X_i) + \frac{2}{N} \sum_{i=1}^{N} \xi_i(f - f^*)(X_i),$$

Since $\mathcal{F}$ is convex, we also have

$$\mathbb{E}\left[\xi(f - f^*)(X)\right] \geq 0.$$

Then,

$$P_N L_f \geq \frac{1}{N} \sum_{i=1}^{N} (f - f^*)^2(X_i) + \frac{2}{N} \sum_{i=1}^{N} (\xi_i(f - f^*)(X_i) - \mathbb{E}\left[\xi_i(f - f^*)(X_i)\right]). \tag{39}$$

Now our goal is to establish a lower bound (Lemma B.1) on the first term of the RHS of (39), and a two-sided bound ((67) and (69)) on the second term when $\|f - f^*\|_{L_2}$ is large. Combining these bounds we will show that if $\|f - f^*\|_{L_2}$ is large then $P_N L_f > 0$ which implies $f$ cannot be a minimizer of empirical risk because for the minimizer $\hat{f}$ we have $P_N L_{\hat{f}} \leq 0$.

**Lemma B.1.** *Let Condition (a) and (b) of Assumption 2.1 be true. Given $f^* \in \mathcal{F}$, set $\mathcal{H} = \mathcal{F} - f^*$. Then, for every $\rho > \omega_\mu(\mathcal{H}, \tau Q_\mathcal{H}(2\tau)/16)$, with probability at least $\mathscr{P}_1$, if $\|f - f^*\|_{L_2} \geq \rho$, we have,*

$$|\{i : |(f - f^*)(X_i)| \geq \tau \|f - f^*\|_{L_2}\}| \geq \frac{NQ_\mathcal{H}(2\tau)}{4}. \tag{40}$$

The proof of Lemma B.1 follows easily by combining the results of Lemma B.2, and Corollary B.1 which we state next.

**Lemma B.2.** *Let $S(L_2)$ be the $L_2(\pi)$ unit sphere and let $\mathcal{H} \subset S(L_2)$. Consider the partition in (6). Under conditions (a) and (b) of Assumption 2.1, by setting $\mu = \frac{NQ_\mathcal{H}(2\tau)c^{\frac{1}{\eta_1}}}{4\mathscr{G}(N)^{\frac{1}{\eta_1}}}$ for some $\mathscr{G}(N) \leq \frac{cQ_\mathcal{H}(2\tau)^{\eta_1}N^{\eta_1}}{4^{\eta_1}}$, if,*

$$\mathfrak{R}_\mu(\mathcal{H}) \leq \frac{\tau Q_\mathcal{H}(2\tau)N}{16\mu}, \tag{41}$$

*then with probability at least $1 - 2\exp\left(-\frac{NQ_\mathcal{H}(2\tau)^3}{2(4-Q_\mathcal{H}(2\tau))^2}\left(\frac{c}{\mathscr{G}(N)}\right)^{\frac{1}{\eta_1}}\right) - \frac{NQ_\mathcal{H}(2\tau)c^{\frac{1}{\eta_1}}}{4\mathscr{G}(N)^{\frac{1}{\eta_1}}}\exp(-\mathscr{G}(N))$, we have*

$$\inf_{h \in \mathcal{H}} |\{i : |h(X_i)| \geq \tau\}| \geq \frac{NQ_\mathcal{H}(2\tau)}{4}. \tag{42}$$

**Remark 5.** *We have the following illustrative instantiations of Lemma B.2:*

1. *If one sets $\mathscr{G}(N) = k \log N \leq \frac{cQ_{\mathcal{H}}(2\tau)^{\eta_1} N^{\eta_1}}{4^{\eta_1}}$, then the statement of Lemma B.2 holds as long as,*

$$\mathfrak{R}_\mu(\mathcal{H}) \leq \frac{\tau(k \log N)^{\frac{1}{\eta_1}}}{4c^{\frac{1}{\eta_1}}}, \tag{43}$$

*with probability at least*

$$1 - 2\exp\left(-\frac{NQ_{\mathcal{H}}(2\tau)^3}{2(4 - Q_{\mathcal{H}}(2\tau))^2}\left(\frac{c}{k \log N}\right)^{\frac{1}{\eta_1}}\right) - \frac{N^{1-k}Q_{\mathcal{H}}(2\tau)c^{\frac{1}{\eta_1}}}{4(k \log N)^{\frac{1}{\eta_1}}}.$$

2. *If one sets $\mathscr{G}(N) = N^r \leq \frac{cQ_{\mathcal{H}}(2\tau)^{\eta_1} N^{\eta_1}}{4^{\eta_1}}$, for some $0 < r < \eta_1$, then the statement of Lemma B.2 holds as long as,*

$$\mathfrak{R}_\mu(\mathcal{H}) \leq \frac{\tau(N)^{\frac{r}{\eta_1}}}{4c^{\frac{1}{\eta_1}}}, \tag{44}$$

*with probability at least*

$$1 - 2\exp\left(-\frac{NQ_{\mathcal{H}}(2\tau)^3}{2(4 - Q_{\mathcal{H}}(2\tau))^2}\left(\frac{c}{N^r}\right)^{\frac{1}{\eta_1}}\right) - \frac{NQ_{\mathcal{H}}(2\tau)c^{\frac{1}{\eta_1}}}{4(N)^{\frac{r}{\eta_1}}}\exp(-N^r).$$

*Proof.* [Proof of Lemma B.2] Let $\psi_u : \mathbb{R}_+ \to [0, 1]$ be the function

$$\psi_u(t) = \begin{cases} 1 & t \geq 2u, \\ \frac{t}{u} - 1 & u \leq t \leq 2u \\ 0 & t < u \end{cases}$$

Similar to (6), let us define sequences of i.i.d blocks $\{\tilde{Z}_i^{(a)}\}_{i=1}^{\mu}$, and $\{\tilde{Z}_i^{(b)}\}_{i=1}^{\mu}$ where the samples within each block are assumed to be drawn from the same $\beta$-mixing distribution of $\{Z_i^{(a)}\}_{i=1}^{\mu}$, and $\{Z_i^{(b)}\}_{i=1}^{\mu}$. Let $\tilde{S}_a = \left(\tilde{Z}_1^{(a)}, \cdots, \tilde{Z}_\mu^{(a)}\right)$, and $\tilde{S}_b = \left(\tilde{Z}_1^{(b)}, \cdots, \tilde{Z}_\mu^{(b)}\right)$. Now let us concentrate on the term $|P_N\psi_u(|h|) - P\psi_u(|h|)|$.

$$|P_N\psi_u(|h|) - P\psi_u(|h|)|$$

$$\leq \left|\frac{1}{N}\sum_{i=1}^{N}\psi_u(h(|X_i|)) - P\psi_u(|h|)\right|$$

$$\leq \left|\frac{1}{N}\sum_{i=1}^{\mu}\sum_{j=1}^{a}\psi_u(|h(X_{(i-1)(a+b)+j}|) + \frac{1}{N}\sum_{i=1}^{\mu}\sum_{j=1}^{b}\psi_u(h(|X_{(i-1)(a+b)+a+j}|)) - P\psi_u(|h|)\right|$$

$$\leq \left|\frac{1}{N}\sum_{i=1}^{\mu}\sum_{j=1}^{a}\left(\psi_u(h(|X_{(i-1)(a+b)+j}|)) - P\psi_u(|h|)\right)\right| + \frac{b\mu}{N}. \tag{45}$$

Using (45) and Corollary 2.7 of [Yu94], for some $a, b, \mu$ to be chosen later such that $(a+b)\mu = N$ we have,

$$\mathbb{P}\left(|P_N\psi_u(|h|) - P\psi_u(|h|)| \geq t + \frac{b\mu}{N}\right) \tag{46}$$

$$\leq \mathbb{P}\left(\left|\frac{1}{N}\sum_{i=1}^{\mu}\sum_{j=1}^{a}\left(\psi_u(h(|X_{(i-1)(a+b)+j}|)) - P\psi_u(|h|)\right)\right| + \frac{b\mu}{N} \geq t + \frac{b\mu}{N}\right) \tag{47}$$

$$= \mathbb{E}\left[\mathbb{1}\left(\left|\frac{1}{N}\sum_{i=1}^{\mu}\sum_{j=1}^{a}\left(\psi_u(h(|X_{(i-1)(a+b)+j}|)) - P\psi_u(|h|)\right)\right| \geq t\right)\right] \tag{48}$$

$$\leq \mathbb{E}\left[\mathbb{1}\left(\left|\frac{1}{N}\sum_{i=1}^{\mu}\sum_{j=1}^{a}\left(\psi_u(h(|\tilde{X}_{(i-1)(a+b)+j}|)) - P\psi_u(|h|)\right)\right| \geq t\right)\right] + (\mu - 1)\beta(b) \tag{49}$$

$$= \mathbb{P}\left(\left|\frac{1}{N}\sum_{i=1}^{\mu}\sum_{j=1}^{a}\left(\psi_u(h(|\tilde{X}_{(i-1)(a+b)+j}|)) - P\psi_u(|h|)\right)\right| \geq t\right) + (\mu - 1)\beta(b) \tag{50}$$

$$= \mathbb{P}\left(\left|\frac{1}{\mu}\sum_{i=1}^{\mu}\tilde{\psi}(\tilde{Z}_i^{(a)})\right| \geq \frac{Nt}{\mu}\right) + (\mu - 1)\beta(b), \tag{51}$$

where

$$\tilde{\psi}(\tilde{Z}_i^{(a)}) = \sum_{j=1}^{a}\left(\psi_u(h(|\tilde{X}_{(i-1)(a+b)+j}|)) - P\psi_u(|h|)\right),$$

and $\mathbb{1}(\cdot)$ is the indicator function. Observe that the function

$$W(\tilde{Z}_1^{(a)}, \tilde{Z}_2^{(a)}, \cdots, \tilde{Z}_\mu^{(a)}) = \mu^{-1}\sum_{i=1}^{\mu}\tilde{\psi}(\tilde{Z}_i^{(a)})$$

has bounded difference with coefficient $2a/\mu$. Then using Mcdiarmid's bounded-difference inequality on $W(\tilde{Z}_1^{(a)}, \tilde{Z}_2^{(a)}, \cdots, \tilde{Z}_\mu^{(a)})$ we get,

$$\mathbb{P}\left(\left|\frac{1}{\mu}\sum_{i=1}^{\mu}\tilde{\psi}(\tilde{Z}_i^{(a)})\right| \geq \frac{Nt}{\mu}\right) \leq 2\exp\left(-\frac{N^2t^2}{2a^2\mu}\right). \tag{52}$$

Combining (51), and (52), we get

$$\mathbb{P}\left(|P_N\psi_u(|h|) - P\psi_u(|h|)| \geq t + \frac{b\mu}{N}\right) \leq 2\exp\left(-\frac{N^2t^2}{2a^2\mu}\right) + (\mu - 1)\beta(b),$$

which implies

$$\mathbb{P}\left(|P_N\psi_u(|h|) - P\psi_u(|h|)| \geq \frac{4\mu}{Nu}\mathbb{R}_\mu(\mathcal{H}) + \frac{b\mu}{N} + \frac{t}{\sqrt{N}}\right)$$

$$\leq 2\exp\left(-\frac{N^2\left(\frac{4\mu}{Nu}\mathbb{R}_\mu(\mathcal{H}) + \frac{t}{\sqrt{N}}\right)^2}{2a^2\mu}\right) + (\mu - 1)\beta(b).$$

Also note that, for any $t$, we have $|P_N\psi_u(|h|) - P\psi_u(|h|)| \geq t$ which implies that we also have $\sup_{h\in\mathcal{H}}|P_N\psi_u(|h|) - P\psi_u(|h|)| \geq t$. Hence,

$$\mathbb{P}\left(\sup_{h\in\mathcal{H}}|P_N\psi_u(|h|) - P\psi_u(|h|)| \geq \frac{4\mu}{Nu}\mathbb{R}_\mu(\mathcal{H}) + \frac{b\mu}{N} + \frac{t}{\sqrt{N}}\right)$$

$$\leq 2\exp\left(-\frac{N^2\left(\frac{4\mu}{Nu}\mathbb{R}_\mu(\mathcal{H}) + \frac{t}{\sqrt{N}}\right)^2}{2a^2\mu}\right) + (\mu - 1)\beta(b). \tag{53}$$

In other words, with probability at least $1 - 2\exp\left(-\frac{N^2\left(\frac{4\mu}{Nu}\mathbb{R}_\mu(\mathcal{H}) + \frac{t}{\sqrt{N}}\right)^2}{2a^2\mu}\right) - (\mu - 1)\beta(b)$, we have

$$\sup_{h\in\mathcal{H}}|P_N\psi_u(|h|) - P\psi_u(|h|)| \leq \frac{4\mu}{Nu}\mathbb{R}_\mu(\mathcal{H}) + \frac{b\mu}{N} + \frac{t}{\sqrt{N}}. \tag{54}$$

Hence, we have

$$P_N\mathbb{1}_{\{|h|\geq u\}} \geq \inf_{h\in\mathcal{H}}\mathbb{P}(|h| \geq 2u) - \sup_{h\in\mathcal{H}}|P_N\psi_u(|h|) - P\psi_u(|h|)|. \tag{55}$$

So, combining (53), and (55), with probability at least $1 - 2\exp\left(-\frac{N^2\left(\frac{4\mu}{Nu}\mathbb{R}_\mu(\mathcal{H}) + \frac{t}{\sqrt{N}}\right)^2}{2a^2\mu}\right) - (\mu - 1)\beta(b)$ we have

$$P_N \mathbf{1}_{\{|h| \geq u\}} \geq \inf_{h \in \mathcal{H}} \mathbb{P}(|h| \geq 2u) - \frac{4\mu}{Nu}\mathbb{R}_\mu(\mathcal{H}) - \frac{b\mu}{N} - \frac{t}{\sqrt{N}}.$$

Now, setting

$$u = \tau \quad t = \frac{\sqrt{N}Q_\mathcal{H}}{4} \quad a = \frac{(4 - Q_\mathcal{H}(2\tau))(\mathscr{G}(N))^{\frac{1}{\eta_1}}}{Q_\mathcal{H}(2\tau)c^{\frac{1}{\eta_1}}} \quad b = \left(\frac{\mathscr{G}(N)}{c}\right)^{\frac{1}{\eta_1}} \quad \mu = \frac{NQ_\mathcal{H}(2\tau)c^{\frac{1}{\eta_1}}}{4\mathscr{G}(N)^{\frac{1}{\eta_1}}}, \tag{56}$$

and using the condition $\mathscr{G}(N) > c$, we get, with probability at least

$$1 - 2\exp\left(-\frac{NQ_\mathcal{H}(2\tau)^3}{2(4 - Q_\mathcal{H}(2\tau))^2}\left(\frac{c}{\mathscr{G}(N)}\right)^{\frac{1}{\eta_1}}\right) - \frac{NQ_\mathcal{H}(2\tau)c^{\frac{1}{\eta_1}}}{4\mathscr{G}(N)^{\frac{1}{\eta_1}}}\exp(-\mathscr{G}(N)),$$

we have

$$P_N \mathbf{1}_{\{|h| \geq u\}} \geq \frac{Q_\mathcal{H}(2\tau)}{4}.$$

∎

**Corollary B.1.** *Let Condition (a) and (b) of Assumption 2.1 be true. Let $\mathcal{H}$ be star-shaped around 0 and assume that there is some $\tau > 0$ for which $Q_\mathcal{H}(2\tau) > 0$. Then for every $\rho > \omega_\mu(\mathcal{H}, \tau Q_\mathcal{H}(2\tau)/16)$, with probability at least*

$$\mathscr{P}_1 := 1 - 2\exp\left(-\frac{NQ_\mathcal{H}(2\tau)^3}{2(4 - Q_\mathcal{H}(2\tau))^2}\left(\frac{c}{\mathscr{G}(N)}\right)^{\frac{1}{\eta_1}}\right) - \frac{NQ_\mathcal{H}(2\tau)c^{\frac{1}{\eta_1}}}{4\mathscr{G}(N)^{\frac{1}{\eta_1}}}\exp(-\mathscr{G}(N)),$$

*for every $h \in \mathcal{H}$ that satisfies $\|h\|_{L_2} \geq \rho$,*

$$|\{i : |h(X_i)| \geq \tau\|h\|_{L_2}\}| \geq N\frac{Q_\mathcal{H}(2\tau)}{4}. \tag{57}$$

*Proof.* [Proof of Corollary B.1] Let $\rho > \omega_\mu(\mathcal{H}, \tau Q_\mathcal{H}(2\tau)/16)$ and as $\mathcal{H}$ is star-shaped around 0,

$$\mathfrak{R}_\mu(\mathcal{H} \cap \rho\mathcal{D}) \leq \frac{\tau Q_\mathcal{H}(2\tau)}{16}\rho. \tag{58}$$

Consider the set,

$$V = \{h/\rho : h \in \mathcal{H} \cap \rho S(L_2)\} \subset S(L_2). \tag{59}$$

Clearly, $Q_V(2\tau) \geq Q_\mathcal{H}(2\tau)$ and

$$\mathfrak{R}_\mu(V) = \mathbb{E}\left[\sup_{h \in \mathcal{H} \cap \rho S(L_2)}\left|\frac{1}{\mu}\sum_{i=1}^\mu \epsilon_i \frac{h(\tilde{X}_i)}{\rho}\right|\right] \leq \frac{\tau Q_\mathcal{H}(2\tau)}{16} \leq \frac{\tau Q_V(2\tau)}{16}. \tag{60}$$

Using Lemma B.2 on the set $V$, we get with probability at least $\mathscr{P}_1$, for every $v \in V$

$$\inf_{h \in \mathcal{H}} |\{i : |v(X_i) \geq \tau|\}| \geq \frac{NQ_V(2\tau)}{4} \geq \frac{NQ_\mathcal{H}(2\tau)}{4}.$$

Now for any $h$ with $\|h\|_{L_2} \geq \rho$, since $\mathcal{H}$ is star-shaped around 0, we have $(\rho/\|h\|_{L_2})h \in \mathcal{H} \cap \rho S(L_2)$ which implies, $h/\|h\|_{L_2} \in V$. So we have (57). ∎

**Theorem B.1** (Restatement of Theorem 3.1). *Consider the LS-ERM procedure. For $\tau_0 < \tau^2 Q_\mathcal{H}(2\tau)/8$, setting $\mu = \frac{N^r Q_\mathcal{H}(2\tau)c^{\frac{1}{\eta_1}}}{4}$, for some constants $c, c' > 0$, and $0 < r < 1$, we have, for any $N \geq 4$,*

1. *under condition (a), (b), (c)-(i), and (d) of Assumption 2.1, for $0 < \iota < \frac{1}{4}$,*

$$\left(\int (\hat{f} - f^*)^2 d\pi\right)^{\frac{1}{2}} = \|\hat{f} - f^*\|_{L_2} \leq \max\left\{N^{-\frac{1}{4}+\iota}, \omega_\mu(\mathcal{F} - \mathcal{F}, \tau Q_{\mathcal{F}-\mathcal{F}}(2\tau)/16)\right\} \tag{61}$$

*with probability at least*

$$1 - 2\exp\left(-\frac{N^r Q_{\mathcal{H}}(2\tau)^3 c^{\frac{1}{\eta_1}}}{2(4 - Q_{\mathcal{H}}(2\tau))^2}\right) - \frac{N^r Q_{\mathcal{H}}(2\tau) c^{\frac{1}{\eta_1}}}{4}\exp(-N^{(1-r)\eta_1}) - N\exp\left(-\frac{(N^{\frac{1}{2}+2\iota}\tau_0)^\eta}{C_1}\right)$$
$$-\exp\left(-\frac{N^{1+4\iota}\tau_0^2}{C_2(1 + NV)}\right) - \exp\left(-\frac{N^{4\iota}\tau_0^2}{C_3}\exp\left((1-\eta)^\eta \frac{(N^{\frac{1}{2}+2\iota}\tau_0)^{\frac{\eta(1-\eta)}{2}}}{C_4 2^\eta}\right)\right), \tag{62}$$

*where $V$ is defined in (8) and $C_1, C_2, C_3$ are some positive constants.*

2. *under condition (a), (b), and (c)-(ii) of Assumption 2.1, for $0 < \iota < (1 - 1/\eta_2)/4$,*

$$\|\hat{f} - f^*\|_{L_2} \leq \max\left\{N^{-\frac{1}{4}\left(1 - \frac{1}{\eta_2}\right)+\iota}, \omega_\mu(\mathcal{F} - \mathcal{F}, \tau Q_{\mathcal{F}-\mathcal{F}}(2\tau)/16)\right\} \tag{63}$$

*with probability at least*

$$1 - 2\exp\left(-\frac{N^r Q_{\mathcal{H}}(2\tau)^3 c^{\frac{1}{\eta_1}}}{2(4 - Q_{\mathcal{H}}(2\tau))^2}\right) - \frac{N^r Q_{\mathcal{H}}(2\tau) c^{\frac{1}{\eta_1}}}{4}\exp(-N^{(1-r)\eta_1}) - 8\tau_0^{-\frac{2\eta_2}{1+\eta_2}} N^{-\frac{4\iota\eta_2}{1+\eta_2}}$$

$$-\frac{2^{\eta_2+3}c'^{\frac{1-\eta_2}{\eta_1}}\tau_0^{-\frac{2\eta_2}{1+\eta_2}}}{\left(\log\left(\tau_0 N^{\frac{1}{2}+\frac{1}{2\eta_2}+2\iota}\right)/2\right)^{\frac{1-\eta_2}{\eta_1}}} N^{-\frac{4\iota\eta_2}{1+\eta_2}} - 2e^{-\frac{\tau_0^{\frac{2\eta_2}{1+\eta_2}}\left(\log\left(\tau_0 N^{\frac{1}{2}+\frac{1}{2\eta_2}+2\iota}\right)/2\right)^{1/\eta_1}}{9c'^{1/\eta_1}}} N^{\frac{4\iota\eta_2}{1+\eta_2}} \tag{64}$$

*Proof.* [Proof of Theorem B.1] We first prove **Part 1**. We will denote the class $\mathcal{F} - f^*$ by $\mathcal{H}$. From Lemma B.2 it follows that if $\rho > \omega_\mu(\mathcal{H}, \tau Q_{\mathcal{F}-\mathcal{F}}(2\tau)/16)$, then with probability at least

$$\mathscr{P}_1 = 1 - 2\exp\left(-\frac{NQ_{\mathcal{H}}(2\tau)^3}{2(4 - Q_{\mathcal{H}}(2\tau))^2}\left(\frac{c}{\mathscr{G}(N)}\right)^{\frac{1}{\eta_1}}\right) - \frac{NQ_{\mathcal{H}}(2\tau)c^{\frac{1}{\eta_1}}}{4\mathscr{G}(N)^{\frac{1}{\eta_1}}}\exp(-\mathscr{G}(N)),$$

for every $f \in \mathcal{F}$ that satisfies $\|f - f^*\|_{L_2} \geq \rho$,

$$\frac{1}{N}\sum_{i=1}^{N}(f - f^*)^2(X_i) \geq \frac{\tau^2\|f - f^*\|_{L_2}^2 Q_{\mathcal{H}}(2\tau)}{4} \tag{65}$$

So, with probability at least $\mathscr{P}_1$, for every $f \in \mathcal{F}$ that satisfies $\|f - f^*\|_{L_2} \geq \rho$,

$$P_N \mathcal{L}_f \geq 2\left(\frac{1}{N}\sum_{i=1}^{N}\xi_i(f - f^*)(X_i) - \mathbb{E}\left[\xi(f - f^*)\right]\right) + \frac{\tau^2\|f - f^*\|_{L_2}^2 Q_{\mathcal{H}}(2\tau)}{4}. \tag{66}$$

When $\|f - f^*\|_{L_2} \geq \mathscr{A}(N) > 2(N\tau_0)^{-1/2}$, we have $\log(N\tau_0\|f - f^*\|_{L_2}^2) \leq 2(N\tau_0\|f - f^*\|_{L_2}^2)^{(1-\eta)/2}/(1-\eta)$. Under Conditions (a), (c)-(i), and (d) of Assumption 2.1, using Lemma 2.1, we get

$$\mathbb{P}\left(\left|\frac{1}{N}\sum_{i=1}^{N}\xi_i(f - f^*)(X_i) - \mathbb{E}\left[\xi(f - f^*)\right]\right| \geq \tau_0\|f - f^*\|_{L_2}^2\right)$$
$$\leq N\exp\left(-\frac{(N\tau_0\|f - f^*\|_{L_2}^2)^\eta}{C_1}\right) + \exp\left(-\frac{N^2\tau_0^2\|f - f^*\|_{L_2}^4}{C_2(1 + NV)}\right)$$

$$+ \exp\left(-\frac{N\tau_0^2\|f-f^*\|_{L_2}^4}{C_3}\exp\left(\frac{(N\tau_0\|f-f^*\|_{L_2}^2)^{\eta(1-\eta)}}{C_4(\log(N\tau_0\|f-f^*\|_{L_2}^2))^{\eta}}\right)\right)$$

$$\leq N\exp\left(-\frac{(N\tau_0\|f-f^*\|_{L_2}^2)^{\eta}}{C_1}\right) + \exp\left(-\frac{N^2\tau_0^2\|f-f^*\|_{L_2}^4}{C_2(1+NV)}\right)$$

$$+ \exp\left(-\frac{N\tau_0^2\|f-f^*\|_{L_2}^4}{C_3}\exp\left(\frac{(1-\eta)^{\eta}(N\tau_0\|f-f^*\|_{L_2}^2)^{\frac{\eta(1-\eta)}{2}}}{C_4 2^{\eta}}\right)\right)$$

$$\leq N\exp\left(-\frac{(N\tau_0\mathscr{A}(N)^2)^{\eta}}{C_1}\right) + \exp\left(-\frac{N^2\tau_0^2\mathscr{A}(N)^4}{C_2(1+NV)}\right)$$

$$+ \exp\left(-\frac{N\tau_0^2\mathscr{A}(N)^4}{C_3}\exp\left(\frac{(1-\eta)^{\eta}(N\tau_0\mathscr{A}(N)^2)^{\frac{\eta(1-\eta)}{2}}}{C_4 2^{\eta}}\right)\right) \equiv \mathscr{P}_2, \tag{67}$$

where

$$V \leq \mathbb{E}\left[(\xi_1(f-f^*)(X_1))^2\right] + 4\sum_{i\geq 0}\mathbb{E}\left[B_i\left(\xi_1(f-f^*)(X_1)\right)^2\right],$$

$\{B_i\}$ is some sequence such that $B_i \in [0,1]$, $\mathbb{E}[B_i] \leq \beta(i)$ and $C_1, C_2, C_3$ are constants which depend on $c, \eta, \eta_1, \eta_2$. Observe that,

$$V \leq \mathbb{E}\left[(\xi_1(f-f^*)(X_1))^2\right] + 4\sum_{i\geq 0}\mathbb{E}\left[B_i\left(\xi_1(f-f^*)(X_1)\right)^2\right]$$

$$\leq \mathbb{E}\left[(\xi_1(f-f^*)(X_1))^2\right] + 4\sum_{i\geq 0}\sqrt{\mathbb{E}[B_i^2]\,\mathbb{E}\left[(\xi_1(f-f^*)(X_1))^4\right]}$$

$$\leq \mathbb{E}\left[(\xi_1(f-f^*)(X_1))^2\right] + 4\sqrt{\mathbb{E}\left[(\xi_1(f-f^*)(X_1))^4\right]}\sum_{i\geq 0}\sqrt{\mathbb{E}[B_i]}$$

$$\leq \mathbb{E}\left[(\xi_1(f-f^*)(X_1))^2\right] + 4\sqrt{\mathbb{E}\left[(\xi_1(f-f^*)(X_1))^4\right]}\sum_{i\geq 0}\sqrt{\beta(i)}$$

$$\leq \mathbb{E}\left[(\xi_1(f-f^*)(X_1))^2\right] + 4\sqrt{\mathbb{E}\left[(\xi_1(f-f^*)(X_1))^4\right]}\sum_{i\geq 0}\exp(-ci^{\eta_1}/2)$$

$$\leq 2^{\frac{2}{\eta_2}} + C4^{1+\frac{2}{\eta_2}}.$$

Combining (66), and (67), with probability at least $\mathscr{P}_1 - \mathscr{P}_2$, for every $f \in \mathcal{F}$ that satisfies $\|f-f^*\|_{L_2} \geq \max(\rho, \mathscr{A}(N))$, we get

$$P_N L_f \geq -2\tau_0\|f-f^*\|_{L_2}^2 + \frac{\tau^2\|f-f^*\|_{L_2}^2 Q_{\mathcal{H}}(2\tau)}{4}.$$

Choosing $\tau_0 < \tau^2 Q_{\mathcal{H}}(2\tau)/8$, we have,

$$P_N L_f > 0.$$

But the empirical minimizer $\hat{f}$ satisfies $P_N L_{\hat{f}} \leq 0$. This implies, together with choosing $\mathscr{A}(N) = N^{-1/4+\iota}$, that with probability at least $\mathscr{P} = \mathscr{P}_1 - \mathscr{P}_2$,

$$\|\hat{f}-f^*\|_{L_2} \leq \max(\omega_{\mu}(\mathcal{F}-\mathcal{F}, \tau Q_{\mathcal{F}-\mathcal{F}}(2\tau)/16), \mathscr{A}(N)),$$

where

$$\mathscr{P} = 1 - 2\exp\left(-\frac{NQ_{\mathcal{H}}(2\tau)^3}{2(4-Q_{\mathcal{H}}(2\tau))^2}\left(\frac{c}{\mathscr{G}(N)}\right)^{\frac{1}{\eta_1}}\right) - \frac{NQ_{\mathcal{H}}(2\tau)c^{\frac{1}{\eta_1}}}{4\mathscr{G}(N)^{\frac{1}{\eta_1}}}\exp(-\mathscr{G}(N))$$

$$-N\exp\left(-\frac{(N^{\frac{1}{2}+2\iota}\tau_0)^{\eta}}{C_1}\right) - \exp\left(-\frac{N^{1+4\iota}\tau_0^2}{C_2(1+NV)}\right) - \exp\left(-\frac{N^{4\iota}\tau_0^2}{C_3}\exp\left((1-\eta)^{\eta}\frac{(N^{\frac{1}{2}+2\iota}\tau_0)^{\frac{\eta(1-\eta)}{2}}}{C_4 2^{\eta}}\right)\right).$$

Choosing $\mathscr{G}(N) = N^{(1-r)\eta_1}$ for some $0 < r < 1$, we get,

$$\mathscr{P} = 1 - 2\exp\left(-\frac{N^r Q_{\mathcal{H}}(2\tau)^3 c^{\frac{1}{\eta_1}}}{2(4 - Q_{\mathcal{H}}(2\tau))^2}\right) - \frac{N^r Q_{\mathcal{H}}(2\tau) c^{\frac{1}{\eta_1}}}{4}\exp(-N^{(1-r)\eta_1})$$

$$- N\exp\left(-\frac{(N^{\frac{1}{2}+2\iota}\tau_0)^\eta}{C_1}\right) - \exp\left(-\frac{N^{1+4\iota}\tau_0^2}{C_2(1 + NV)}\right)$$

$$- \exp\left(-\frac{N^{4\iota}\tau_0^2}{C_3}\exp\left((1-\eta)^\eta \frac{(N^{\frac{1}{2}+2\iota}\tau_0)^{\frac{\eta(1-\eta)}{2}}}{C_4 2^\eta}\right)\right),$$

and

$$\mu = \frac{N^r Q_{\mathcal{H}}(2\tau) c^{\frac{1}{\eta_1}}}{4}. \tag{68}$$

We now prove **part 2**. Since Lemma B.1 only depends on Condition (a) and (b) of Assumption 2.1, Lemma B.1 remains unchanged in this case. To deal with the multiplier process we need a concentration result similar to Lemma 2.1. So we use the concentration inequality we proved in Lemma 2.2. When $\|f - f^*\|_{L_2} \geq \mathscr{A}(N)$, using Lemma 2.2 we have

$$\mathbb{P}\left(\left|\frac{1}{N}\sum_{i=1}^N \xi_i(f - f^*)(X_i) - \mathbb{E}\left[\xi(f - f^*)\right]\right| \geq \tau_0 \|f - f^*\|_{L_2}^2\right)$$

$$\leq \frac{2^{\eta_2+3}}{(d_2 \log N\tau_0\|f - f^*\|_{L_2}^2)^{\frac{1-\eta_2}{\eta_1}}} N(N\tau_0\|f - f^*\|_{L_2}^2)^{-(1+d_1(\eta_2-1))} + 8N(N\tau_0\|f - f^*\|_{L_2}^2)^{-(1+d_2c')}$$

$$+ 2e^{-\frac{(N\tau_0\|f-f^*\|_{L_2}^2)^{2-2d_1}(d_2 \log(N\tau_0\|f-f^*\|_{L_2}^2))^{1/\eta_1}}{9N}}$$

$$\leq \frac{2^{\eta_2+3}}{(d_2 \log(N\tau_0\mathscr{A}(N)^2))^{\frac{1-\eta_2}{\eta_1}}} N(N\tau_0\mathscr{A}(N)^2)^{-(1+d_1(\eta_2-1))} + 8N(N\tau_0\mathscr{A}(N)^2)^{-(1+d_2c')}$$

$$+ 2e^{-\frac{(N\tau_0\mathscr{A}(N)^2)^{2-2d_1}(d_2 \log(N\tau_0\mathscr{A}(N)^2))^{1/\eta_1}}{9N}} \equiv \mathscr{P}_2. \tag{69}$$

We will choose $d_1$ suitably to allow $\mathscr{A}(N)$ to decrease with $N$ as fast as possible while ensuring $\lim_{N\to\infty} \mathscr{P}_2 \to 0$. Combining (66), and (69), with probability at least $\mathscr{P}_1 - \mathscr{P}_2$, for every $f \in \mathcal{F}$ that satisfies $\|f - f^*\|_{L_2} \geq \max(\rho, \mathscr{A}(N))$, we get

$$P_N L_f \geq -2\tau_0\|f - f^*\|_{L_2}^2 + \frac{\tau^2\|f - f^*\|_{L_2}^2 Q_{\mathcal{H}}(2\tau)}{4}.$$

Choosing $\tau_0 < \tau^2 Q_{\mathcal{H}}(2\tau)/8$, we have,

$$P_N L_f > 0.$$

But the empirical minimizer $\hat{f}$ satisfies $P_N L_{\hat{f}} \leq 0$. This implies, together with choosing $\mathscr{A}(N) = N^{-(1-1/\eta_2)/4+\iota}$, $d_1 = 1/(1 + \eta_2)$, $d_2 = (\eta_2 - 1)/(\eta_2 + 1)$, and $\iota < (1 - 1/\eta_2)/4$, that with probability at least $\mathscr{P} = \mathscr{P}_1 - \mathscr{P}_2$,

$$\|\hat{f} - f^*\|_{L_2} \leq \max(\omega_\mu(\mathcal{F} - \mathcal{F}, \tau Q_{\mathcal{F}-\mathcal{F}}(2\tau)/16), \mathscr{A}(N)),$$

where

$$\mathscr{P} = 1 - 2\exp\left(-\frac{NQ_{\mathcal{H}}(2\tau)^3}{2(4 - Q_{\mathcal{H}}(2\tau))^2}\left(\frac{c}{\mathscr{G}(N)}\right)^{\frac{1}{\eta_1}}\right) - \frac{NQ_{\mathcal{H}}(2\tau)c^{\frac{1}{\eta_1}}}{4\mathscr{G}(N)^{\frac{1}{\eta_1}}}\exp(-\mathscr{G}(N))$$

$$- \frac{2^{\eta_2+3}\tau_0^{-\frac{2\eta_2}{1+\eta_2}}}{\left(\log\left(\tau_0 N^{\frac{1}{2}+\frac{1}{2\eta_2}+2\iota}\right)/2\right)^{\frac{1-\eta_2}{\eta_1}}} N^{-\frac{4\iota\eta_2}{1+\eta_2}} - 8\tau_0^{-\frac{2\eta_2}{1+\eta_2}} N^{-\frac{4\iota\eta_2}{1+\eta_2}} - 2e^{-\frac{\tau_0^{\frac{2\eta_2}{1+\eta_2}}\left(\log\left(\tau_0 N^{\frac{1}{2}+\frac{1}{2\eta_2}+2\iota}\right)/2\right)^{1/\eta_1}}{9}} N^{\frac{4\iota\eta_2}{1+\eta_2}}.$$

Choosing $\mathscr{G}(N) = N^{(1-r)\eta_1}$ for some $0 < r < 1$, we get,

$$\mathscr{P} = 1 - 2\exp\left(-\frac{N^r Q_{\mathcal{H}}(2\tau)^3 c^{\frac{1}{\eta_1}}}{2(4 - Q_{\mathcal{H}}(2\tau))^2}\right) - \frac{N^r Q_{\mathcal{H}}(2\tau) c^{\frac{1}{\eta_1}}}{4}\exp(-N^{(1-r)\eta_1})$$

$$- \frac{2^{\eta_2+3}\tau_0^{-\frac{2\eta_2}{1+\eta_2}}}{\left(\log\left(\tau_0 N^{\frac{1}{2}+\frac{1}{2\eta_2}+2\iota}\right)/2\right)^{\frac{1-\eta_2}{\eta_1}}} N^{-\frac{4\iota\eta_2}{1+\eta_2}} - 8\tau_0^{-\frac{2\eta_2}{1+\eta_2}} N^{-\frac{4\iota\eta_2}{1+\eta_2}} - 2e^{-\frac{\tau_0^{\frac{2\eta_2}{1+\eta_2}}\left(\log\left(\tau_0 N^{\frac{1}{2}+\frac{1}{2\eta_2}+2\iota}\right)/2\right)^{1/\eta_1}}{9}} N^{\frac{4\iota\eta_2}{1+\eta_2}},$$

and

$$\mu = \frac{N^r Q_{\mathcal{H}}(2\tau) c^{\frac{1}{\eta_1}}}{4}. \tag{70}$$

∎

*Proof.* [Proof of Corollary 3.1] Note that Assumption 3.1 implies Condition (b) of Assumption 2.1 as shown in Lemma 4.1 in [Men15]. Under Assumption 3.1 with $p = 8$, using Cauchy-Schwarz inequality we have,

$$V \leq \mathbb{E}\left[(\xi_1(f - f^*)(X_1))^2\right] + 4\sqrt{\mathbb{E}\left[(\xi_1(f - f^*)(X_1))^4\right]}\sum_{i\geq 0}\exp(-ci^{\eta_1}/2)$$

$$\leq \sqrt{\mathbb{E}\left[\xi_1^4\right]}\sqrt{\mathbb{E}\left[((f - f^*)(X_1))^4\right]} + 4C\sqrt{\sqrt{\mathbb{E}\left[\xi_1^8\right]}\sqrt{\mathbb{E}\left[((f - f^*)(X_1))^8\right]}}$$

$$\leq M_1^2\|f - f^*\|_{L_2}^2\left(\sqrt{\mathbb{E}\left[\xi_1^4\right]} + 4C\left(\mathbb{E}\left[\xi_1^8\right]\right)^{\frac{1}{4}}\right)$$

$$\leq M_2^2\|f - f^*\|_{L_2}^2,$$

for some constant $M_2$. Then, from (67) we have,

$$\mathbb{P}\left(\left|\frac{1}{N}\sum_{i=1}^N \xi_i(f - f^*)(X_i) - \mathbb{E}\left[\xi(f - f^*)\right]\right| \geq \tau_0\|f - f^*\|_{L_2}^2\right)$$

$$\leq N\exp\left(-\frac{(N\tau_0\|f - f^*\|_{L_2}^2)^\eta}{C_1}\right) + \exp\left(-\frac{N^2\tau_0^2\|f - f^*\|_{L_2}^4}{C_2(1 + NV)}\right)$$

$$+ \exp\left(-\frac{N\tau_0^2\|f - f^*\|_{L_2}^4}{C_3}\exp\left(\frac{(N\tau_0\|f - f^*\|_{L_2}^2)^{\eta(1-\eta)}}{C_4(\log(N\tau_0\|f - f^*\|_{L_2}^2))^\eta}\right)\right)$$

$$\leq N\exp\left(-\frac{(N\tau_0\|f - f^*\|_{L_2}^2)^\eta}{C_1}\right) + \exp\left(-\frac{N^2\tau_0^2\|f - f^*\|_{L_2}^4}{C_2(1 + NM_2^2\|f - f^*\|_{L_2}^2)}\right)$$

$$+ \exp\left(-\frac{N\tau_0^2\|f - f^*\|_{L_2}^4}{C_3}\exp\left(\frac{(N\tau_0\|f - f^*\|_{L_2}^2)^{\eta(1-\eta)}}{C_4(\log(N\tau_0\|f - f^*\|_{L_2}^2))^\eta}\right)\right).$$

If $N\|f - f^*\|_{L_2}^2 \geq N\mathscr{A}(N)^2 \geq \max(1/M_2^2, 1/\tau_0)$, then

$$\mathbb{P}\left(\left|\frac{1}{N}\sum_{i=1}^N \xi_i(f - f^*)(X_i) - \mathbb{E}\left[\xi(f - f^*)\right]\right| \geq \tau_0\|f - f^*\|_{L_2}^2\right)$$

$$\leq N\exp\left(-\frac{(N\tau_0\|f - f^*\|_{L_2}^2)^\eta}{C_1}\right) + \exp\left(-\frac{N\tau_0^2\|f - f^*\|_{L_2}^2}{2C_2 M_2^2}\right)$$

$$+ \exp\left(-\frac{N\tau_0^2\|f - f^*\|_{L_2}^4}{C_3}\exp\left(\frac{(1 - \eta)^\eta(N\tau_0\mathscr{A}(N)^2)^{\frac{\eta(1-\eta)}{2}}}{C_4 2^\eta}\right)\right),$$

and the second term dominates the third term in the above expression. Now if choose $\mathscr{A}(N) = N^{-1/2+\iota}$, then with probability at least

$$\mathscr{P} = 1 - \exp\left(-\frac{N^{2\iota}\tau_0^2}{2C_2 M_2^2}\right) - \exp\left(-\frac{N^{4\iota-1}\tau_0^2}{C_3}\exp\left(\frac{(1 - \eta)^\eta(N^{2\iota}\tau_0)^{\frac{\eta(1-\eta)}{2}}}{C_4 2^\eta}\right)\right),$$

we get

$$\|\hat{f} - f^*\|_{L_2} \leq \max\left(N^{-1/2+\iota}, \omega_\mu(\mathcal{F} - \mathcal{F}, \tau Q_{\mathcal{F}-\mathcal{F}}(2\tau)/16)\right).$$

∎

## B.2  Proofs for convex loss

Recall the decomposition (1)

$$P_N L_f \geq \frac{1}{16N} \sum_{i=1}^{N} \ell''(\widetilde{\xi}_i)(f - f^*)^2(X_i) + \frac{1}{N} \sum_{i=1}^{N} \ell'(\xi_i)(f - f^*)(X_i).$$

Since $\mathcal{F}$ is convex, we also have

$$\mathbb{E}\left[\ell'(\xi)(f - f^*)(X)\right] \geq 0.$$

Then,

$$P_N L_f \geq \frac{1}{16N} \sum_{i=1}^{N} \ell''(\widetilde{\xi}_i)(f - f^*)^2(X_i) + \frac{1}{N} \sum_{i=1}^{N} (\ell'(\xi_i)(f - f^*)(X_i) - \mathbb{E}\left[\ell'(\xi_i)(f - f^*)(X_i)\right]).$$

(71)

Now our goal is to establish a lower bound (Proposition B.1) on the first term of the RHS of (71), and a two-sided bound ((93) and (94)) on the second term when $\|f - f^*\|_{L_2}$ is large. Combining these bounds we will show that if $\|f - f^*\|_{L_2}$ is large then $P_N L_f > 0$ which implies $f$ cannot be a minimizer of empirical risk because for the minimizer $\hat{f}$ we have $P_N L_{\hat{f}} \leq 0$. Let $\rho(t_1, t_2) :=$ $\inf\{l''(x) : x \in [t_1, t_2], 0 \leq t_1 < t_2\}$. First we prove the following extension of bounded difference inequality to the $\beta$-mixing sequence which we will use frequently in our proofs.

**Lemma B.3** (Bounded difference inequality for strictly stationary $\beta$-mixing sequence). *Let $\{U_i\}_{i=1}^N$ be a sample from a strictly stationary $\beta$-mixing sequence, $|U_i| \leq M$, and $\mathbb{E}[U_i] = U^*$. Let $N > a, b, \mu > 0$ be such that $(a+b)\mu = N$. Then with probability at least $1 - \exp\left(-\frac{(t-2b\mu)^2}{2\mu a^2 M^2}\right) - 2M(\mu - 1)\beta(b)$, we have $\forall t > 2b\mu$,*

$$\sum_{i=1}^{N} U_i \leq NU^* + t.$$

*Proof.* Consider the partition as in (6). Then, using Corollary 2.7 of [Yu94], we get $\forall t > 2b\mu$,

$$\mathbb{P}\left(\sum_{i=1}^{N}(U_i - U^*) \geq t\right)$$

$$\leq \mathbb{P}\left(\sum_{i=1}^{\mu}\sum_{j=1}^{a}(U_i - U^*) \geq t - 2b\mu\right)$$

$$\leq \mathbb{P}\left(\sum_{i=1}^{\mu}\sum_{j=1}^{a}(\tilde{U}_{(a+b)(i-1)+j} - U^*) \geq t - 2b\mu\right) + 2M(\mu - 1)\beta(b).$$

where $\sum_{j=1}^{a}(\tilde{U}_{(a+b)(i-1)+j} - U^*)$ is an `iid` sequence for $i = 1, 2, \cdots, \mu$. Using bounded difference inequality,

$$\mathbb{P}\left(\sum_{i=1}^{N}(U_i - U^*) \geq t\right) \leq \exp\left(-\frac{(t - 2b\mu)^2}{2\mu a^2 M^2}\right) + 2M(\mu - 1)\beta(b).$$

So with probability at least $1 - \exp\left(-\frac{(t-2b\mu)^2}{2\mu a^2 M^2}\right) - 2M(\mu - 1)\beta(b)$,

$$\sum_{i=1}^{N} U_i \leq NU^* + t.$$

Lemma B.4-B.6 are needed to prove Lemma B.7 which is the main result needed to prove Proposition B.1.

**Lemma B.4.** *Let $X_i, i = 1, 2, \cdots, N$ be a sample from a sequence for which condition (a) of Assumption 2.1 is true. For every $0 < Q_{\mathcal{H}}(2\tau) < 1$, we have that with probability at least $1 - c_1 Q_{\mathcal{H}}(2\tau)^{1-\frac{1}{\eta_1}} N^{\eta_1/(1+\eta_1)} e^{-c_2 Q_{\mathcal{H}}(2\tau)^{1+\frac{1}{\eta_1}} N^{\eta_1/(1+\eta_1)}}$, for some constants $c_1, c_2 > 0$, there is a subset $S \subset \{1, 2, \cdots, N\}$ such that $|S| \geq N(1 - Q_{\mathcal{H}}(2\tau))$, and $\forall i \in S$,*

$$|X_i| \leq \frac{2\|X_i\|_{L_2}}{\sqrt{Q_{\mathcal{H}}(2\tau)}}. \tag{72}$$

*Proof.* Let $\zeta_i = \mathbb{1}\left(|X_i| \geq \frac{2\|X_i\|_{L_2}}{\sqrt{Q_{\mathcal{H}}(2\tau)}}\right)$. Then, by Markov's inequality,

$$\mathbb{E}[\zeta_i] = \mathbb{P}\left(|X_i| \geq \frac{2\|X_i\|_{L_2}}{\sqrt{Q_{\mathcal{H}}(2\tau)}}\right) \leq Q_{\mathcal{H}}(2\tau)/4.$$

Then using Lemma B.3, we have with probability at least $1 - \exp\left(-\frac{(t-2b\mu)^2}{2\mu a^2}\right) - 2(\mu - 1)\beta(b)$,

$$\sum_{i=1}^{N} \zeta_i \leq \frac{NQ_{\mathcal{H}}(2\tau)}{4} + t.$$

Now, setting

$$t = \frac{3NQ_{\mathcal{H}}(2\tau)}{4} \quad a = \frac{(4 - Q_{\mathcal{H}}(2\tau))N^{\frac{1}{1+\eta_1}}}{c^{\frac{1}{\eta_1}} Q_{\mathcal{H}}(2\tau)^{1-\frac{1}{\eta_1}}} \quad b = \frac{Q_{\mathcal{H}}(2\tau)^{\frac{1}{\eta_1}} N^{\frac{1}{1+\eta_1}}}{c^{\frac{1}{\eta_1}}} \quad \mu = \frac{N^{\frac{\eta_1}{1+\eta_1}} c^{\frac{1}{\eta_1}} Q_{\mathcal{H}}(2\tau)^{\frac{\eta_1 - 1}{\eta_1}}}{4},$$
$$\tag{73}$$

we have $\sum_{i=1}^{N} \zeta_i \leq NQ_{\mathcal{H}}(2\tau)$, with probability at least

$$1 - c_1 Q_{\mathcal{H}}(2\tau)^{1-\frac{1}{\eta_1}} N^{\eta_1/(1+\eta_1)} e^{-c_2 Q_{\mathcal{H}}(2\tau) N^{\eta_1/(1+\eta_1)}}.$$

$\blacksquare$

**Lemma B.5.** *Let $X_i, i = 1, 2, \cdots, N$ be a sample from a sequence for which condition (a) and (b) of Assumption 2.1 is true. Then with probability at least with*

$$1 - c_1 Q_{\mathcal{H}}(2\tau)^{1-\frac{1}{\eta_1}} N^{\eta_1/(1+\eta_1)} e^{-c_2 Q_{\mathcal{H}}(2\tau) N^{\eta_1/(1+\eta_1)}},$$

*there is a subset $S \subset \{1, 2, \cdots, N\}$ such that $|S| \geq 3NQ_{\mathcal{H}}(2\tau)/4$, and $\forall i \in S$, $|X_i| \geq 2\tau\|X_i\|_{L_2}$.*

*Proof.* Let $\zeta_i = \mathbb{1}(|X_i| \geq 2\tau\|X_i\|_{L_2})$. Using condition (b) of Assumption 2.1, we have $\mathbb{E}[\zeta_i] > Q_{\mathcal{H}}(2\tau)$. Then using Lemma B.3, with probability at least

$$1 - c_1 Q_{\mathcal{H}}(2\tau)^{1-\frac{1}{\eta_1}} N^{\eta_1/(1+\eta_1)} e^{-c_2 Q_{\mathcal{H}}(2\tau) N^{\eta_1/(1+\eta_1)}},$$

we have

$$3NQ_{\mathcal{H}}(2\tau)/4 \leq \sum_{i=1}^{N} \zeta_i \leq 5N\mathbb{E}[\zeta_i]/4.$$

$\blacksquare$

**Lemma B.6.** *Let $X_i, i = 1, 2, \cdots, N$ be a sample from a sequence for which conditions (a) and (b) of Assumption 2.1 is true. Then with probability at least with*

$$1 - c_1 Q_{\mathcal{H}}(2\tau)^{1-\frac{1}{\eta_1}} N^{\eta_1/(1+\eta_1)} e^{-c_2 Q_{\mathcal{H}}(2\tau) N^{\eta_1/(1+\eta_1)}},$$

*there is a subset $S \subset \{1, 2, \cdots, N\}$ such that $|S| \geq NQ_{\mathcal{H}}(2\tau)/2$, and $\forall i \in S$,*

$$2\tau\|X_i\|_{L_2} \leq |X_i| \leq \frac{2\|X_i\|_{L_2}}{\sqrt{Q_{\mathcal{H}}(2\tau)}}.$$

*Proof.* The proof is immediate from Lemma B.4, and Lemma B.5. ∎

**Lemma B.7.** *Let $\mathcal{H}$ be a class of function which is star-shaped around $0$ and satisfies condition (b) of Assumption 2.1. If $\zeta_1 \sim 2\tau Q_{\mathcal{H}}(2\tau)^{3/2}$, $\zeta_2 \sim 2\tau Q_{\mathcal{H}}(2\tau)$, and $r = \|h\|_{L_2} > \omega_Q(\zeta_1, \zeta_2)$, there is a set $V_r \subset \mathcal{H} \cap rS(L_2)$ such that there is an event $\mathcal{A}$ with probability at least $1 - c_6 Q_{\mathcal{H}}(2\tau)^{1-\frac{1}{\eta_1}} N^{\eta_1/(1+\eta_1)} e^{-c_7 Q_{\mathcal{H}}(2\tau) N^{\eta_1/(1+\eta_1)}}$ we have:*

1.
$$|V_r| \leq \exp(c_2' Q_{\mathcal{H}}(2\tau) N^{\eta_1/(1+\eta_1)}/2), \tag{74}$$

   *where $c_2' \leq 1/1000$*

2. *For every $v \in V_r$ there is a subset $S_v \subset \{1, 2, \cdots, N\}$ such that $|S_v| \geq Q_{\mathcal{H}}(2\tau)N/2$, and for every $i \in S_v$,*

$$2\tau r \leq |v(X_i)| \leq \frac{c_3 r}{\sqrt{Q_{\mathcal{H}}(2\tau)}}. \tag{75}$$

3. *For every $h \in \mathcal{H} \cap rS(L_2)$ there is some $v \in V_r$, and a subset $K_h \subset S_v$, containing at least $3/4$ of the coordinates of $S_v$, and for every $k \in K_h$,*

$$\tau \|h\|_{L_2} \leq |h(X_k)| \leq c_9 \left( 2\tau + \frac{1}{\sqrt{Q_{\mathcal{H}}(2\tau)}} \right) \|h\|_{L_2}, \tag{76}$$

   *and $h(X_k)$ and $v(X_k)$ have the same sign.*

*Proof.* Let $r = \|h\|_{L_2} > \omega_Q(\zeta_1, \zeta_2)$. Let $V_r \subset H \cap rS(L_2)$ be a maximal $\rho$-separated set such that

$$|V_r| \leq \exp(c_2' Q_{\mathcal{H}}(2\tau) N^{\eta_1/(1+\eta_1)}/2)$$

where $c_2' = \min(c_2, 1/500)$. Applying Lemma B.6 on all the elements of $V_r$, using union bound we obtain that with probability at least

$$1 - c_1 Q_{\mathcal{H}}(2\tau)^{1-\frac{1}{\eta_1}} N^{\eta_1/(1+\eta_1)} e^{-c_2 Q_{\mathcal{H}}(2\tau) N^{\eta_1/(1+\eta_1)}/2},$$

for every $v \in V_r$ there is a subset $S_v$ such that $|S_v| \geq N Q_{\mathcal{H}}(2\tau)/2$ and for all $i \in S_v$, we have

$$2\tau \|v(X_i)\|_{L_2} \leq |v(X_i)| \leq \frac{c_3 \|v(X_i)\|_{L_2}}{\sqrt{Q_{\mathcal{H}}(2\tau)}}. \tag{77}$$

Since we have assumed $r > \omega_1(\zeta_1)$, from Sudakov's inequality we have,

$$\rho \leq c_4 \frac{\sqrt{2}\mathbb{E}\left[\|G\|_{\mathcal{H} \cap rS(L_2)}\right]}{\sqrt{c_2 Q_{\mathcal{H}}(2\tau) N^{\eta_1/(1+\eta_1)}}} \leq \frac{c_5 \zeta_1 r}{\sqrt{Q_{\mathcal{H}}(2\tau)}}, \tag{78}$$

where $c_5 = \sqrt{2} c_4/\sqrt{c_2}$. For all $h \in \mathcal{H} \cap rS(L_2)$, let $h_v \in V_r$ so that $\|h - h_v\|_{L_2} \leq \rho$. Now let $\delta_h = \mathbb{1}_{(|h-h_v|>\tau r)}$ and put

$$\Delta_r = \{\delta_h : h \in \mathcal{H} \cap rS(L_2)\}. \tag{79}$$

Define a function $\psi_1(t) = \max(\min(t/(\tau r), 1), 0)$. Observe that $\delta_h(X) \leq \psi_1(|h - h_v|(X))$. Now we want to show that the number of points where $|h - h_v| > \tau r$ is small.

$$\mathbb{E}\left[\sup_{\delta_h \in \Delta_r} \frac{1}{N} \sum_{i=1}^{N} \delta_h(X_i)\right]$$

$$\leq \mathbb{E}\left[\sup_{h \in \mathcal{H} \cap rS(L_2)} \frac{1}{N} \sum_{i=1}^{N} \psi_1(|h - h_v|(X_i))\right]$$

$$\leq \mathbb{E}\left[\sup_{h \in \mathcal{H} \cap rS(L_2)} \frac{1}{N} \sum_{i=1}^{N} \left(\psi_1(|h - h_v|(X_i)) - \mathbb{E}\left[\psi_1(|h - h_v|(X))\right]\right)\right] + \mathbb{E}\left[\sup_{h \in \mathcal{H} \cap rS(L_2)} \mathbb{E}\left[\psi_1(|h - h_v|(X))\right]\right],$$

where $X \sim \pi$. Consider the partition introduced in (6). Then,

$$\mathbb{E}\left[\sup_{\delta_h \in \Delta_r} \frac{1}{N}\sum_{i=1}^{N}\delta_h(X_i)\right]$$

$$\leq \mathbb{E}\left[\sup_{h \in \mathcal{H} \cap rS(L_2)} \frac{1}{N}\sum_{i=1}^{\mu}\sum_{j=1}^{a}\left(\psi_1(|h-h_v|\,(X_{(a+b)(i-1)+j})) - \mathbb{E}\left[\psi_1(|h-h_v|\,(X))\right]\right)\right]$$

$$+\frac{2b\mu}{N} + \frac{1}{\tau r}\mathbb{E}\left[\sup_{h \in \mathcal{H} \cap rS(L_2)} \mathbb{E}\left[|h-h_v|\,(X)\right]\right]$$

$$\leq \frac{\mu}{N}\sum_{j=1}^{a}\mathbb{E}\left[\sup_{h \in \mathcal{H} \cap rS(L_2)} \frac{1}{\mu}\sum_{i=1}^{\mu}\left(\psi_1(|h-h_v|\,(X_{(a+b)(i-1)+j})) - \mathbb{E}\left[\psi_1(|h-h_v|\,(X))\right]\right)\right]$$

$$+\frac{2b\mu}{N} + \frac{\rho}{\tau r}$$

$$\leq \frac{\mu}{N}\sum_{j=1}^{a}\mathbb{E}\left[\sup_{h \in \mathcal{H} \cap rS(L_2)} \frac{1}{\mu}\sum_{i=1}^{\mu}\left(\psi_1(|h-h_v|\,(\tilde{X}_{(a+b)(i-1)+j})) - \mathbb{E}\left[\psi_1(|h-h_v|\,(X))\right]\right)\right]$$

$$+2(\mu-1)\beta(a+b) + \frac{2b\mu}{N} + \frac{\rho}{\tau r}.$$

Now using symmetrization, we get

$$\mathbb{E}\left[\sup_{\delta_h \in \Delta_r} \frac{1}{N}\sum_{i=1}^{N}\delta_h(X_i)\right] \leq \frac{\mu}{N}\sum_{j=1}^{a}\mathbb{E}\left[\sup_{h \in \mathcal{H} \cap rS(L_2)} \frac{1}{\mu}\sum_{i=1}^{\mu}Q_{\mathcal{H}}(2\tau)_i\psi_1(|h-h_v|\,(\tilde{X}_{(a+b)(i-1)+j}))\right]$$

$$+2(\mu-1)\beta(a+b) + \frac{2b\mu}{N} + \frac{\rho}{\tau r}.$$

Since $\psi_1(|\cdot|)$ is a $1/(\tau r)$-Lipschitz continuous mapping, using properties of Rademacher complexity we have

$$\mathbb{E}\left[\sup_{h \in \mathcal{H} \cap rS(L_2)} \frac{1}{\mu}\sum_{i=1}^{\mu}Q_{\mathcal{H}}(2\tau)_i\psi_1(|h-h_v|\,(\tilde{X}_{(a+b)(i-1)+j}))\right]$$

$$\leq \frac{1}{\tau r}\mathbb{E}\left[\sup_{h \in \mathcal{H} \cap rS(L_2)} \frac{1}{\mu}\sum_{i=1}^{\mu}Q_{\mathcal{H}}(2\tau)_i(h-h_v)(\tilde{X}_{(a+b)(i-1)+j}))\right].$$

Since we assumed $r > \omega_2(\zeta_2)$, and using (78) we have,

$$\mathbb{E}\left[\sup_{\delta_h \in \Delta_r} \frac{1}{N}\sum_{i=1}^{N}\delta_h(X_i)\right] \leq \frac{a\zeta_2\mu}{N\tau} + 2(\mu-1)\beta(a+b) + \frac{2b\mu}{N} + \frac{c_5\zeta_1}{\tau\sqrt{Q_{\mathcal{H}}(2\tau)}}.$$

Choosing

$$\zeta_1 \sim 2\tau Q_{\mathcal{H}}(2\tau)^{\frac{3}{2}} \quad \zeta_2 \sim 2\tau Q_{\mathcal{H}}(2\tau) \quad a \sim \frac{(4-Q_{\mathcal{H}}(2\tau))N^{1/(1+\eta_1)}}{c^{\frac{1}{\eta_1}}} \tag{80}$$

$$b \sim \frac{Q_{\mathcal{H}}(2\tau)N^{1/(1+\eta_1)}}{c^{\frac{1}{\eta_1}}} \quad \text{and} \quad \mu \sim \frac{N^{\eta_1/(1+\eta_1)}c^{\frac{1}{\eta_1}}}{4}, \tag{81}$$

we have,

$$\mathbb{E}\left[\sup_{\delta_h \in \Delta_r} \frac{1}{N}\sum_{i=1}^{N}\delta_h(X_i)\right] \leq \frac{Q_{\mathcal{H}}(2\tau)}{32}.$$

Now we use Lemma B.3, with the following choice

$$t = \frac{NQ_{\mathcal{H}}(2\tau)}{32} \quad a = \frac{(4-\frac{Q_{\mathcal{H}}(2\tau)}{16})N^{1/(1+\eta_1)}}{c^{\frac{1}{\eta_1}}Q_{\mathcal{H}}(2\tau)^{1-\frac{1}{\eta_1}}} \quad b = \frac{Q_{\mathcal{H}}(2\tau)^{\frac{1}{\eta_1}}N^{1/(1+\eta_1)}}{32c^{\frac{1}{\eta_1}}} \quad \mu = \frac{N^{\eta_1/(1+\eta_1)}c^{\frac{1}{\eta_1}}Q_{\mathcal{H}}(2\tau)^{\frac{\eta_1-1}{\eta_1}}}{4}.$$

With probability at least $1 - c_6 Q_{\mathcal{H}}(2\tau)^{1-\frac{1}{\eta_1}} N^{\eta_1/(1+\eta_1)} e^{-c_7 Q_{\mathcal{H}}(2\tau) N^{\eta_1/(1+\eta_1)}}$ we have,

$$\frac{1}{N} \sum_{i=1}^{N} \sup_{\delta_h \in \Delta_r} \delta_h(X_i) \leq \mathbb{E}\left[ \frac{1}{N} \sum_{i=1}^{N} \sup_{\delta_h \in \Delta_r} \delta_h(X_i) \right] + \frac{t}{N} \leq \frac{Q_{\mathcal{H}}(2\tau)}{16}.$$

Then $\forall h \in \mathcal{H} \cap rS(L_2)$,

$$|\{i : |h - h_v|(X_i) \leq \tau r\}| \geq \left( 1 - \frac{Q_{\mathcal{H}}(2\tau)}{16} \right) N. \tag{82}$$

Recall that $h_v \in V_r$, and $|S_{h_v}| \geq N Q_{\mathcal{H}}(2\tau)/2$. Let

$$K_h = \{k : |h - h_v|(X_k) \leq \tau r\} \cap S_{h_v}. \tag{83}$$

Then $|K_h| \geq 3N Q_{\mathcal{H}}(2\tau)/8 \geq N Q_{\mathcal{H}}(2\tau)/4$. Also, $\forall k \in K_h$,

$$|h(X_k)| \geq |h_v(X_k)| - |(h - h_v)(X_k)| \geq 2\tau r - \tau r = \tau r. \tag{84}$$

This also implies that $h(X_k)$ and $h_v(X_k)$ have same signs. Similarly, using (77) we get

$$|h(X_k)| \leq |h_v(X_k)| + |(h - h_v)(X_k)| \leq c_9 (2\tau + \frac{1}{\sqrt{Q_{\mathcal{H}}(2\tau)}}) \|h\|_{L_2}. \tag{85}$$

Combining (84) and (85) we have (76). This also implies that $h(X_k)$ and $v(X_k)$ have the same sign. $\blacksquare$

**Lemma B.8** ([Men18, Lemma 4.8]). *Let $1 \leq k \leq m/40$ and set $\mathscr{D} \subset \{-1, 0, 1\}^m$ of cardinality at most $\exp(k)$. For every $d = (d(i))_{i=1}^m \in \mathscr{D}$ put $S_d = \{i : d(i) \neq 0\}$ and assume that $|S_d| \geq 40k$. If $\{\epsilon_i\}_{i=1}^m$ are independent, symmetric $\{-1, 1\}$-valued random variables, then with probability at least $1 - 2\exp(-k)$,*

$$\inf_{d \in \mathscr{D}} |\{i \in S_d : sgn(d(i)) = \epsilon_i\}| \geq k/3.$$

**Lemma B.9.** *Conditioned on the event $\mathcal{A}$ as mentioned in Lemma B.7, with probability at least $1 - 2\exp(-c_2 Q_{\mathcal{H}}(2\tau) N)$ we have: for every $h \in \mathcal{H}_{f^*} \coloneqq \mathcal{F} - f^*$ with $\|h\|_{L_2} \geq r$, there is a subset $\mathcal{S}_{1,h} \subset \{1, 2, \cdots, N\}$ such that $|\mathcal{S}_{1,h}| \geq Q_{\mathcal{H}}(2\tau) N/24$. and for every $i \in \mathcal{S}_{1,h}$,*

$$\tau \|h\|_{L_2} \leq |h(X_i)| \leq c_9 \left( 2\tau + \frac{1}{\sqrt{Q_{\mathcal{H}}(2\tau)}} \right) \|h\|_{L_2}, \qquad sgn(h(X_i)) = \epsilon_i, \tag{86}$$

*where $\{\epsilon_i\}_{i=1}^N$ are independent, symmetric $\{-1, 1\}$-valued random variables.*

*Proof.* For a $h \in \mathcal{H}$, let $\|h\|_{L_2} = r$ and let $h_v$ be as in Lemma B.7. Recall from (76), that there is a subset $K_h \subset S_{h_v}$ containing at least $3/4$ of the coordinates of $S_{h_v}$ for which,

$$\tau r \leq |h(X_j)| \leq c_9 \left( 2\tau + \frac{1}{\sqrt{Q_{\mathcal{H}}(2\tau)}} \right) r,$$

and $h(X_j)$ and $h_v(X_j)$ have the same sign. Define

$$d_{h_v} = \{sgn(h_v(X_i)) \mathbb{1}_{S_{h_v}}(X_i)\}_{i=1}^N, \qquad \mathcal{D} = \{d_{h_v} : h_v \in V_r\}.$$

Using Lemma B.8, on the set $\mathcal{D} = \{d_{h_v} : d_{h_v} \in V_r\}$ for $k = N Q_{\mathcal{H}}(2\tau)/1000$, and observing that every $d_{h_v} \in \mathcal{D}$, $|\{i : d_{h_v}(i) \neq 0\}| \geq N Q_{\mathcal{H}}(2\tau)/2 \geq 40k$ (recall that $|S_{h_v}| \geq N Q_{\mathcal{H}}(2\tau)/2$), we get with probability at least $1 - 2\exp(-c_2 Q_{\mathcal{H}}(2\tau) N)$, for every $h_v \in V_r$, $d_{h_v}(i) = \epsilon_i$ on at least $1/3$ of the coordinates of $S_{h_v}$. Then it follows that on at least $1/12$ of the coordinates of $S_{h_v}$, $h(X_j) = \epsilon_j$. Since $\mathcal{H}_{f^*}$ is assumed to be star-shaped the same result holds when $\|h\|_{L_2} \geq r$. $\blacksquare$

**Proposition B.1.** *With probability at least $1 - c_9 Q_{\mathcal{H}}(2\tau)^{1-\frac{1}{\eta_1}} N^{\eta_1/(1+\eta_1)} e^{-c_{10} Q_{\mathcal{H}}(2\tau) N^{\eta_1/(1+\eta_1)}}$, for every $f \in \mathcal{F}$ which satisfies $\|f - f^*\|_{L_2} \geq 2\omega_Q$ we have*

$$\frac{1}{N} \sum_{i=1}^{N} \ell''(\widetilde{\xi}_i)(f - f^*)^2(X_i) \geq c_{16} Q_{\mathcal{H}}(2\tau) \rho(0, t_0) \tau^2 \|f - f^*\|_{L_2}^2. \tag{87}$$

*where $t_0 = c_{11}(2\tau + 1/\sqrt{Q_{\mathcal{H}}(2\tau)}) (\|\xi\|_{L_2} + \|f - f^*\|_{L_2})$.*

*Proof.* [Proof of Proposition B.1] Recall the decomposition of $P_N L_f$ (1). For every $(X, Y)$ the midpoint $\tilde{\xi}$ belongs to the interval with end points $-\xi$ and $(f - f^*)(X) - \xi$ where $f \in \mathcal{F}$. So,

$$|\tilde{\xi}_i| \leq |\xi_i| + |(f - f^*)(X_i)|.$$

Let $\|f - f^*\|_{L_2} > 2\omega_Q$. Now from Lemma B.9, with probability at least

$$1 - c_9 Q_{\mathcal{H}}(2\tau)^{1 - \frac{1}{\eta_1}} N^{\eta_1/(1+\eta_1)} e^{-c_{10} Q_{\mathcal{H}}(2\tau) N^{\eta_1/(1+\eta_1)}},$$

we have a subset $\mathcal{S}_{1,h} \subset \{1, 2, \cdots, N\}$ such that $|\mathcal{S}_{1,h}| \geq Q_{\mathcal{H}}(2\tau) N/24$, and for every $i \in \mathcal{S}_{1,h}$,

$$|(f - f^*)(X_i)| \leq c_9 (2\tau + 1/\sqrt{Q_{\mathcal{H}}(2\tau)}) \|f - f^*\|_{L_2}.$$

Using Markov's inequality,

$$\mathbb{P}(|\xi_i| > 10\|\xi\|_{L_2}/\sqrt{Q_{\mathcal{H}}(2\tau)}) \leq \frac{Q_{\mathcal{H}}(2\tau).}{100}$$

Now taking $U_i = \mathbb{1}\left(|\xi_i| \leq \frac{c_9 \|\xi\|_{L_2}}{\sqrt{Q_{\mathcal{H}}(2\tau)}}\right)$, and using Lemma B.3, and choosing parameters as in (73) we get, with probability at least $1 - c_1 Q_{\mathcal{H}}(2\tau)^{1 - \frac{1}{\eta_1}} N^{\eta_1/(1+\eta_1)} e^{-c_2 Q_{\mathcal{H}}(2\tau) N^{\eta_1/(1+\eta_1)}}$,

$$\left| \left\{ i : |\xi_i| \leq \frac{c_9 \|\xi\|_{L_2}}{\sqrt{Q_{\mathcal{H}}(2\tau)}} \right\} \right| \geq N(1 - Q_{\mathcal{H}}(2\tau)/50).$$

This implies that with probability at least $1 - c_{16} Q_{\mathcal{H}}(2\tau)^{1 - \frac{1}{\eta_1}} N^{\eta_1/(1+\eta_1)} e^{-c_{17} Q_{\mathcal{H}}(2\tau) N^{\eta_1/(1+\eta_1)}}$ we have,

$$|\tilde{\xi}_i| \leq c_{11} (2\tau + 1/\sqrt{Q_{\mathcal{H}}(2\tau)}) \left( \|\xi\|_{L_2} + \|f - f^*\|_{L_2} \right).$$

Set $t_0 = c_{11}(2\tau + 1/\sqrt{Q_{\mathcal{H}}(2\tau)})(\|\xi\|_{L_2} + \|f - f^*\|_{L_2})$. Using Lemma B.9, with probability at least $1 - c_9 Q_{\mathcal{H}}(2\tau)^{1 - \frac{1}{\eta_1}} N^{\eta_1/(1+\eta_1)} e^{-c_{10} Q_{\mathcal{H}}(2\tau) N^{\eta_1/(1+\eta_1)}}$,

$$\frac{1}{N} \sum_{i=1}^{N} \ell''(\tilde{\xi}_i)(f - f^*)^2(X_i) \geq c_{16} Q_{\mathcal{H}}(2\tau) \rho(0, t_0) \tau^2 \|f - f^*\|_{L_2}^2. \tag{88}$$

∎

Using Proposition B.1, and proving the two-sided bounds for the second term on the RHS of (71) in (93) and (94), we have Proposition B.2.

**Proposition B.2.** *Consider ERM with loss functions that satisfy Assumption 3.2. For $\tau_0 < c_2 Q_{\mathcal{F} - \mathcal{F}}(2\tau) \rho(0, t_0) \tau^2$, $t_0 = \mathcal{O}((2\tau + 1/\sqrt{Q_{\mathcal{H}}(2\tau)})(\|\xi\|_{L_2} + \|f - f^*\|_{L_2}))$, setting $\mu = N^{\eta_1/(1+\eta_1)}$, for some constants $c, c' > 0$, we have, for any $N \geq 4$, the following:*

1. *Under conditions (a), (b), (c)-(i), and (d) of Assumption 2.1, for $0 < \iota < \frac{1}{4}$,*

$$\|\hat{f} - f^*\|_{L_2} \leq \max\left\{ N^{-\frac{1}{4} + \iota}, 2\omega_Q(\mathcal{F} - \mathcal{F}, N, Q_{\mathcal{H}}(2\tau)^{3/2}, Q_{\mathcal{H}}(2\tau)) \right\}, \tag{89}$$

   *with probability at least (for $V$ is defined in (8) and some positive $c_9, c_{10}, \widetilde{C}_3$)*

$$1 - c_9 Q_{\mathcal{H}}(2\tau)^{1 - \frac{1}{\eta_1}} N^{\eta_1/(1+\eta_1)} e^{-c_{10} Q_{\mathcal{H}}(2\tau)^{1 + \frac{1}{\eta_1}} N^{\frac{\eta_1}{1+\eta_1}}} - \widetilde{C}_3 N \exp\left( -(N^{\frac{1}{2} + 2\iota} \tau_0)^\eta / C_1 \right).$$

2. *Under conditions (a), (b), and (c)-(ii) of Assumption 2.1, for $0 < \iota < (1 - 1/\eta_2)/4$,*

$$\|\hat{f} - f^*\|_{L_2} \leq \max\left\{ N^{-\frac{(1 - 1/\eta_2)}{4} + \iota}, 2\omega_Q(\mathcal{F} - \mathcal{F}, N, Q_{\mathcal{H}}(2\tau)^{3/2}, Q_{\mathcal{H}}(2\tau)) \right\}, \tag{90}$$

   *with probability at least (for constants $c_9, c_{10}, \widetilde{C}_4 > 0$)*

$$1 - c_9 Q_{\mathcal{H}}(2\tau)^{1 - \frac{1}{\eta_1}} N^{\eta_1/(1+\eta_1)} e^{-c_{10} Q_{\mathcal{H}}(2\tau)^{1 + \frac{1}{\eta_1}} N^{\eta_1/(1+\eta_1)}} - \widetilde{C}_4 \tau_0^{-\frac{2\eta_2}{1+\eta_2}} N^{-\frac{4\iota\eta_2}{1+\eta_2}}. \tag{91}$$

*Proof.* [Proof of Proposition B.2] We first prove **part 1**. We will denote the class $\mathcal{F} - f^*$ by $\mathcal{H}$. From Proposition B.1 it follows that for every $f \in \mathcal{F}$ which satisfies $\|f - f^*\|_{L_2} \geq 2\omega_Q$ with probability at least

$$\mathscr{P}_{1,c} = 1 - c_9 Q_{\mathcal{H}}(2\tau)^{1 - \frac{1}{\eta_1}} N^{\eta_1/(1+\eta_1)} e^{-c_{10} Q_{\mathcal{H}}(2\tau)^{1 + \frac{1}{\eta_1}} N^{\eta_1/(1+\eta_1)}},$$

we have

$$\frac{1}{N} \sum_{i=1}^{N} \ell''(\widetilde{\xi}_i)(f - f^*)^2(X_i) \geq c_{16} Q_{\mathcal{H}}(2\tau)\rho(0,t_2)\tau^2 \|f - f^*\|_{L_2}^2.$$

So, with probability at least $\mathscr{P}_{1,c}$, for every $f \in \mathcal{F}$ that satisfies $\|f - f^*\|_{L_2} \geq 2\omega_Q$,

$$P_N \mathcal{L}_f \geq \left( \frac{1}{16N} \sum_{i=1}^{N} l'(\xi_i)(f - f^*)(X_i) - \mathbb{E}\left[ l'(\xi)(f - f^*) \right] \right) + c_{16} Q_{\mathcal{H}}(2\tau)\rho(0,t_2)\tau^2 \|f - f^*\|_{L_2}^2.$$
(92)

When $\|f - f^*\|_{L_2} \geq \mathscr{A}(N) > 2(N\tau_0)^{-1/2}$, we have $\log(N\tau_0 \|f - f^*\|_{L_2}^2) \leq 2(N\tau_0 \|f - f^*\|_{L_2}^2)^{(1-\eta)/2}/(1-\eta)$. Under Conditions (1), (3), and (4) of Assumption 2.1, using Lemma 2.1, we get

$$\mathbb{P}\left( \left| \frac{1}{N} \sum_{i=1}^{N} l'(\xi_i)(f - f^*)(X_i) - \mathbb{E}\left[ l'(\xi)(f - f^*) \right] \right| \geq \tau_0 \|f - f^*\|_{L_2}^2 \right)$$

$$\leq N \exp\left( -\frac{(N\tau_0 \|f - f^*\|_{L_2}^2)^{\eta}}{C_1} \right) + \exp\left( -\frac{N^2 \tau_0^2 \|f - f^*\|_{L_2}^4}{C_2(1 + NV)} \right)$$

$$+ \exp\left( -\frac{N\tau_0^2 \|f - f^*\|_{L_2}^4}{C_3} \exp\left( \frac{(N\tau_0 \|f - f^*\|_{L_2}^2)^{\eta(1-\eta)}}{C_4(\log(N\tau_0 \|f - f^*\|_{L_2}^2))^{\eta}} \right) \right)$$

$$\leq N \exp\left( -\frac{(N\tau_0 \|f - f^*\|_{L_2}^2)^{\eta}}{C_1} \right) + \exp\left( -\frac{N^2 \tau_0^2 \|f - f^*\|_{L_2}^4}{C_2(1 + NV)} \right)$$

$$+ \exp\left( -\frac{N\tau_0^2 \|f - f^*\|_{L_2}^4}{C_3} \exp\left( \frac{(1-\eta)^{\eta}(N\tau_0 \|f - f^*\|_{L_2}^2)^{\frac{\eta(1-\eta)}{2}}}{C_4 2^{\eta}} \right) \right)$$

$$\leq N \exp\left( -\frac{(N\tau_0 \mathscr{A}(N)^2)^{\eta}}{C_1} \right) + \exp\left( -\frac{N^2 \tau_0^2 \mathscr{A}(N)^4}{C_2(1 + NV)} \right)$$

$$+ \exp\left( -\frac{N\tau_0^2 \mathscr{A}(N)^4}{C_3} \exp\left( \frac{(1-\eta)^{\eta}(N\tau_0 \mathscr{A}(N)^2)^{\frac{\eta(1-\eta)}{2}}}{C_4 2^{\eta}} \right) \right) \equiv \mathscr{P}_{2,c},$$
(93)

where

$$V \leq \mathbb{E}\left[ (\ell'(\xi_1)(f - f^*)(X_1))^2 \right] + 4 \sum_{i \geq 0} \mathbb{E}\left[ B_i (\ell'(\xi_1)(f - f^*)(X_1))^2 \right],$$

$\{B_i\}$ is some sequence such that $B_i \in [0,1]$ and $\mathbb{E}[B_i] \leq \beta(i)$, and $C_1, C_2, C_3$ are constants which depend on $c, \eta, \eta_1, \eta_2$. Observe that,

$$V \leq \mathbb{E}\left[ (\ell'(\xi_1)(f - f^*)(X_1))^2 \right] + 4 \sum_{i \geq 0} \mathbb{E}\left[ B_i (\ell'(\xi_1)(f - f^*)(X_1))^2 \right]$$

$$\leq \mathbb{E}\left[ (\ell'(\xi_1)(f - f^*)(X_1))^2 \right] + 4 \sum_{i \geq 0} \sqrt{\mathbb{E}[B_i^2] \mathbb{E}\left[ (\ell'(\xi_1)(f - f^*)(X_1))^4 \right]}$$

$$\leq \mathbb{E}\left[ (\ell'(\xi_1)(f - f^*)(X_1))^2 \right] + 4 \sqrt{\mathbb{E}\left[ (\ell'(\xi_1)(f - f^*)(X_1))^4 \right]} \sum_{i \geq 0} \sqrt{\mathbb{E}[B_i]}$$

$$\leq \mathbb{E}\left[ (\ell'(\xi_1)(f - f^*)(X_1))^2 \right] + 4 \sqrt{\mathbb{E}\left[ (\ell'(\xi_1)(f - f^*)(X_1))^4 \right]} \sum_{i \geq 0} \sqrt{\beta(i)}$$

$$\leq \mathbb{E}\left[(\ell'(\xi_1)(f-f^*)(X_1))^2\right] + 4\sqrt{\mathbb{E}\left[(\ell'(\xi_1)(f-f^*)(X_1))^4\right]}\sum_{i\geq 0}\exp(-ci^{\eta_1}/2)$$

$$\leq 2^{\frac{2}{\eta_2}} + C4^{1+\frac{2}{\eta_2}}.$$

Combining (92), and (93), with probability at least $\mathscr{P}_{1,c} - \mathscr{P}_{2,c}$, for every $f \in \mathcal{F}$ that satisfies $\|f-f^*\|_{L_2} \geq \max(2\omega_Q, \mathscr{A}(N))$, we get

$$P_N L_f \geq -2\tau_0\|f-f^*\|_{L_2}^2 + c_{16}Q_{\mathcal{H}}(2\tau)\rho(0,t_2)\tau^2\|f-f^*\|_{L_2}^2.$$

Choosing $\tau_0 < c_{16}Q_{\mathcal{H}}(2\tau)\rho(0,t_2)\tau^2/4$, we have, $P_N L_f > 0$. But the empirical minimizer $\hat{f}$ satisfies $P_N L_{\hat{f}} \leq 0$. This implies, together with choosing $\mathscr{A}(N) = N^{-1/4+\iota}$, that with probability at least $\mathscr{P} = \mathscr{P}_{1,c} - \mathscr{P}_{2,c}$,

$$\|\hat{f}-f^*\|_{L_2} \leq \max(2\omega_Q(\mathcal{F}-\mathcal{F},N,Q_{\mathcal{H}}(2\tau)^{\frac{3}{2}},Q_{\mathcal{H}}(2\tau)),\mathscr{A}(N)),$$

where

$$\mathscr{P} = 1 - c_9 Q_{\mathcal{H}}(2\tau)^{1-\frac{1}{\eta_1}}N^{\eta_1/(1+\eta_1)}e^{-c_{10}Q_{\mathcal{H}}(2\tau)^{1+\frac{1}{\eta_1}}N^{\eta_1/(1+\eta_1)}}$$

$$-N\exp\left(-\frac{(N^{\frac{1}{2}+2\iota}\tau_0)^\eta}{C_1}\right) - \exp\left(-\frac{N^{1+4\iota}\tau_0^2}{C_2(1+NV)}\right) - \exp\left(-\frac{N^{4\iota}\tau_0^2}{C_3}\exp\left((1-\eta)^\eta\frac{(N^{\frac{1}{2}+2\iota}\tau_0)^{\frac{\eta(1-\eta)}{2}}}{C_4 2^\eta}\right)\right).$$

We now prove **part 2**. When $\|f-f^*\|_{L_2} \geq \mathscr{A}(N)$, using Lemma 2.2 we have

$$\mathbb{P}\left(\left|\frac{1}{N}\sum_{i=1}^N l'(\xi_i)(f-f^*)(X_i) - \mathbb{E}\left[l'(\xi)(f-f^*)\right]\right| \geq \tau_0\|f-f^*\|_{L_2}^2\right)$$

$$\leq \frac{2^{\eta_2+3}}{(d_2\log N\tau_0\|f-f^*\|_{L_2}^2)^{\frac{1-\eta_2}{\eta_1}}}N(N\tau_0\|f-f^*\|_{L_2}^2)^{-(1+d_1(\eta_2-1))} + 8N(N\tau_0\|f-f^*\|_{L_2}^2)^{-(1+d_2c')}$$

$$+ 2e^{-\frac{(N\tau_0\|f-f^*\|_{L_2}^2)^{2-2d_1}(d_2\log(N\tau_0\|f-f^*\|_{L_2}^2))^{1/\eta_1}}{9N}}$$

$$\leq \frac{2^{\eta_2+3}}{(d_2\log(N\tau_0\mathscr{A}(N)^2))^{\frac{1-\eta_2}{\eta_1}}}N(N\tau_0\mathscr{A}(N)^2)^{-(1+d_1(\eta_2-1))} + 8N(N\tau_0\mathscr{A}(N)^2)^{-(1+d_2c')}$$

$$+ 2e^{-\frac{(N\tau_0\mathscr{A}(N)^2)^{2-2d_1}(d_2\log(N\tau_0\mathscr{A}(N)^2))^{1/\eta_1}}{9N}} \equiv \mathscr{P}_{2,c}. \tag{94}$$

We will choose $d_1$ suitably to allow $\mathscr{A}(N)$ to decrease with $N$ as fast as possible while ensuring $\lim_{N\to\infty}\mathscr{P}_{2,c} \to 0$. Combining (66), and (69), with probability at least $\mathscr{P}_{1,c} - \mathscr{P}_{2,c}$, for every $f \in \mathcal{F}$ that satisfies $\|f-f^*\|_{L_2} \geq \max(2\omega_Q, \mathscr{A}(N))$, we get

$$P_N L_f \geq -2\tau_0\|f-f^*\|_{L_2}^2 + c_{16}Q_{\mathcal{H}}(2\tau)\rho(0,t_2)\tau^2\|f-f^*\|_{L_2}^2.$$

Choosing $\tau_0 < c_{16}Q_{\mathcal{H}}(2\tau)\rho(0,t_2)\tau^2/4$, we have, $P_N L_f > 0$. But the empirical minimizer $\hat{f}$ satisfies $P_N L_{\hat{f}} \leq 0$. This implies, together with choosing $\mathscr{A}(N) = N^{-(1-1/\eta_2)/4+\iota}$, $d_1 = 1/(1+\eta_2)$, $d_2 = (\eta_2-1)/(\eta_2+1)$, and $\iota < (1-1/\eta_2)/4$, that with probability at least $\mathscr{P}_c = \mathscr{P}_{1,c} - \mathscr{P}_{2,c}$,

$$\|\hat{f}-f^*\|_{L_2} \leq \max(2\omega_Q(\mathcal{F}-\mathcal{F},N,\zeta_1,\zeta_2),N^{-(1-1/\eta_2)/4+\iota}),$$

where

$$\mathscr{P}_c = 1 - c_9 Q_{\mathcal{H}}(2\tau)^{1-\frac{1}{\eta_1}}N^{\eta_1/(1+\eta_1)}e^{-c_{10}Q_{\mathcal{H}}(2\tau)^{1+\frac{1}{\eta_1}}N^{\eta_1/(1+\eta_1)}} - \frac{2^{\eta_2+3}\tau_0^{-\frac{2\eta_2}{1+\eta_2}}}{\left(\log\left(\tau_0 N^{\frac{1}{2}+\frac{1}{2\eta_2}+2\iota}\right)/2\right)^{\frac{1-\eta_2}{\eta_1}}}N^{-\frac{4\iota\eta_2}{1+\eta_2}}$$

$$-8\tau_0^{-\frac{2\eta_2}{1+\eta_2}}N^{-\frac{4\iota\eta_2}{1+\eta_2}} - 2e^{-\frac{\tau_0^{\frac{2\eta_2}{1+\eta_2}}\left(\log\left(\tau_0 N^{\frac{1}{2}+\frac{1}{2\eta_2}+2\iota}\right)/2\right)^{1/\eta_1}}{9}N^{\frac{4\iota\eta_2}{1+\eta_2}}}.$$

$\blacksquare$

Note that Proposition B.2 is exactly same as Theorem 3.2 except for the fact one needs $\ell$ to be strongly convex in $[-t_0, t_0]$ instead of $[-t_2, t_2]$ where $t_2$ is of the order $\mathcal{O}((2\tau + 1/\sqrt{Q_{\mathcal{H}}(2\tau)})\|\xi\|_{L_2})$. So now we will show that empirical minimizer $\hat{f} \in \mathcal{F}$ satisfies $\|\hat{f} - f^*\|_{L_2} \leq \max(\|\xi\|_{L_2}, 2\omega_Q)$ with high probability. One has the following result from [Men18]:

$$\{h - f^* : h \in \mathcal{F}, \|h - f^*\|_{L_2} \geq R\} \subset \{\lambda(f - f^*) : \lambda \geq 1, f \in \mathcal{F}, \|f - f^*\|_{L_2} = R\}. \quad (95)$$

**Lemma B.10** ([Men18, Lemma 5.6]). *When (87) is true, if* $\|f - f^*\|_{L_2} \geq \max(\|\xi\|_{L_2}, 2\omega_Q)$, *and* $\lambda \geq 1$, *then*

$$\frac{1}{N}\sum_{i=1}^{N}\ell''(\widetilde{\xi_i})(\lambda(f - f^*))^2(X_i) \geq \lfloor\lambda\rfloor c_{16}Q_{\mathcal{H}}(2\tau)\rho(0, t_0)\tau^2 \max\left(\|\xi\|_{L_2}^2, 4\omega_Q^2\right). \quad (96)$$

**Lemma B.11.** *With probability at least* $1 - \mathscr{P}_{2,c}$ *with* $\tau_0 = c_{16}Q_{\mathcal{H}}(2\tau)\rho(0, t_0)\tau^2/4$, *we have*

$$\|\hat{f} - f^*\|_{L_2} \leq \max(\|\xi\|_{L_2}, 2\omega_Q).$$

*Proof.* From (93), with probability at least $1 - \mathscr{P}_{2,c}$ we have,

$$\left|\frac{1}{N}\sum_{i=1}^{N}l'(\xi_i)(f - f^*)(X_i) - \mathbb{E}\left[l'(\xi)(f - f^*)\right]\right| \leq \tau_0\|f - f^*\|_{L_2}^2.$$

To make the dependency of $\mathscr{P}_{2,c}$ on $\tau_0$ explicit, we use the notation $\mathscr{P}_{2,c,\tau_0}$ to denote $\mathscr{P}_{2,c}$ for this proof. If $\|f - f^*\|_{L_2} \leq \max(\|\xi\|_{L_2}, 2\omega_Q)$, choosing $\tau_0 = c_{16}Q_{\mathcal{H}}(2\tau)\rho(0, t_0)\tau^2/4$, with probability at least $1 - \mathscr{P}_{2,c,c_{16}Q_{\mathcal{H}}(2\tau)\rho(0,t_0)\tau^2/4}$, for the same $\lambda \geq 1$ as in Lemma B.10, we have,

$$\left|\frac{1}{N}\sum_{i=1}^{N}l'(\xi_i)(\lambda(f - f^*))(X_i) - \mathbb{E}\left[l'(\xi)(\lambda(f - f^*))\right]\right| \leq \frac{c_{16}\lambda Q_{\mathcal{H}}(2\tau)\rho(0, t_0)\tau^2}{4}\max(\|\xi\|_{L_2}, 2\omega_Q)^2.$$

If $\|f - f^*\|_{L_2} = \max(\|\xi\|_{L_2}, 2\omega_Q)$, and $\lambda \geq 1$, we also have

$$\frac{1}{N}\sum_{i=1}^{N}\ell''(\widetilde{\xi_i})(\lambda(f - f^*))^2(X_i) - \left|\frac{1}{N}\sum_{i=1}^{N}l'(\xi_i)(\lambda(f - f^*))(X_i) - \mathbb{E}\left[l'(\xi)(\lambda(f - f^*))\right]\right|$$

$$\geq \lfloor\lambda\rfloor c_{16}Q_{\mathcal{H}}(2\tau)\rho(0, t_0)\tau^2 \max\left(\|\xi\|_{L_2}^2, 4\omega_Q^2\right) - \frac{c_{16}\lambda Q_{\mathcal{H}}(2\tau)\rho(0, t_0)\tau^2}{4}\max(\|\xi\|_{L_2}, 2\omega_Q)^2$$

$$> 0.$$

So by (95), and Lemma B.10, the empirical minimizer $\hat{f}$ satisfies,

$$\|\hat{f} - f^*\|_{L_2} \leq \max(\|\xi\|_{L_2}, 2\omega_Q).$$

∎

*Proof.* [Proof of Theorem 3.2] Combining Lemma B.11 with the two parts of Proposition B.2 gives us Theorem 3.2. ∎

**Corollary B.2.** *For the convex ERM procedure, under Assumptions 2.1, with condition (b) replaced by Assumption 3.1 with $p = 8$, for some $0 < \iota < \frac{1}{2}$ and $r, \mu$ and $\tau_0$ same as in Theorem 3.2, for sufficiently large $N$, we have*

$$\|\hat{f} - f^*\|_{L_2} \leq \max\left(N^{-\frac{1}{2}+\iota}, 2\omega_Q(\mathcal{F} - \mathcal{F}, N, Q_{\mathcal{H}}(2\tau)^{\frac{3}{2}}, Q_{\mathcal{H}}(2\tau))\right) \quad (97)$$

*with probability at least (for some constants $c_9, c_{10}, \tilde{C}_2 > 0$)*

$$1 - c_9 Q_{\mathcal{H}}(2\tau)^{1 - \frac{1}{\eta_1}}N^{\eta_1/(1+\eta_1)}e^{-c_{10}Q_{\mathcal{H}}(2\tau)^{1+\frac{1}{\eta_1}}N^{\eta_1/(1+\eta_1)}} - \tilde{C}_2 N\exp\left(-(N^{2\iota}\tau_0)^\eta/M_1\right).$$

*Proof.* [Proof of Corollary B.2] The proof is same as Corollary 3.1 and hence we omit it here. ∎

# C Details of Section 4.1

## C.0.1 Verification of Assumption 2.1 for Example 4.1

[WZLL20] showed that the time series given by (19) is stable, strict sense stationary, with $X_i \sim \mathrm{sw}(\eta_X)$, for some $1 > \eta_X > 0$. As shown in [WLT20], $\{(X_i, Y_i)\}$ is a strictly stationary sequence; thus, we obtain that is also a $\beta$-mixing sequence with exponentially decaying coefficients as in condition (a) of Assumption 2.1. Now we verify the small-ball condition (b) of Assumption 2.1. Let, for any $\theta = (\theta_1, \cdots, \theta_d) \neq 0$, $d_1$ denote the set of non-zero coordinates of $\theta$. W.l.o.g lets assume $T = 1, 2, \cdots, d_1$. Let $\mathbb{E}\left[X_i^2\right] = \sigma_i^2$ and $\sigma_0 = \min_{1 \leq i \leq d_1} \sigma_i$. Then $\mathbb{E}\left[\left(\theta^\top X\right)^2\right] = \sum_{i=1}^{d_1} \theta_i^2 \sigma_i^2 \geq \sigma_0^2 \|\theta\|_2^2$. Since $X_i \sim SW(\eta_X)$, we have $\|\theta_i X_i\|_8 \leq K_1 |\theta_i| 8^{\eta_X}$. So,

$$\|\theta^\top X\|_8 \leq K_1 \|\theta\|_1 8^{\eta_X} \leq \frac{K_1 \|\theta\|_1 8^{\eta_X}}{\sigma_0 \|\theta\|_2} \|\theta^\top X\|_2 \leq \frac{K_1 \sqrt{d_1} 8^{\eta_X}}{\sigma_0} \|\theta^\top X\|_2. \tag{98}$$

Then using Lemma 4.1 of [Men15], for any $0 < u < 1$, we have $\mathbb{P}\left(\left|\theta^\top X\right| \geq u \|\theta^\top X\|_{L_2}\right) \geq \left((1 - u^2)/(K_1^2 8^{2/\eta_X + 1})\right)^{4/3}$. So condition (b) of Assumption 2.1 is true here. This also implies that Assumption 3.1 is true in this case for $p = 8$. As an immediate consequence of [VGNA20, Proposition 2.3], we have that condition (c)-(i) in Assumption 2.1 is valid here with $\eta_2 = \max(\eta_X, \eta_\xi) < 1$. Since $1/\eta_2 > 1$, condition (d) holds true.

**Proposition C.1.** *Consider the learning problem described above. Then with probability at least*

$$1 - \widetilde{C}_1 N^r Q_{\mathcal{H}}(2\tau) c^{\frac{1}{\eta_1}} \exp(-N^{(1-r)\eta_1}) - \widetilde{C}_2 N \exp\left(-(N^{2\iota}\tau_0)^\eta / M_1\right),$$

*we have*

$$\|\hat{f} - f^*\|_{L_2} \leq \max\left\{\frac{2c_3 R \log(ed)^{\frac{1}{\eta}}}{\sqrt{Q_{\mathcal{H}}(2\tau)} c^{\frac{1}{2\eta_1}}} N^{-\frac{1}{2}+\iota}, N^{-\frac{1}{2}+\iota}\right\}.$$

The proof of Proposition C.1 could be found in the Appendix C.0.1.

**Remark 6.** *In a related setting (i.e., assuming $\theta^*$ is exactly $s$-sparse) [WLT20, Corollary 9] presents parameter estimation error which is of the same order as $\|\hat{f} - f^*\|_{L_2}$ (indeed, for simplicity $R$ could be thought of being at the same order as $s$) since we assume $X$ has finite variance. So with slightly better probability guarantee, we recover the same rate ($\iota$ can be arbitrarily close to 0) as [WLT20, Corollary 9] in the above proposition.*

**Lemma C.1** (Lemma 6.4 of [Men15]). *If $W = (w_i)_{i=1}^d$ is a random vector on $\mathbb{R}^d$, then for every integer $1 \leq k \leq d$,*

$$\mathbb{E}\left[\sup_{t \in \sqrt{k} B_1^d \cap B_2^d} \langle W, t \rangle\right] \leq 2\mathbb{E}\left[\left(\sum_{i=1}^k w_i^{*2}\right)^{\frac{1}{2}}\right],$$

*where $(w_i^*)_{i=1}^d$ is a monotone non-increasing reaarangement of $(|w_i|)_{i=1}^d$.*

**Lemma C.2.** *Let $w_1, w_2, \cdots, w_d$ are independent copies of a mean-zero, variance 1 random variable $w \sim \mathrm{sw}(\eta)$. Then for all $p \geq 1 \wedge \eta$, $\|w\|_{L_p} \leq K_1 p^{\frac{1}{\eta}}$ for some constant $K_1 > 0$. Then for every $1 \leq k \leq d$,*

$$\mathbb{E}\left[\left(\sum_{i=1}^k w_i^{*2}\right)^{\frac{1}{2}}\right] \leq \sqrt{2k} K_1 \left(\log(ed)\right)^{1/\eta}.$$

*Proof.* [Proof of Lemma C.2] For $1 \leq j \leq d$, and $p \geq 2$,

$$\mathbb{P}(w_j^* \geq t) \leq \binom{d}{j} \mathbb{P}^j(|w| > t) \leq \binom{d}{j} \left(\frac{\|z\|_{L_p}}{t}\right)^{jp}.$$

Setting $t = u K_1 \left(\log(ed/j)\right)^{1/\eta}$ and $p = \log(ed/j)$, we get

$$\mathbb{P}\left(w_j^* \geq u K_3\right) \leq \left(\frac{1}{u}\right)^{j \log(ed/j)}, \tag{99}$$

where $K_3 = K_1 \left( \log(ed/j) \right)^{1/\eta}$. Using (99) we will bound $\mathbb{E}\left[ w_j^{*2} \right]$. For some $v$,

$$
\begin{aligned}
\mathbb{E}\left[ w_j^{*2} \right] &= \int_0^\infty \mathbb{P}(w_j^{*2} > u) du \\
&= \int_0^v \mathbb{P}(w_j^{*2} > u) du + \int_v^\infty \mathbb{P}(w_j^{*2} > u) du \\
&\leq v + \int_0^\infty \mathbb{P}(w_j^{*2} > u + v) du \\
&\leq v + \int_0^\infty \left( \frac{K_3}{\sqrt{u+v}} \right)^{j \log(ed/j)} du \\
&= v - K_5 \left[ \frac{(u+v)^{1 - j \log(ed/j)/2}}{j \log(ed/j)/2 - 1} \right]_0^\infty \qquad [\text{where } K_5 = K_3^{j \log(ed/j)}] \\
&= v + K_5 \left[ \frac{v^{1 - j \log(ed/j)/2}}{j \log(ed/j)/2 - 1} \right].
\end{aligned}
$$

To minimize the upper bound on $\mathbb{E}\left[ w_j^{*2} \right]$ we choose

$$
v = K_5^{\frac{2}{j \log(ed/j)}} = K_3^2 = K_1^2 \left( \log(ed/j) \right)^{2/\eta}.
$$

and get

$$
\mathbb{E}\left[ w_j^{*2} \right] \leq 2 K_1^2 \left( \log(ed/j) \right)^{2/\eta}.
$$

For any $1 \leq k \leq d$, using Jensen's inequality,

$$
\mathbb{E}\left[ \left( \sum_{i=1}^k w_i^{*2} \right)^{\frac{1}{2}} \right] \leq \left( \sum_{i=1}^k \mathbb{E}\left[ w_i^{*2} \right] \right)^{\frac{1}{2}} \leq \left( \sum_{i=1}^k 2 K_1^2 \left( \log(ed/i) \right)^{2/\eta} \right)^{\frac{1}{2}} \leq \sqrt{2k} K_1 \left( \log(ed) \right)^{1/\eta}.
$$

$\blacksquare$

*Proof.* [Proof of Proposition C.1] In order to provide a bound on $\|\hat{f} - f^*\|_{L_2}$, we need to compute the order of $\omega_\mu(\mathcal{F}_R - \mathcal{F}_R, \tau Q_{\mathcal{H}_R}(2\tau)/16)$. Based on Lemma C.1 and C.2 it is easy to see that, in a similar way to [Men15],

$$
\mathbb{E}\left[ \sup_{f \in \mathcal{F}_R \cap s\mathcal{D}_{f^*}} \left| \frac{1}{\sqrt{N}} \sum_{i=1}^N \epsilon_i (f - f^*)(X_i) \right| \right] \leq \begin{cases} c_1 K_1 R \left( \log ed \right)^{\frac{1}{\eta}} & (R/s)^2 > d/4, \\ c_2 K_1 s \sqrt{d} & u \leq (R/s)^2 \leq d/4, \end{cases}
$$

where $c_1, c_2$ are constants. Hence, following similar steps as in the proof of [Men15, Lemma 4.6], we have

$$
\omega_\mu(\mathcal{F}_R - \mathcal{F}_R, \tau Q_{\mathcal{H}_R}(2\tau)/16) \leq \begin{cases} \frac{c_3 R}{\sqrt{\mu}} \log (ed)^{\frac{1}{\eta}} & \text{if } \mu \leq c_1 d, \\ 0 & \text{if } \mu > c_1 d. \end{cases} \tag{100}
$$

From (100), choosing $r = 1 - 2\iota$ by Theorem 3.1, for sufficiently large $N$, we have with probability at least

$$
1 - \tilde{C}_1 \frac{N^{1 - 2\iota} Q_{\mathcal{H}}(2\tau) c^{\frac{1}{\eta_1}}}{4} \exp(-N^{2\iota \eta_1}) - \tilde{C}_2 N \exp\left( -\frac{(N^{2\iota} \tau_0)^\eta}{M_1} \right),
$$

we have

$$
\|\hat{f} - f^*\|_{L_2} \leq \max\left( \frac{2 c_3 R \log(ed)^{\frac{1}{\eta}}}{\sqrt{Q_{\mathcal{H}}(2\tau)} c^{\frac{1}{2\eta_1}}} N^{-\frac{1}{2} + \iota}, N^{-\frac{1}{2} + \iota} \right).
$$

$\blacksquare$

# D Proofs of Section 4.2

**Lemma D.1.** *Let $\{X_i'\}_{i=1}^{\mu}$ be an* ***iid*** *sample with independent coordinates $X_{i,j}' \sim L(\eta_3, d_j)$ and let $w$ be a random vector with coordinates*

$$w_j = \frac{1}{\sqrt{\mu}} \sum_{i=1}^{\mu} X_{i,j}' \qquad j = 1, 2, \cdots, d. \tag{101}$$

*Then we have*

$$\mathbb{P}\left(|w_j| \geq t\right) \leq C_3 \left( d_j^{\eta_3 - 2p - 1} \mu^{1 - \frac{\eta_3}{2}} t^{\eta_3 - 2p} + d_j^{-2} t^{-p} \right), \tag{102}$$

*for some constant $C_3 > 0$.*

*Proof.* [Proof of Lemma D.1] Using the symmetry of the distribution of $w_j$ we can write,

$$\mathbb{P}\left(|w_j| \geq t\right) \leq 2\mathbb{P}\left(w_j \geq t\right). \tag{103}$$

Setting $p = \eta_3 - 0.5\iota$, using Theorem 2.1 of [Che07] we get for any $t > 0$

$$\mathbb{P}\left(w_j \geq t\right) \leq C_p t^{-p} \max\left( r_{\mu,p}(t), (r_{\mu,2}(t))^{\frac{p}{2}} \right) + \exp\left( -\frac{d_j^2 t^2}{16 \sigma_{X,2}^2} \right), \tag{104}$$

where

$$C_{1,p} = 2^{2p+1} \max\left( p^p, p^{p/2+1} e^p \int_0^\infty x^{p/2-1} (1-x)^{-p} dx \right),$$

and for any $k \in \{p, 2\}$,

$$r_{\mu,k}(t) = \sum_{i=1}^{\mu} \mathbb{E}\left[ \left| \frac{X_{i,j}'}{\sqrt{\mu}} \right|^k \mathbb{1}\left( \left| \frac{X_{i,j}'}{\sqrt{\mu}} \right| \geq \frac{3\sigma_{X,2}^2}{t d_j^2} \right) \right].$$

Now,

$$
\begin{aligned}
\mathbb{E}\left[ \left| \frac{X_{i,j}'}{\sqrt{\mu}} \right|^p \mathbb{1}\left( \left| \frac{X_{i,j}'}{\sqrt{\mu}} \right| \geq \frac{3\sigma_{X,2}^2}{t d_j^2} \right) \right] &= \int_{-\infty}^{\infty} \left| \frac{x}{\sqrt{\mu}} \right|^p \mathbb{1}\left( \left| \frac{x}{\sqrt{\mu}} \right| \geq \frac{3\sigma_{X,2}^2}{t d_j^2} \right) \frac{\eta_3 (|x| d_j)^{\eta_3 - 1}}{2 \left( 1 + (|x| d_j)^{\eta_3} \right)^2} dx \\
&= \int_{\frac{3\sigma_{X,2}^2 \sqrt{\mu}}{t d_j^2}}^{\infty} \left( \frac{x}{\sqrt{\mu}} \right)^p \frac{\eta_3 (x d_j)^{\eta_3 - 1}}{\left( 1 + (x d_j)^{\eta_3} \right)^2} dx \\
&\leq \frac{\eta_3}{d_j^{2p - \eta_3 + 1} \mu^{\frac{\eta_3}{2}} (\eta_3 - p)} \left( \frac{3\sigma_{X,2}^2}{t} \right)^{p - \eta_3} \\
&\leq C_2 d_j^{\eta_3 - 2p - 1} \mu^{-\frac{\eta_3}{2}} t^{\eta_3 - p}, \tag{105}
\end{aligned}
$$

where $C_2$ is a constant which depends on $\eta_3$ and $p$. Then

$$r_{\mu,p}(t) \leq C_2 d_j^{\eta_3 - 2p - 1} \mu^{1 - \frac{\eta_3}{2}} t^{\eta_3 - p}.$$

The term $r_{\mu,2}(t)$ can similarly be bounded as follows:

$$r_{\mu,2}(t) = \sum_{i=1}^{\mu} \mathbb{E}\left[ \left| \frac{X_{i,j}'}{\sqrt{\mu}} \right|^2 \mathbb{1}\left( \left| \frac{X_{i,j}'}{\sqrt{\mu}} \right| \geq \frac{3\sigma_{X,2}}{t d_j^2} \right) \right] \leq \frac{\sigma_{X,2}^2}{d_j^2}. \tag{106}$$

Using (105), and (106), from (104) we get

$$\mathbb{P}\left(|w_j| \geq t\right) \leq C_3 \left( d_j^{\eta_3 - 2p - 1} \mu^{1 - \frac{\eta_3}{2}} t^{\eta_3 - 2p} + d_j^{-2} t^{-p} \right),$$

for some constant $C_3 > 0$. ∎

**Lemma D.2.** *Let $w_1, w_2, \cdots, w_d$ are independent copies of a random variable such that* (102) *is true for all $j = 1, 2, \cdots, d$, i.e.,*

$$\mathbb{P}\left(|w_j| \geq t\right) \leq C_3 \left(d^{\eta_3 - 2p - 1}\mu^{1 - \frac{\eta_3}{2}}t^{\eta_3 - 2p} + d^{-2}t^{-p}\right) \quad j = 1, 2, \cdots, d,$$

*for $\eta_3 > 2 + 2\iota$, and $p = \eta_3 - 0.5\iota$. Let $\{w_j^*\}_{j=1}^d$ be the non-increasing arrangement of $\{|w_j|\}_{j=1}^d$. Then for every $1 \leq k \leq d$,*

$$\mathbb{E}\left[\left(\sum_{i=1}^k w_i^{*2}\right)^{\frac{1}{2}}\right] \leq C_6 \sqrt{k}\left(d^{\eta_3/(2p) - 1/2 + 1/p}d_1^{\eta_3/2 - p - \eta_3/p + 3/2 - 2/p}\mu^{1/2 - \eta_3/4} + d^{1/p}d_1^{-2/p}\right),$$

*for some constant $C_6 > 0$ which depends on $\eta_3$ and $p$.*

*Proof.* [Proof of Lemma D.2] First note that we have

$$\mathbb{P}\left(w_1^{*2} \geq t\right) = \mathbb{P}\left(w_1^* \geq \sqrt{t}\right) \leq \sum_{j=1}^d \mathbb{P}\left(|w_j| \geq \sqrt{t}\right) \leq C_3\left(dd_1^{\eta_3 - 2p - 1}\mu^{1 - \eta_3/2}t^{\eta_3/2 - p} + dd_1^{-2}t^{-p/2}\right).$$

Now, using (102), for any $v > 0$ (to be chosen later), we have

$$\begin{aligned}
\mathbb{E}\left[w_1^{*2}\right] &= \int_0^\infty \mathbb{P}(w_1^{*2} \geq t)dt \\
&\leq v + \int_0^\infty \mathbb{P}(w_1^{*2} \geq t + v)dt \\
&\leq v + \int_0^\infty C_3\left(dd_1^{\eta_3 - 2p - 1}\mu^{1 - \eta_3/2}(t + v)^{\eta_3/2 - p} + dd_1^{-2}(t + v)^{-p/2}\right)dt \\
&\leq v + C_4\left(dd_1^{\eta_3 - 2p - 1}\mu^{1 - \eta_3/2}v^{\eta_3/2 - p + 1} + dd_1^{-2}v^{1 - p/2}\right),
\end{aligned}$$

where $C_4 = C_3 \max(1/(p - 1 - \eta_3/2), 1/(p/2 - 1))$. Choosing $v = d^{2/p}d_1^{-4/p}$, we get

$$\mathbb{E}\left[w_1^{*2}\right] \leq C_5\left(d^{\eta_3/p - 1 + 2/p}d_1^{\eta_3 - 2p - 2\eta_3/p + 3 - 4/p}\mu^{1 - \eta_3/2} + d^{2/p}d_1^{-4/p}\right),$$

where $C_5 = C_4 + 1$. Using Jensen's inequality, we have

$$\begin{aligned}
\mathbb{E}\left[\left(\sum_{j=1}^k w_j^{*2}\right)^{\frac{1}{2}}\right] &\leq \left(\sum_{j=1}^k \mathbb{E}\left[w_j^{*2}\right]\right)^{\frac{1}{2}} \leq \left(k\mathbb{E}\left[w_1^{*2}\right]\right)^{\frac{1}{2}} \\
&\leq C_6\sqrt{k}\left(d^{\eta_3/(2p) - 1/2 + 1/p}d_1^{\eta_3/2 - p - \eta_3/p + 3/2 - 2/p}\mu^{1/2 - \eta_3/4} + d^{1/p}d_1^{-2/p}\right),
\end{aligned}$$

where $C_6 = \sqrt{C_5}$, thereby completing the proof. ∎

*Proof.* [Proof of Proposition 4.1] We start by obtaining a bound on the term $\omega_\mu(\mathcal{F}_R - \mathcal{F}_R, \tau Q_{\mathcal{H}_R}(2\tau)/16)$. Let $w$ be a random vector with coordinates

$$w_j = \frac{1}{\sqrt{\mu}}\sum_{i=1}^\mu X'_{i,j} \quad j = 1, 2, \cdots, d. \tag{107}$$

Let $\{w_j^*\}_{j=1}^d$ be the non-increasing arrangement of $\{|w_j|\}_{j=1}^d$. Then, we have

$$\begin{aligned}
\mathbb{E}\left[\sup_{f \in \mathcal{F}_R \cap s\mathcal{D}_{f^*}}\left|\frac{1}{\sqrt{\mu}}\sum_{i=1}^\mu \epsilon_i(f - f^*)(X'_i)\right|\right] &\leq \mathbb{E}\left[\sup_{t \in B_1^d(2R) \cap B_2^d(s)}\left\langle\frac{1}{\sqrt{\mu}}\sum_{i=1}^\mu X'_i, t\right\rangle\right] \\
= \mathbb{E}\left[\sup_{t \in B_1^d(2R) \cap B_2^d(s)}\langle w, t\rangle\right] &= s\mathbb{E}\left[\sup_{t \in B_1^d(2R/s) \cap B_2^d(1)} w^\top t\right] \leq 2s\mathbb{E}\left[\left(\sum_{j=1}^{(2R/s)^2} w_j^{*2}\right)^{\frac{1}{2}}\right].
\end{aligned}$$

If $(2R/s)^2 < d$, using Lemma D.2 we get

$$\mathbb{E}\left[\sup_{t \in B_1^d(2R) \cap B_2^d(s)} w^\top t\right] \leq 4C_6 R\left(d^{\eta_3/(2p)-1/2+1/p}d_1^{\eta_3/2-p-\eta_3/p+3/2-2/p}\mu^{1/2-\eta_3/4} + d^{1/p}d_1^{-2/p}\right)$$

$$\leq C_7 R d^{1/p+\iota/8},$$

when $d_1 \geq C_6'$ for some constants $C_6', C_7 > 0$.

If $(2R/s)^2 \geq d$,

$$\mathbb{E}\left[\sup_{t \in B_1^d(2R) \cap B_2^d(s)} w^\top t\right] \leq 2s\sigma_{X,2}\sqrt{d}.$$

So when $(2R/s)^2 \geq d$,

$$\mathbb{E}\left[\sup_{f \in \mathcal{F}_R \cap s\mathcal{D}_{f^*}} \left|\frac{1}{\mu}\sum_{i=1}^\mu \epsilon_i(f-f^*)(X_i')\right|\right] \leq \gamma s,$$

for all $s > 0$. When $\mu \leq C_8 d^{1+2/p+\iota/4}$, we have $(2R/s) \leq \sqrt{d}$ for

$$s \geq \frac{C_7 R d^{1/p+\iota/8}}{\gamma\sqrt{\mu}}.$$

When $\mu > C_8 d^{1+2/p+\iota/4}$, we have $(2R/s) \geq \sqrt{d}$ for

$$s \leq \frac{C_7 R d^{1/p+\iota/8}}{\gamma\sqrt{\mu}}.$$

Combining the above facts, and choosing $\mu = \frac{N^r Q_{\mathcal{H}}(2\tau)c^{\frac{1}{\eta_1}}}{4}$, and $r = 1 - 2\iota$ we get

$$\omega_\mu(\mathcal{F}_R - \mathcal{F}_R, \tau Q_{\mathcal{H}_R}(2\tau)/16) \leq \begin{cases} \frac{C_9 R}{\tau Q_{\mathcal{H}_R}(2\tau)^{3/2}}d^{1/p+\iota/8}N^{-1/2+\iota} & \text{if } \mu \leq C_8 d^{1+2/p}, \\ 0 & \text{if } \mu > C_8 d^{1+2/p}, \end{cases}$$

where $C_9 = 32C_7/c^{1/(2\eta_1)}$. Then using part 2 of Theorem 3.1, we get,

$$\|\hat{f} - f^*\|_{L_2} \leq \max\left\{N^{-\frac{1}{4}\left(1-\frac{1}{\eta_2}\right)+\iota}, \frac{C_9 R}{\tau Q_{\mathcal{H}_R}(2\tau)^{3/2}}d^{1/p+\iota/8}N^{-1/2+\iota}\right\},$$

with probability given by at least (14). ∎

# E   Proofs of Section 4.3

*Proof.* [Proof of Proposition 4.2] Since we assumed $X$ to be Gaussian, $\mathcal{F}_R$ is a $L_g$-subGaussian function class for some constant $L_g > 0$. So as shown in Section 6.5.2, we have,

$$\omega_Q(\mathcal{F} - \mathcal{F}, N, \zeta_1, \zeta_2) \leq \begin{cases} \frac{c_3(L_g)R}{\sqrt{N}}\sqrt{\log(ed/N)} & \text{if } N \leq c_1(L_g)d, \\ \frac{c_4(L_g)R}{\sqrt{d}} & \text{if } c_1(L_g)d < N \leq c_2(L_g)d, \\ 0 & \text{if } N > c_2(L_g)d, \end{cases}$$

where $c_i(L_g), i = 1, 2, 3, 4$ are constants dependent on only $L_g$. Then using Corollary B.2, we have

$$\|\hat{f} - f^*\|_{L_2} \leq \max\left(N^{-\frac{1}{2}+\iota}, 2\frac{c_3(L_g)R}{\sqrt{N}}\sqrt{\log(ed/N)}\right),$$

with probability at least (for some constants $c_9, c_{10}, \tilde{C}_2 > 0$)

$$1 - c_9\epsilon^{1-\frac{1}{\eta_1}}N^{\eta_1/(1+\eta_1)}e^{-c_{10}\epsilon^{1+\frac{1}{\eta_1}}N^{\eta_1/(1+\eta_1)}} - \tilde{C}_2 N\exp\left(-(N^{2\iota}\tau_0)^\eta/M_1\right).$$

∎

# F  A Note on Condition (c) in Assumption 2.1 and $\alpha_N^*(\gamma, \delta)$ in [Men15]

In this section, we discuss the relationship between Condition (c)-(i) of our Assumption 2.1 and the multiplier process based assumption in [Men15, Equation 2.2 and $\alpha_N^*(\gamma, \delta)$]. For simplicity, we consider the following simple model. Let $\{X_i\}_{i \in \mathbb{Z}^+}$ is an iid sequence of symmetric, zero-mean, random vectors. Let $\{Y_i\}_{i \in \mathbb{Z}^+}$, $Y_i \in \mathbb{R}$ denote the sequence given by $Y_i = {\theta^*}^\top X_i + \xi_i$, where $\theta^* \in B_1^1(R)$ and $\{\xi_i\}_{i=1}^N$ is an iid sequence and independent of $X_i$, $\forall i$, and $\xi_i \sim N(0, \sigma_1^2)$. The function class $\mathcal{F}$ we consider is $\mathcal{F} := \mathcal{F}_R = \left\{ \langle \theta, \cdot \rangle : \theta \in B_1^d(R) \right\}$. Now let us assume $\frac{1}{N} \sum_{i=1}^N X_i \xi_i$ is heavy-tailed random vector, with the tail lower bounded by $N\exp(-M(Nt))$, for some positive increasing function of $t$, $M(t)$, i.e., $\mathbb{P}\left( \left| N^{-1} \sum_{i=1}^N X_i \xi_i \right| > t \right) \geq M_3 N\exp(-M(Nt))$ for some $M_3 > 0$. Specifically setting $M(t) = t^\eta, \eta > 0$, and $M(t) = \eta_2 \log t, \eta_2 > 2$ one recovers (9) and (10). Now, recall from [Men15] that,

$$\alpha_N^*(\gamma, \delta) := \inf \left\{ s > 0 : \mathbb{P}\left( \sup_{\theta \in B_1^1(2R) \cap B_2^1(s)} \left| \frac{1}{N} \sum_{i=1}^N \xi_i X_i \theta \right| \leq \gamma s^2 \right) \geq 1 - \delta \right\}. \tag{108}$$

Note that, for $s > 0$,

$$\sup_{\theta \in B_1^1(2R) \cap B_2^1(s)} \left| N^{-1} \sum_{i=1}^N \xi_i X_i \theta \right| = \left| N^{-1} \sum_{i=1}^N \xi_i X_i \right| \min(2R, s).$$

We also have,

$$\mathbb{P}\left( \left| \frac{1}{N} \sum_{i=1}^N \xi_i X_i \right| \leq \gamma s^2 / \min(2R, s) \right) \leq 1 - M_3 N \exp\left( -M\left( \gamma N s^2 / \min(2R, s) \right) \right).$$

Then, from (108), when $s = \alpha_N^*(\gamma, \delta)$,

$$\delta \geq N M_3 \exp\left( -M\left( \gamma N \alpha_N^*(\gamma, \delta)^2 / \min(2R, \alpha_N^*(\gamma, \delta)) \right) \right).$$

Hence, if we want a non-trivial bound on the generalization error, we need $\alpha_N^*(\gamma, \delta)^2 \leq N^{-m_0}$ for some $m_0 > 0$. Set $2R > N^{-m_0/2}$. When $M(t) \sim t^{\gamma_2}, \gamma_2 > 0$, $\frac{1}{N} \sum_{i=1}^N \xi_i X_i$ has a sub-weibull tail. If it has a polynomially decaying tail, i.e., $M(t) = M_4 \log t$ for some constant $M_4 > 0$, then

$$\delta \geq N M_3 \exp\left( -M_4 \log\left( \gamma N^{1-m_0/2} \right) \right) = M_3 \gamma^{-M_4} N^{1 - (1 - \frac{m_0}{2})M_4}.$$

This implies that if $\frac{1}{N} \sum_{i=1}^N \xi_i X_i$ has a polynomially decaying tail, one gets a polynomial probability statement on the rate using complexity measure $\alpha_N^*(\gamma, \delta)$. Note that, since we are considering iid setting, choosing $m_0 < 1$ would allow $\alpha_N^*(\gamma, \delta)$ to be of the order of $N^{-1/2+\iota}$ where $\iota > 0$ is a small number. Recall that the rates we obtain in Theorem 3.1, and 3.2 are for $\beta$-mixing case. Indeed the worse rates are due to the presence of the third terms on the RHS of (9), and (10) – one needs to choose $\mathcal{A}(N)$ (used in the proofs of Theorem 3.1, and 3.2) suitably so that the third terms on the RHS of (9), and (10) decay to 0 as $N \to \infty$.