# OpenReview forum: "On Empirical Risk Minimization with Dependent and Heavy-Tailed Data"
_NeurIPS.cc/2021/Conference — NeurIPS 2021 Poster_

### Official Review · Reviewer_v9BD · 2021-07-04

**Rating:** 7
**Confidence:** 4

**Summary:**

The paper considers the performance of ERM on heavy-tailed and dependent data. It proves general convergence rates with a few illustrative examples.

Update
----
I have read the author(s)' response to my review and maintain my rating.

**Limitations And Societal Impact:**

It would be nice if the author(s) could provide lower bound to complement their results. This is generally doable in the heavy-tailed statistics and small-ball method literature. I wonder if the author(s) have given any thought.

As this is a theory work, I believe it will not lead to negative societal impact.

**Main Review:**

Originality
---
The problem considered by the paper is new. The setting of dealing with heavy-tailed *and* dependent data has not been studied in the literature.


Quality
---
This is a complete work with full set of theoretical results. I've read the main paper and some proofs in the appendix and do not notice any major issue.

The author(s) has acknowledged the assumptions and potential weaknesses of the work.

Clarity
---
The writing is generally clear. However, I find some of the exposition in the main paper reads overly technical. It makes sense to provide more intuitions when applicable.

A missing reference might be "Learning from Weakly Dependent Data under Dobrushin’s Condition" (COLT 19), where the dependence is modeled by graphical model assumptions.


**Time Spent Reviewing:**

2

---

> ### Author Response · Authors · 2021-08-10
> **Response to Reviewer v9BD**
>
> We thank the reviewer for the positive feedback on our work. We are currently working on the lower bounds and plan to report our results in the future. Thanks for point out the interesting paper -- we will include a discussion in the revision.

---

> > ### Comment · Reviewer_v9BD · 2021-08-26
> > **reply**
> >
> > Thank you for the response. I maintain my rating and recommend accept.

---

### Official Review · Reviewer_ynYv · 2021-07-09

**Rating:** 6
**Confidence:** 3

**Summary:**

The authors derive risk bounds for empirical risk minimization under heavy-tailed $\beta$-mixing observations. They provide general bounds for convex, locally strongly convex, loss functions and tighter bounds for the squared loss. They give three illustrative models for which they provide explicit bounds, the most important one being the Huber loss for which they find a justification of its usage as a robust lost in presence of sub-Weibull noise distributions.

**Ethical Concerns:**

None.

**Limitations And Societal Impact:**

The main motivation, i.e., explaining the role of Huber loss as a robust loss for heavy tailed data, is not fully achieved as derivations are limited to Gaussian noise (which is restrictive, if relevant at all).

**Main Review:**

The contribution is timely and novel. The high level of technicalities makes the paper a though read, especially for a venue such as NeurIPS where the space is limited; a large portion of the bibliography refers to work published in Annals of Statistics, which could be a better fit for this kind of work.

Apart from the above point, there a few minor typos/presentation issues:
* p.2 l.39: could you provide real world/practical examples of dependent GDP? You are here mostly referring to theoretical work.
* p.2 l.42 and so on: you are using "non-iid" everywhere in the paper to refer to not independent but the the $Z_i$ are still identically distributed (as the DGP is supposed stationary). You thus shouldn't use "non-iid" instead of non independent DGP as "non-iid" could mean non-independent, non id or both.
* p.2 l.44: the distribution is $\beta$-mixing not the data.
* p.2 l.70: PAC is not defined.
* p.3 l.87: which $p^{th}$ moment? Any?
* p.3 l.88: what is $\alpha$?
* p.3 Def 1: the notion of stationarity considered here should be made explicit.
* p.3 eq. (1): shouldn't it be $P_N\ell_f$?
* p.3 l.106: $\xi_i$ is nowhere defined in the paper.
* p.3 l.108: RHS is not defined.
* p.3 l.119: what is $\xi_\infty$?
* p.3 Assumption 2.1: it is not clear that you will require either (c)-(i) or (c)-(ii), maybe add either after "satisfy''?
* p.5 l.165: "we only require local ...' what do you require exactly? For them to exists? To be finite?
* p.6 l.203: it should be (c)-(i) and not (3)-(i).
* p.6 Corollary 3.1: does this hold for both (c)-(i) and (c)-(ii)?
* p.6 l.233: "our result''.
* p.7 l.234: why introducing the notion of local convexity only now when it has been already used multiple time? This should be introduced the first time the notion is used.
* p.7 l.246: what is a "robust loss'' in this context? I guess that the authors have in mind the Huber loss studied later but this unclear at that stage.
* p.9 l.316: "similar result will hold''. Does it hold or will you prove it in future work? If the proof is not provided (I couldn't find it in Appendices) this significantly limits the claims of the authors and their capacity to prove the "robustness" of the Huber loss.
* p.9 l.350: it is not clear how the results currently show the advantage of the Huber loss as Example 3 is limited to Gaussian (sw?) case. Please, modify the sentence to better reflect the precise contribution of the paper.

**Time Spent Reviewing:**

4

---

> ### Author Response · Authors · 2021-08-10
> **Response to Reviewer ynYv**
>
> We thank the reviewer for the careful reading and suggestions for improving presentation.
>
>
> We first address the comment: “The main motivation, i.e., explaining the role of Huber loss as a robust loss for heavy tailed data, is not fully achieved as derivations are limited to Gaussian noise (which is restrictive, if relevant at all).” and the related point raised under bullet point p.9 l.316, p.9 l.350.
>
>
>
> Please note that we address the case of heavy-tailed data in Example 3. In example 3, we assume that only the **variance of the noise ($\xi_i$), and consequently response ($Y_i$) exists**. In other words, we just assume the **tail of the distribution of the noise to decay polynomially** with exponent $\eta_4>2$ **unlike Gaussian noise which has exponential tail**. It is only $X_i$ which is assumed to be subgaussian. So the benefits of Huber loss can still be realized from this example.
>
> Response to other minor typos/comments:
>
> p.2 l.39: Indeed, (exponential) mixing assumption holds in several time-series appliations. For example, [R1] showed that certain ARMA processes can be modeled as an exponentially $\beta$-mixing stochastic process. Furthermore [R2] showed that globally exponentially stable ``unforced'' dynamical systems subjected to finite-variance continuous density input noise give rise to exponentially $\beta$ mixing stochastic process; see also [R3] and [R4]. Such data generating process has motivated researchers to develop learning theory results under mixing assumptions (see, for example, the papers cited under related works in lines 64-79). We will add the above specific examples to the revision as concrete motivation.
>
> p.2 l.42, p.2 l.44, p.2 l.70: We will make the changes.
>
> p.3 l.87:
> As we have mentioned in l.87 $p\geq 1$.
>
> p.3 l.88: The parameter $\alpha$ is the exponent of the entropy condition, for example, from [7, Example 4]. Such conditions are common in the analysis of empirical risk minimization procedures. We will clarify this in our revision.
>
> p.3 Def 1:
> We will change it to “strictly stationary”.
>
> p.3 eq. (1):
> We will make the change.
>
> p.3 l.106:
> $\xi_i$ is the noise sequence in the sample setting,  corresponding to the noise $\xi$ in the population setting defined in p.1 l.28. We will define $\xi_i$ explicitly.
>
> p.3 l.108:
> We will define it.
>
> p.3 l.119:
> Please note that the sequence $\xi_i$, for $i=1, 2, \ldots$ is a strictly stationary sequence. We present the specific assumption in terms of $\xi_\infty$. Due to strict stationarity, a distributional property holding for $\xi_\infty$ means that the same property holds for each $\xi_i$. We will change it to $\xi_1$ to avoid confusion.
>
> p.3 Assumption 2.1:
> We will.
>
> p.5 l.165:
> We meant to say that the bounds will be in terms of local Rademacher-based complexity measure. We will make it clearer.
>
> p.6 l.203:
> We will change it.
>
> p.6 Corollary 3.1:
> This is under (c)-(i). We will state it clearly.
>
> p.6 l.233:
> We will make the change.
>
> p.7 l.234:
> We will state it before.
>
> p.7 l.246:
> We will state it clearly.
>
> p.9 l.316:
> We have not provided the proof here. Although we want to point out that, **it is only $X$ which is assumed to sub-Gaussian and not the response or the noise as we have pointed above**.
>
> p.9 l.350:
> Example 3 is not limited to Gaussian case. Please see our comment above.
>
>
> $[R1]$ A. Mokkadem, Mixing Properties of ARMA Processes, Stochastic Processes and their Applications 29 (2) (1988) 309–315.
>
> $[R2]$ M. Vidyasagar, R. L. Karandikar, A Learning Theory Approach to System Identification and Stochastic Adaptive Control, Journal of Process Control 18 (3–4) (2008) 421 – 430.
>
> $[R3]$ Amir-massoud Farahmand and Csaba Szepesvari. Regularized least-squares regression: Learning from a $\beta$-mixing sequence. Journal of Statistical Planning and Inference, 142(2):493 – 505, 2012.
>
> $[R4]$ Wong, Kam Chung, Zifan Li, and Ambuj Tewari. "Lasso guarantees for $\beta $-mixing heavy-tailed time series." The Annals of Statistics 48, no. 2 (2020): 1124-1142.

---

> > ### Comment · Reviewer_ynYv · 2021-08-26
> > **Answer**
> >
> > Thank for taking the time to reply to all my comments, that have been addressed satisfactorily. I would however maintain my current score.

---

> > > ### Author Response · Authors · 2021-08-26
> > > **reply**
> > >
> > > Thanks for the update. Please let us know if you have any other additional concerns and we will be happy to clarify them.

---

### Official Review · Reviewer_MgVF · 2021-07-15

**Rating:** 8
**Confidence:** 3

**Summary:**

The paper analyses empirical risk minimization over a locally strongly convex function class in the context where the data generating process returned non-IID data, and can be potentially heavy-tailed. The authors give specific realizations of their analysis to sparse linear function class.

This work aims to bridge the gap between analyses that handle IID data and heavy-tailed data separately.

**Limitations And Societal Impact:**

This is a primarily theoretical work, and there is no immediate potential negative societal impact arising from this work.

**Main Review:**

### Originality
The work extends previously known analyses in non-trivial directions.

### Quality
I believe the submission is technically sound to the best of my understanding. The authors don't discuss any apparent weakness in their analyses, or any scope of improvement. The work is a complete in the sense that the authors have studied a very specific problem (combination of heavy-tailed and non-IID data for locally convex losses) and address it.

### Clarity
The submission is well organized, and is clearly written. However, the reader would benefit from the main messages of the propositions in this work. For instance, while the precision of the theorem statement in Theorem 3.1 is appreciated, but the presence of too many symbols somewhat obscures the essence of the theorem. A similar statement could be made with regard to the readability of Theorem 3.2; perhaps it would have been beneficial to add some remarks distilling the statement of the theorem (as done for Theorem 3.1). I definitely don't view this as grounds to rejecting the work, but this can immensely help with how accessible such results can be.

### Significance
Due to the non-triviality of this analysis, I believe it is significant to members of the statistical learning community. However, there is no apparent practical significance, as the authors analyse a commonly used and known methodology.

-----

Additional detailed comments below:

- Line 106: what is $\xi\_{i}$? Is it $Y\_{i} - f^{*}(X\_{i})$?

- Is Assumption 2.1b equivalent to saying that there exists a function in $f \in \mathcal{F} - \mathcal{F}$ such that $|f| \geq 2\tau ||f||\_{L\_{2}}$? I am confused about the probability of this event.

- Is Assumption 2.1c roughly stating that probability of "small" deviations (exponentially) decreases faster than probability of "large" deviations (polynomially)? Could the authors give an example of a simple random variable that satisfies this specific condition for better intuition of the assumption?

- In Assumption 2.1d, is it the case that $\eta$ is fixed; in which case it is clear why that condition can be referred to as a "tradeoff".

- In Definition 2 of the Generalized Rademacher complexity, could the authors elaborate about how this is an extension of the Rademacher complexity to non-IID  setting, as the definition states that $\{\tilde{X}\_{i}\}\_{i=1}^{\mu}$ is an IID dataset? Also could the authors comment on how one could expect $\omega\_{\mu}(\mathcal{F} - \mathcal{F}, \gamma)$ to vary with $\mu$, or point to some results that discuss this?

- In Definition 3: how do you define the supremum over Gaussian processes $G_{h}$?

- In Lemma 2.1: what is the $\vee, \wedge$ notation? Is it is the minimum, maximum shorthand?

- As the authors remark, the speciality of Lemma 2.2 lies in its generality, as setting one can recover existing results for the IID case from the statement of the lemma.

- Based on my understanding, Theorem 3.1 tells us that for the the squared loss, the error between ERM and the true risk minimizer is atmost a constant (or decays with N, albeit slowly) with probability at least 1 - $\delta$, where $\delta$ is at most $O(N^r \exp(-N^{1-r}) + N \exp(-N^\text{cons}\_{1}))$ (when 2.1c-i is satisfied), and $O(N^r \exp(-N^{1-r}) + N^{-\text{cons}\_{2}})$ (when 2.1c-ii is satisfied). Is this correct?

- There is a typo on lines 203, 205: it should read "conditions (a), (b), (c)-(i)" (and "(c)-(ii)") respectively.

- Example 4.1 is a very simple and interesting example, whereas example 4.2 seems to be slightly contrived. Could the authors provide some motivation to study either example: are they merely provided to show the generality of the proven results?
If so, the authors could have included some proof sketches for these examples and show how the original theorem have been directly utilized.

**Time Spent Reviewing:**

2.5

---

> ### Author Response · Authors · 2021-08-10
> **Response to Reviewer MgVF**
>
> We thank the reviewer for the positive feedback, careful reading and suggestions for improving presentation. Answers for detailed comments are provided below.
>
> 1.	Yes.
>
> 2. Assumption 2.1b says that there is a $\tau$ such that for all functions $ f \in \mathcal{F}-\mathcal{F} $, we have, $P(|f|\geq2\tau \|\|f\|\|_{L_2} )>0$.
>
> Intuitively, it models heavy-tailedness by restricting the mass allowed near any small neighborhoods of zero; thus, forcing the tails to be necessarily heavy. For example, as shown in Lemma 4.1 of [1], this holds if
>
> $\|\|f\|\|_{L_a} < k_2 $
> $\|\|f\|\|_{L_b}$
> where $ k_2 > 0 $ is a constant, $a=2$, and $b=1$; or $a>2$, and $b=2$. This norm equivalence holds for linear function class when the $X$ follows a sub-Weibull distribution.
>
> 3.	Define $V:=\xi_\infty (f-f^*)(X_\infty)$. Then Assumption 2.1c (i), and (ii) are equivalent to assuming random variable $V$ to be sub-Weibull (e.g. Weibull, Exponential, Gaussian random variable) and having Polynomial tail (e.g. Pareto, Cauchy, $t$ random variable) respectively.
>
> 4.	Yes $\eta$ is fixed with the condition $1/\eta>1$. In other words, $\eta_2<1+1/(\eta_1-1)$.
>
> 5.	In Definition 2 the original data is from a strictly stationary sequence, and $\mu$ depends on $N$. For example, in Theorem 3.1 $\mu=N^rQ_{\mathcal{H}}(2\tau)c^{1/\eta_1}/4$. Given $N$, $\mu$ can be interpreted as an effective sample size from the stationary distribution. In the proof of Proposition 4.1, 4.2, and 4.3, we show that $\omega_\mu$ is $O(1/\sqrt{\mu})$ (ignoring other parameters). The derivations are in the supplementary material.
>
> 6.	Here $G_h$ denotes a mean-zero canonical Gaussian process indexed by $h\in\mathcal{H}$. Please note that here the supremum is taken over the index set $\mathcal{H}$. A simple example to think of is case of $\mathcal{H}$ being $d$-dimensional linear functionals of the form $\langle v, \cdot \rangle$ where $\langle \cdot, \cdot \rangle$ is the Euclidean inner product. In this case, the supremum is over the set of $d$-dimensional linear functionals characterized by $\mathbb{R}^d$.
>
> 7.	Yes. We will explictly mention this in the revision.
>
> 8.	We agree.
>
> 9.	Yes. The error between the true minimizer and ERM decays with $N$ when $\omega_\mu$ decays with $N$ as we show in our examples.
>
> 10.	Thanks for pointing this out. We will fix it.
>
> 11.	In Example 1, $X$ is being generated from an $AR(1)$ process which is an important and common time-series to study. $AR(1)$ with heavy-tailed innovation arises in financial data analysis [A1]. In Example 2, the data is being generated from a 50-50 mixture of two semi-Pareto processes which has applications in hydrological studies, and reliability studies (see [A2]).
>
> We show that the assumptions required to apply Theorem 3.1 or Theorem 3.2 for Proposition 4.1, 4.2, and 4.3 are satisfied, in Section C.1, D.1, and E.1 respectively. For the lack of space we could not include it in the main paper. However, as an additional page will be allowed for camera-ready version to address reviewers' concerns, we will add proof sketches as suggested.
>
> $[A1]$ Emberchts P., Klüppelberg C., Mikosch T. (1997) Time Series Analysis for Heavy-Tailed Processes. In: Modelling Extremal Events. Applications of Mathematics (Stochastic Modelling and Applied Probability), vol 33. Springer, Berlin, Heidelberg.
>
> $[A2]$ RN Pillai. Semi-pareto processes. Journal of applied probability, pages 461–465, 1991.

---

> > ### Comment · Reviewer_MgVF · 2021-08-21
> > **Thank you for your rebuttal**
> >
> > Thank you for presenting clarifications / answers to my questions above. I will be preserving my rating in light of this, and believe that this work is complete and can be accepted.

---

### Official Review · Reviewer_9G9p · 2021-07-19

**Rating:** 6
**Confidence:** 3

**Summary:**

This submission is a technical paper, which is concerned with obtaining bounds for the Empirical Risk Minimization (ERM) procedure for regression, when the data is both heavy-tailed and dependent (non-iid).
The performance of ERM in the heavy-tailed context is by now fairly well-understood, in part thanks to the observation that two-sided concentration for the empirical risk is not needed to establish excess risk bounds for ERM, and that one-sided guarantee suffice (functions with large population risk have large empirical risk).
The authors' contribution is to extend previous results in the iid setting to the dependent (specifically, mixing or exponentially-mixing) setting.
While both the heavy-tailed and dependent settings have been considered separately, this works combines these two cases.

As in the iid case, the analysis involves both obtaining lower isomorphic (small ball) estimates for the function class, and controlling the interaction between the noise and the functions in the class (the so-called multiplier inequality).

The main applications of the generic results obtained for square and convex loss (Section 3) are described in Section 4.
For regression over L^1 balls, the authors obtain bounds under square loss for both exponential and polynomial tails of the noise, as well as bounds for Huber loss under polynomially-tailed noise (but sub-Gaussian data).

**Limitations And Societal Impact:**

yes

**Main Review:**

In terms of quality, this seems to be a technically sound and reasonable paper. Due to a high number of papers to review, I did not check the proofs in the 30 pages of seemingly dense and technical appendix, but I did read the main text.

The authors could do a better job in terms of clarity, for instance in the way their bounds are stated. For instance, essentially all bounds are stated in the following way (simplifying a bit):
"with probability 1 - f(N) - exp (-N^{2i})", the risk is bounded by max(g(N), N^{-1/2 + i})..."
which is a somewhat convoluted way of writing the bound (the dependence on the confidence is in the exponent 2i of the sample size N in the bound); it would be much more readable to write e.g. "for delta < h(N), with probability 1-delta, the risk is bounded by max( {log(1/delta)/N}^{1/2}, l(N) )".

In terms of significance, the motivation for studying the dependent case is not completely clear in the present version, and without some justification it could seem like a somewhat arbitrary extension of heavy-tailed statistical learning. However, this may be due to the fact that I am not familiar with the dependent setting. Perhaps this could be motivated by arguing that the (exponential) mixing assumptions hold for time series, when covariates are moving windows of present and recent variables, if this is true? If not, I feel some reasonable example or motivation would help.

Although the paper seems to be mainly about combining techniques of the dependent case with those of the heavy-tailed setting, one aspect I did appreciate is that the authors provided an explicit control of the multiplier process under some precise assumptions on the noise-function interaction, a step which is not done in Mendelson's JACM paper.

I wonder whether Neurips really is the right venue for such a paper. This is a technical contribution, combining techniques from different settings to obtain results when data is both heavy-tailed and dependent.
It is 40 pages long, with fairly technical proofs. Given ML conferences' reviewing model, it seems unlikely that this submission will be properly checked and reviewed.
From the authors' perspective, this means that acceptance to Neurips would be meaningless for such a paper. From the perspective of the conference, it could be that this technical extension/combination paper would mainly appeal to a small fraction of the Neurips audience, given the nature of the contribution, the topic and the way the results are presented (although the latter can be remedied, e.g. by highlighting the results of Section 4 and simplifying their formulation).

This being said, I do think this work could be presented at Neurips, in spite of the seemingly less-than-ideal fit. I hesitated to put a higher recommendation (in spite of some reservations on the motivation and clarity), but prefer to first see how this submission is received by other referees.

Minor comments:

* The fact that the rates for square loss are worse under the small-ball than the moment equivalence (n^{-1/4} vs n^{-1/2}) is strange. Indeed, both conditions typically serve to obtain small ball estimates on the class, which they both provide (though the small-ball assumption should lead to a worse leading constant, involving the Q function). What is this worse rate due to? (the control of the multiplier term?)

* The (polynomial) dependence on d in the sparsity bound of Proposition 4.2 is not very readable (is the exponent small or large?).

* It should be clarified in Assumption 2.1 that conditions (2) and (3) are separate conditions ("or" rather than "and", with the understanding that 2 is stronger than 3).

* Line 256: "... sub-Weibull random variables and Pareto random variables, that are canonical models of heavy-tailed data in the literature."
Calling sub-Weibull distributions heavy-tailed is a stretch, since they exhibit tails decaying exponentially in t. While the term 'heavy tails' does not have a formal definition, it is more suitable to describe heavier (e.g. polynomial) tails.

# Update

I thank the authors for their detailed response. Overall, I would maintain my initial evaluation: I think the paper can certainly be accepted, but also that it would substantially benefit from a significant revision, in particular in how the results are stated. In my opinion it would be in the authors' interest to do this for the sake of readability, regardless of whether the paper is accepted for presentation. This should not be interpreted as a strong criticism or reservation; in fact, I did hesitate to raise my score, but decided to stand by my initial evaluation after taking another look at the paper.

**Time Spent Reviewing:**

4

---

> ### Author Response · Authors · 2021-08-10
> **Response to Reviewer 9G9p**
>
> We thank the reviewer for the detailed comments and suggestions for improving presentation.
>
> **Clarity of Theorem statements:** Thanks for the suggestion. We will use the format suggested in the revision.
>
> **Motivation:**  Indeed, (exponential) mixing assumption holds in several time-series appliations. For example, [R1] showed that certain ARMA processes can be modeled as an exponentially $\beta$-mixing stochastic process. Furthermore [R2] showed that globally exponentially stable "unforced" dynamical systems subjected to finite-variance continuous density input noise give rise to exponentially $\beta$ mixing stochastic process; see also [R3] and [R4]. Such data generating process has motivated researchers to develop learning theory results under mixing assumptions (see, for example, the papers cited under related works in lines 64-79). We will add the above specific examples to the revision as concrete motivation.
>
>
> **Explicit control of the multiplier process:** Thanks for pointing this out. This is one of our major contributions for the non-iid setting as pointed out in lines 44-51.
>
> **The fact that the rates for square loss are worse under ...:** Norm equivalence is sufficient condition for small-ball condition but not necessary. For example, small-ball condition can hold even if no moment greater than the second moment exists as we show in Example 2 and 3. So norm equivalence is a strictly stronger assumption.
>
> The worse rate is indeed due to the multiplier term. Recall $V$ defined in (8). Under norm equivalence one can show that $V$ has an upper bound proportional to $||f-f^*||_{L_2}^2$ (page 23, Line 660) whereas under only small-ball condition it is upper bounded by a constant (page 21, Line 635).
>
>
> **(polynomial) dependence on $d$:** Ignoring terms depending on $\iota$ the polynomial dependence is $d^{1/\eta_2}$.
>
> **Assumption 2.1 -- conditions (2) and (3) are separate conditions:** We will clarify this.
>
> **Line 256:** We used the terminology in recent literature (for example [63]) referring to subWeibull distributions as heavier-tailed distributions than, for example, sub-exponential distributions. However, we agree it is a stretch, and we will change it in the revision as suggested.
>
>
> Finally, taking the reviewer's suggestions (by highlighting the results of Section 4 and simplifying their formulation), we will improve the presentation to make our work widely accessible. We would also like to highlight that several important prior works on learning theory in the non-iid setting appeared at NeurIPS (including [46, 60]), and motivated us to work on this particular problem.
>
>
> $[R1]$ A. Mokkadem, Mixing Properties of ARMA Processes, Stochastic Processes and their Applications 29 (2) (1988) 309–315.
>
> $[R2]$ M. Vidyasagar, R. L. Karandikar, A Learning Theory Approach to System Identification and Stochastic Adaptive Control, Journal of Process Control 18 (3–4) (2008) 421 – 430.
>
> $[R3]$ Amir-massoud Farahmand and Csaba Szepesvari. Regularized least-squares regression: Learning from a $\beta$-mixing sequence. Journal of Statistical Planning and Inference, 142(2):493 – 505, 2012.
>
> $[R4]$ Wong, Kam Chung, Zifan Li, and Ambuj Tewari. "Lasso guarantees for $\beta $-mixing heavy-tailed time series." The Annals of Statistics 48, no. 2 (2020): 1124-1142.

---

### Decision · Program_Chairs · 2021-09-27

**Decision:**

Accept (Poster)

**Comment:**

The paper focuses on obtaining bounds for the Empirical Risk Minimization (ERM) procedure for regression, when the data is both heavy-tailed and non-iid/dependent.  This work combined both the heavy-tailed and non-iid case.  Reviewers were positive.  However, the work is rather technical, and they correctly point out that the authors would make the paper substantially more accessible to the typical NeurIPS reader if they spend time on incorporating reviewer suggestions for improving the presentation.